# FIXED-BUDGET BEST ARM IDENTIFICATION WITH VARIANCE-DEPENDENT REGRET BOUNDS

## ABSTRACT

We investigate the problem of fixed-budget *best arm identification* (BAI) for minimizing expected simple regret. In an adaptive experiment, a decision-maker draws one of multiple arms based on past observations and observes the outcome of the drawn arm. At the end of the experiment, the decision-maker recommends an arm with the highest expected outcome. We evaluate the decision based on the *expected simple regret*, which is the difference between the expected outcomes of the best arm and the recommended arm. Due to inherent uncertainty, we consider the *worst-case* analysis for the expected simple regret. First, we derive asymptotic lower bounds for the worst-case expected simple regret, which are characterized by the variances of potential outcomes (leading factor). Based on the lower bounds, we propose the *Adaptive-Sampling* (AS)-*Augmented Inverse Probability Weighting* (AIPW) strategy, which utilizes the AIPW estimator in recommending the best arm. Our theoretical analysis shows that the AS-AIPW strategy is asymptotically minimax optimal, meaning that the leading factor of its worst-case expected simple regret matches our derived worst-case lower bound. Finally, we validate the proposed method's effectiveness through simulation studies. To improve efficiency, we also discuss the use of contextual information as a generalization of the standard BAI, though our result holds novelty even without contextual information.

## 1 INTRODUCTION

We consider adaptive experiments with multiple arms, such as slot machine arms, different therapies, and distinct unemployment assistance programs. Rather than performing hypothesis testing or estimating the expected outcomes, our focus is on identifying the best arm, which is an arm with the highest expected outcome. This task is known as the best arm identification (BAI) and is a variant of the stochastic multi-armed bandit (MAB) problem (Thompson, 1933; Lai & Robbins, 1985). In this study, we investigate *fixed-budget BAI*, where we aim to minimize the expected simple regret after an adaptive experiment with a fixed number of rounds, known as a *budget* (Bubeck et al., 2009; 2011).

In each round of our experiment, a decision-maker sequentially draws one of the arms based on past observations and immediately observes a corresponding outcome of the drawn arm generated from a bandit model. At the end of the experiment, the decision-maker recommends an estimated best arm at the end of the experiment. The decision-maker decides its action following a *strategy*. We measure the performance of the strategy using the expected simple regret, which is the difference between the maximum expected outcome that could be achieved with complete knowledge of the distributions and the expected outcome of the arm recommended by the strategy. Due to the inherent uncertainty, we evaluate the performance under the worst-case criterion among a given class of bandit models.

For bandit models with finite variances, we derive worst-case lower bounds for the expected simple regret. The lower bounds' leading factors are characterized by the variances of potential outcomes. We then propose the Adaptive-Sampling (AS)-Augmented Inverse Probability Weighting (AIPW) strategy and show that it is asymptotically minimax optimal, meaning that the leading factor of its worst-case expected simple regret matches that of the lower bound.

Bubeck et al. (2011) derives worst-case lower bounds for the expected simple regret under bandit models with bounded supports. Their lower bounds only rely on the boundedness of the bandit models and do not depend on any other distributional information. This study derives lower bounds that depend on the variances of the bandit models, which means that we use more distributional

information. Our lower bounds are based on change-of-measure arguments with the Kullback-Leibler (KL) divergence, which has been used to derive tight lower bounds in existing studies (Lai & Robbins, 1985; Kaufmann et al., 2016). Note that the variance appears as the second-order expansion of the KL divergence where the gaps between the expected outcomes of best and suboptimal arms are zero.

In BAI, the proportion of times each arm is drawn plays a critical role, referred to as a target allocation ratio. In many settings, target allocation ratios do not have closed-form solutions. However, in our case, we derive analytical solutions for several specific cases characterized by the standard deviations or variances of the outcomes. When there are only two arms, the target allocation ratio is the ratio of the standard deviation of outcomes. When there are three or more arms without contextual information, the target allocation ratio is the ratio of the variance of outcomes.

This result contrasts with the findings of Bubeck et al. (2011), which reports that a strategy drawing each arm with an equal ratio and recommending an arm with the highest sample average of observed outcomes is minimax optimal for bandit models with bounded outcome supports. In contrast, our results suggest drawing arms based on the ratio of the standard deviations or variances of outcomes. This difference stems from the use of distributional information (variances) in experimental design.

Furthermore, to improve efficiency, we additionally consider a scenario where a decision-maker can observe contextual information before drawing an arm. Unlike in the conventional contextual bandits, our goal still lies in the identification of the arm with the highest unconditional expected outcome rather than the conditional expected outcome. This setting is motivated by the goals of average treatment effect (ATE) estimation (van der Laan, 2008; Hahn et al., 2011) and BAI with fixed confidence and contextual information (Russac et al., 2021; Kato & Ariu, 2021). Our findings indicate that utilizing contextual information can reduce the expected simple regret, even if our focus is on the unconditional best arm. Note that this setting is a generalization of fixed-budget BAI without contextual information, and our result holds novelty even in the absence of contextual information.

In summary, our contributions include (i) lower bounds for the worst-case expected simple regret; (ii) analytical solutions for the target allocation ratios, characterized by the variances of the outcomes; (iii) the AS-AIPW strategy; (iv) asymptotic minimax optimality of the AS-AIPW strategy; These findings contribute to a variety of subjects, including statistical decision theory, in addition to BAI.

**Organization.** Section 2 formulates our problem. Section 3 establishes lower bounds for the worst-case expected simple regret and a target allocation ratio. In Section 4, we introduce the AS-AIPW strategy. Then, Section 5 presents the upper bounds with its asymptotic minimax optimality. Finally, we introduce related work in Section 6 and Appendix A. Open problems are discussed in Appendix J.

## 2 PROBLEM SETTING

We investigate the following setup. Given a fixed number of rounds $T$, referred to as a budget, in each round $t = 1, 2, \dots, T$, a decision-maker observes a contextual information (covariate) $X_t \in \mathcal{X}$ and draws arm $A_t \in [K] = \{1, 2, \dots, K\}$. Here, $\mathcal{X} \subset \mathbb{R}^D$ is a space of contextual information[1]. The decision-maker then immediately observes an outcome (or reward) $Y_t$ linked to the drawn arm $A_t$. This setting is referred to as the bandit feedback or Neyman-Rubin causal model (Neyman, 1923; Rubin, 1974), in which the outcome in round $t$ is $Y_t = \sum_{a \in [K]} \mathbb{1}[A_t = a] Y_t^a$, where $Y_t^a \in \mathbb{R}$ is a potential independent (random) outcome, and $Y_t^1, Y_t^2, \dots, Y_a^K$ are conditionally independent given $X_t$. We assume that $X_t$ and $Y_t^a$ are independent and identically distributed (i.i.d.) over $t \in [T] = \{1, 2, \dots, T\}$. Our goal is to find an arm with the highest expected outcome marginalized over the contextual distribution of $X_t$ with a minimal expected simple regret after the round $T$.

We define our goal formally. Let $P$ be a joint distribution of $(Y^1, Y^2, \dots, Y^K, X)$, and $(Y_t^1, Y_t^2, \dots, Y_t^K, X_t)$ be an i.i.d. copy of $(Y^1, Y^2, \dots, Y^K, X)$ in round $t$. We refer to the distribution of the potential outcome $(Y^1, Y^2, \dots, Y^K, X)$ a full-data bandit model (Tsiatis, 2007). For a given full-data bandit model $P$, let $\mathbb{P}_P$ and $\mathbb{E}_P$ denote the probability law and expectation with respect to $P$, respectively. Besides, let $\mu^a(P) = \mathbb{E}_P[Y_t^a]$ denote the expected outcome marginalized over $X$. Let $\mathcal{P}$ denote the set of all $P$. An algorithm in BAI is referred to as a *strategy*. With the sigma-algebras $\mathcal{F}_t = \sigma(X_1, A_1, Y_1, \dots, X_t, A_t, Y_t)$, we define a strategy as a pair $((A_t)_{t \in [T]}, \widehat{a}_T)$, where $(A_t)_{t \in [T]}$ is a sampling rule and $\widehat{a}_T \in [K]$ is a recommendation rule. Following the sampling

---

[1]BAI without contextual information corresponds to a case where $\mathcal{X}$ is a singleton.

rule, the decision-maker draws arm $A_t \in [K]$ in each round $t$ based on the past observations $\mathcal{F}_{t-1}$ and observed context $X_t$. Following the recommendation rule, the decision-maker returns an estimator $\widehat{a}_T$ of the best arm $a^*(P) := \arg\max_{a \in [K]} \mu^a(P)$ based on observations up to round $T$. Here, $A_t$ is $\mathcal{F}_t$-measurable, and $\widehat{a}_T$ is $\mathcal{F}_T$-measurable. For a bandit model $P \in \mathcal{P}$ and a strategy $\pi \in \Pi$, let us define the simple regret as $r_T(P)(\pi) := \max_{a \in [K]} \mu^a(P) - \mu^{\widehat{a}_T}(P)$. Our goal is to find a strategy that minimizes the expected simple regret, defined as

$$\mathbb{E}_P[r_T(P)(\pi)] = \mathbb{E}_P\left[\max_{a \in [K]} \mu^a(P) - \mu^{\widehat{a}_T}(P)\right] = \sum_{b \in [K]} \underbrace{\Delta^b(P)}_{\text{gap}} \underbrace{\mathbb{P}_P\left(\widehat{a}_T = b\right)}_{\text{probability of misidentification}}, \quad (1)$$

where $\Delta^a(P)$ is a gap $\max_{b \in [K]} \mu^b(P) - \mu^a(P) = \mu^{a^*(P)}(P) - \mu^a(P)$, and the expectation is taken over the randomness of $\widehat{a}_T \in [K]$ and $\widetilde{\mathcal{P}} \subseteq \mathcal{P}$ is a specific class of bandit models.

With this objective, we first derive the worst-case lower bounds for the expected simple regret in Section 3. Then, we propose an algorithm in Section 4 and show the minimax optimality in Section 5.

**Notation.** Let $\mu^a(P)(x) := \mathbb{E}_P[Y^a | X = x]$ be the conditional expected outcome given $x \in \mathcal{X}$ for $a \in [K]$. Similarly, we define $\nu^a(P)(x) := \mathbb{E}_P[(Y^a)^2 | X = x]$ and $\nu^a(P) := \mathbb{E}_P[\nu^a(P)(X)]$. Let $(\sigma^a(P)(x))^2$ and $(\sigma^a(P))^2$ be the conditional and unconditional variances of $Y_t^a$; that is, $(\sigma^a(P)(x))^2 = \nu^a(P)(x) - (\mu^a(P)(x))^2$ and $(\sigma^a(P))^2 = \nu^a(P) - (\mu^a(P))^2$. For all $P \in \mathcal{P}^*$ and $a \in [K]$, let $P^a$ be the joint distributions of $(Y^a, X)$.

## 3 LOWER BOUNDS FOR THE WORST-CASE EXPECTED SIMPLE REGRET

This section presents lower bounds for the worst-case expected simple regret. In Eq. (1), for each fixed $P \in \widetilde{\mathcal{P}}$, the *probability of misidentification* $\mathbb{P}_P\left(\widehat{a}_T \notin \arg\max_{a \in [K]} \mu^a(P)\right)$ converges to zero with an exponential rate, while $\Delta^{\widehat{a}_T}(P)$ is the constant. Therefore, we disregard $\Delta^{\widehat{a}_T}(P)$, and the convergence rate of $\mathbb{P}_P\left(\widehat{a}_T \notin \arg\max_{a \in [K]} \mu^a(P)\right)$ dominates the expected simple regret. In this case, to evaluate the convergence rate of $\mathbb{P}_P\left(\widehat{a}_T \notin \arg\max_{a \in [K]} \mu^a(P)\right)$, we utilize large deviation upper bounds. In contrast, if we examine the worst case among $\widetilde{\mathcal{P}}$, which includes a bandit model such that the gaps between the expected outcomes converge to zero with a certain order of the sample size $T$, a bandit model $P$ whose gaps converge to zero at a rate of $O(1/\sqrt{T})$ dominates the expected simple regret. For the gap $\Delta^b(P)$, the worst-case simple regret is approximately given by $\sup_{P \in \widetilde{\mathcal{P}}} \mathbb{E}_P[r_T(P)(\pi)] \approx \sup_{P \in \widetilde{\mathcal{P}}} \sum_{b \in [K]} \Delta^b(P) \exp\left(-T\left(\Delta^b(P)\right)^2 / C^b\right)$, where $(C^b)_{b \in [b]}$ are constants. Then, the maximum is obtained when $\Delta^a(P) = \sqrt{\frac{C^b}{T}}$ for a constant $C^b > 0$, which balances the regret caused by the gap $\Delta^b(P)$ and probability of misidentification $\mathbb{P}_P\left(\widehat{a}_T = b\right)$.

### 3.1 RECAP: LOWER BOUNDS FOR BANDIT MODELS WITH BOUNDED SUPPORTS

Bubeck et al. (2011) shows a (non-asymptotic) lower bound for bandit models with a bounded support, where a strategy with the uniform sampling rule is optimal. Let us denote bandit models with bounded outcomes by $P^{[0,1]}$, where each potential outcome $Y_t^a$ is in $[0,1]$. Then, the authors show that for all $T \geq K \geq 2$, any strategy $\pi \in \Pi$ satisfies $\sup_{P \in P^{[0,1]}} \mathbb{E}_P[r_T(P)(\pi)] \geq \frac{1}{20}\sqrt{\frac{K}{T}}$. For this worst-case lower bound, the authors show that a strategy with the uniform sampling rule and empirical best arm (EBA) recommendation rule is optimal, where we draw $A_t = a$ with probability $1/K$ for all $a \in [K]$ and $t \in [T]$ and recommend an arm with the highest sample average of the observed outcomes, which is referred to as the uniform-EBA strategy. Under the uniform-EBA strategy $\pi^{\text{Uniform-EBM}}$, the expected simple regret is bounded as $\sup_{P \in \mathcal{P}^{[0,1]}} \mathbb{E}_P\left[r_T\left(\pi^{\text{Uniform-EBM}}\right)(P)\right] \leq 2\sqrt{\frac{K \log K}{T+K}}$. Thus, the upper bound matches the lower bound if we ignore the $\log K$ and constant terms.

Although the uniform-EBA strategy is nearly optimal, a question remains whether, by using more distributional information, we can show tighter lower bounds or develop more efficient strategies. As the lower bound in Bubeck et al. (2011) is referred to as a distribution-free lower bound, the lower

bound does not utilize distributional information, such as variance. In this study, although we still consider the worst-case expected simple regret, we develop lower and upper bounds depending on distributional information.

Specifically, we characterize the bounds by the variances of outcomes. Recall that the worst-case expected simple regret is dominated by an instance of a bandit model with $\Delta^a(P) = O(1/\sqrt{T})$. Here, also recall that tight lower bounds are derived from the KL divergence (Lai & Robbins, 1985; Kaufmann et al., 2016), whose second-order Taylor expansion with respect to the gap $\Delta^a(P)$ corresponds to the variance (inverse of the Fisher information). Therefore, the tight lower bounds employing distributional information in the worst-case expected simple regret should be characterized by the variances (the second-order Taylor expansion of the KL divergence). In the following sections, we consider worst-case lower bounds characterized by the variances of bandit models.

## 3.2 Local Location-Shift Bandit Models

In this section, we derive asymptotic lower bounds for the worst-case expected simple regret. To derive lower bounds, we often utilize an alternative hypothesis. We consider a bandit model whose expected outcomes are the same among the $K$ arms. We refer to it as the null bandit model.

**Definition 3.1** (Null bandit models). A bandit model $P \in \mathcal{P}$ is called a null bandit model if the expected outcomes are equal: $\mu^1(P) = \mu^2(P) = \cdots = \mu^K(P)$.

Then, we consider a class of bandit models with a unique fixed variance for null bandit models, called local location-shift bandit models. Furthermore, we assume that potential outcomes are conditionally sub-Gaussian, and their parameters are bounded. We define our bandit models as follows.

**Definition 3.2** (Local location-shift bandit models). A class of bandit models $\mathcal{P}^*$ are called *local location-shift bandit models* if it contains all bandit models that satisfy the following conditions:

**(i) Absolute continuity.** For all $P, Q \in \mathcal{P}^*$ and $a \in [K]$, $P^a$ and $Q^a$ are mutually absolutely continuous and have density functions with respect to some reference measure.

**(ii) Invariant contextual information.** For all $P \in \mathcal{P}^*$, the distributions of contextual information $X$ are the same. Let $\mathbb{E}^X$ be an expectation operator over $X$ and $\zeta(x)$ be the density of $X$.

**(iii) Unique conditional variance.** All null bandit models $P^\sharp \in \mathcal{P}^*$ ($\mu^1(P^\sharp) = \mu^2(P^\sharp) = \cdots = \mu^K(P^\sharp)$) have the unique conditional variance; that is, for all $P^\sharp \in \mathcal{P}^*$, there exists an universal constant $\sigma^a(x) > C$ such that for all $P^\sharp \in \mathcal{P}^*$, $\left(\sigma^a(P^\sharp)(x)\right)^2 = (\sigma^a(x))^2$ holds.

**(iv) Bounded moments.** There exist known universal constants $0 < \underline{C} < \overline{C} < \infty$ such that for all $P \in \mathcal{P}^*$, $a \in [K]$, and $x \in \mathcal{X}$, $|\mu^a(P)(x)| < \overline{C}$ and $\underline{C} < \sigma^a(P)(x) < \overline{C}$ hold.

**(v) Parallel shift.** There exists an universal constant $C > 0$, independent from $P$, such that for all $P \in \mathcal{P}^*$, $x \in \mathcal{X}$, and $a, b \in [K]$, $\left|\left(\mu^a(P)(x) - \mu^b(P)(x)\right)\right| \leq C \left|\mu^a(P) - \mu^b(P)\right|$.

**(vi) Continuity.** For all $a \in [K]$, $x \in \mathcal{X}$, and a null bandit model $P^\sharp \in \mathcal{P}^*$, $\lim_{n \to \infty} (\sigma^a(P_n)(x))^2 = (\sigma^a(x))^2$ holds for a sequence of bandit models $\{P_n\}$ such that $\lim_{n \to \infty} \mu^a(P_n)(x) = \mu^a(P^\sharp)(x)$.

Our lower bounds are characterized by $\sigma^a(x)$, a conditional variance of null bandit models.

Local location-shift models are a common assumption in statistical inference (Lehmann & Casella, 1998). A key example are Gaussian and Bernoulli distributions. Under Gaussian distributions with fixed variances, for all $P$, the variances are fixed, and only mean parameters shift. Such models are generally called location-shift models. Additionally, we can consider Bernoulli distributions if $\mathcal{P}^*$ includes one instance of $\mu^1(P) = \cdots = \mu^K(P)$ to specify one fixed variance $\sigma^a(x)$. Furthermore, our bandit models are nonparametric within the class and include a wider range of bandit models, similar to the approach of Hahn et al. (2011) and Barrier et al. (2022).

In (iv), we assume that there are *known* constants $0 < \underline{C} < \overline{C} < \infty$. They are introduced for technical purposes in theoretical analysis. In application, we set a sufficiently large value for $\overline{C}$ and a sufficiently small value for $\underline{C}$. Note that the lower boundedness of $\sigma^a(P)(x)$ is assumed for sampling each arm with a positive probability and the constant $\underline{C}$ plays a role similar to the forced sampling (Garivier & Kaufmann, 2016) and the parameter $\beta$ in top-two sampling (Russo, 2020) for fixed-confidence BAI. Thus, in BAI, we usually assume the existence of such constants.

When contextual information is unavailable, condition (v) can be omitted. Although condition (v) may seem restrictive, its inclusion is not essential for achieving efficiency gains through the utilization of contextual information; that is, the upper bound can be smaller even when this condition is not met. However, it is required in order to derive a matching upper bound for the following lower bounds.

## 3.3 Asymptotic Lower Bounds under Local Location-Shift Bandit Models

We consider a restricted class of strategies such that under null bandit models, any strategy in this class recommends one of the arms with an equal probability $(1/K)$.

**Definition 3.3** (Null consistent strategy). Under any null bandit models $P \in \mathcal{P}$ $(\mu^1(P) = \mu^2(P) = \cdots = \mu^K(P))$, any null consistent strategy satisfies that for any $a, b \in [K]$, $\left| \mathbb{P}_P (\widehat{a}_T = a) - \mathbb{P}_P (\widehat{a}_T = b) \right| \to 0$ holds as $T \to \infty$. This implies that $\left| \mathbb{P}_P (\widehat{a}_T = a) - 1/K \right| = o(1)$.

This restriction is introduced to characterize the lower bounds by using the variances of the worst-case where the gap $\mu^a(P) - \mu^b(P)$ between two arms $a, b \in [K]$ is small.

For each $x \in \mathcal{X}$, we refer to $\frac{1}{T} \sum_{t=1}^{T} \mathbb{1}[A_t = a]$ as an allocation ratio under a strategy. Let us also define an average allocation ratio under $P \in \mathcal{P}$ and a strategy as $\kappa_{T,P}(a|x) = \mathbb{E}_P \left[ \frac{1}{T} \sum_{t=1}^{T} \mathbb{1}[A_t = a] | X_t = x \right]$, which plays an important role in the proof of our lower bound. Let $\mathcal{W}$ be a set of all measurable functions $\kappa_{T,P} : [K] \times \mathcal{X} \to (0, 1)$ such that $\sum_{a \in [K]} \kappa_{T,P}(a|x) = 1$ for each $x \in \mathcal{X}$. Then, we show the following lower bound. The proof is shown in Appendix D.

**Theorem 3.4.** *For $K \geq 2$, any null consistent strategy satisfies*

$$\sup_{P \in \mathcal{P}^*} \sqrt{T} \mathbb{E}_P[r_T(P)(\pi)] \geq \frac{1}{12} \inf_{w \in \mathcal{W}} \max_{a \in [K]} \sqrt{\mathbb{E}^X \left[ (\sigma^a(X))^2 / w(a|X) \right]} + o(1) \quad \text{as } T \to \infty.$$

We refer to $w^* = \arg\min_{w \in \mathcal{W}} \max_{a \in [K]} \sqrt{\mathbb{E}^X \left[ \frac{(\sigma^a(X))^2}{w(a|X)} \right]}$ as the *target allocation ratio*, which corresponds to an allocation ratio that an optimal strategy aims to achieve and is used to define a sampling rule in our proposed strategy. When there is no contextual information, we can obtain an analytical solution for $w^*$.

*Remark* 3.5 (Asymptotic lower bounds under allocation constraints). When we restrict the target allocation ratio, we can restrict the class $\mathcal{W}$. For example, when we need to draw specific arms at a predetermined ratio, we consider a class $\mathcal{W}^\dagger$ such that for all $w \in \mathcal{W}^\dagger$, $\sum_{a \in [K]} w(a|x) = 1$ and $w(b|x) = C$ for all $x \in \mathcal{X}$, some $b \in [K]$, and a constant $C \in (0, 1)$.

## 3.4 Lower Bounds without Contextual Information

Our result generalizes BAI without contextual information, where $\mathcal{X}$ is a singleton. For simplicity, we denote $w(a|x)$ by $w(a)$. When there is no contextual information, we can obtain the following lower bound with an analytical solution of the target allocation ratio $w^* \in \mathcal{W}$.[2]

**Corollary 3.6.** *Any null consistent strategy satisfies* $\sup_{P \in \mathcal{P}^*} \sqrt{T} \mathbb{E}_P[r_T(P)(\pi)] \geq \frac{1}{12} \sqrt{\sum_{a \in [K]} (\sigma^a)^2} + o(1)$ *as* $T \to \infty$, *where the target allocation ratio is* $w^*(a) = \frac{(\sigma^a)^2}{\sum_{b \in [K]} (\sigma^b)^2}$.

*Remark* 3.7 (Efficiency gain by using the contextual information). When contextual information is available, if we substitute suboptimal allocation ratio $w(a|x) = \frac{(\sigma^a(x))^2}{\sum_{b \in [K]} (\sigma^b(x))^2}$ into $\max_{a \in [K]} \sqrt{\mathbb{E}^X \left[ \frac{(\sigma^a(X))^2}{w(a|X)} \right]}$, a lower bound is $\frac{1}{12} \sqrt{\sum_{a \in [K]} \mathbb{E}^X \left[ (\sigma^a(X))^2 \right]} + o(1)$, which can be tightened by optimizing with respect to $w$. It is worth noting that by using the law of total variance, $(\sigma^a)^2 \geq \mathbb{E}^X \left[ (\sigma^a(X))^2 \right]$. Therefore, by utilizing contextual information, we can tighten the lower bounds as $\sqrt{\sum_{a \in [K]} (\sigma^a)^2} \geq \sqrt{\sum_{a \in [K]} \mathbb{E}^X \left[ (\sigma^a(X))^2 \right]} \geq \min_{w \in \mathcal{W}} \max_{a \in [K]} \sqrt{\mathbb{E}^X \left[ \frac{(\sigma^a(X))^2}{w(a|X)} \right]}$. This improvement implies efficiency gain by using contextual information $X_t$.

---

[2] Because $\mathcal{X}$ is a singleton, we consider $\mathcal{W} := \{(w(a))_{a \in [K]} | w(a) \in [0, 1], \sum_{a \in [K]} w(a) = 1\}$.

### 3.5 Refined Lower Bounds for Two-Armed Bandits

For two-armed bandits ($K = 2$), we can refine the lower bound as follows. In this case, we can also obtain an analytical solution of the target allocation ratio even when there is contextual information.

**Theorem 3.8.** *When $K = 2$, any null consistent strategy satisfies*

$$\sup_{P \in \mathcal{P}^*} \sqrt{T} \mathbb{E}_P[r_T(P)(\pi)] \geq \frac{1}{12} \sqrt{\mathbb{E}^X \left[ (\sigma^1(X) + \sigma^2(X))^2 \right]} + o(1)$$

*as $T \to \infty$, where the target allocation ratio is $w^*(a|x) = \sigma^a(x)/(\sigma^1(x) + \sigma^2(x))$ for all $x \in \mathcal{X}$.*

This lower bound is derived as a solution of $\sup_{P \in \mathcal{P}^*} \sqrt{T} \mathbb{E}_P[r_T(P)(\pi)] \geq \frac{1}{12} \inf_{w \in \mathcal{W}}$ $\sqrt{\mathbb{E}^X \left[ \frac{(\sigma^1(X))^2}{w(1|X_t)} + \frac{(\sigma^2(X_t))^2}{w(2|X_t)} \right]} + o(1)$. Here, note that $\inf_{w \in \mathcal{W}} \mathbb{E}^X \left[ \frac{(\sigma^1(X))^2}{w(1|X_t)} + \frac{(\sigma^2(X_t))^2}{w(2|X_t)} \right] \geq$ $\inf_{w \in \mathcal{W}} \max_{a \in [2]} \sqrt{\mathbb{E}^X \left[ \frac{(\sigma^a(X))^2}{w(a|X)} \right]}$. Therefore, this lower bound is tighter than that in Theorem 3.4.

This target allocation ratio is the same as that in Kaufmann et al. (2016) for the probability of misidentification minimization. The proofs are shown in Appendix F.

Note that for $K \geq 3$, we have an analytical solution of the target allocation ratio only when there is no contextual information, and the target allocation ratio is the ratio of the variances. In contrast, for $K = 2$, we can obtain analytical solutions of the target sample allocation ratio even when there is contextual information, and it is the ratio of the (conditional) standard deviation.

## 4 The AS-AIPW Strategy

This section introduces our strategy, which comprises the following sampling and recommendation rules. In each round, we adaptively estimate the variances $(\sigma^a(P)(x))^2$ and draw arms following a probability using the estimates of the variances. At the end, we recommend the best arm using the AIPW estimator. We call this strategy the *AS-AIPW strategy*. We show a pseudo-code in Algorithm 1.

### 4.1 Target allocation ratio

First, we define target allocation ratios, which are used to determine our sampling rule. From the results in Section 3, the target allocation ratios are given as follows:

$$w^*(1|x) = \sigma^a(x)/\sigma^1(x) + \sigma^2(x) \quad \text{and} \quad w^*(2|x) = 1 - w^*(1|x) \qquad \text{for } x \in \mathcal{X} \text{ if } K \geq 3,$$

$$w^* = \arg\min_{w \in \mathcal{W}} \max_{a \in [K]} \sqrt{\mathbb{E}^X \left[ (\sigma^a(X))^2/w(a|X) \right]} \qquad \text{if } K \geq 3.$$

As discussed in Section 3, when $K \geq 3$, there is no closed form for $w^*$ except for a case without contextual information, where the target allocation ratio is $w^*(a) = \frac{(\sigma^a)^2}{\sum_{b \in [K]} (\sigma^b)^2}$ for all $a \in [K]$.

The target allocation ratios depend on the variances. Therefore, when the variances are unknown, this target allocation ratio is also unknown. In our strategy, we estimate them during an adaptive experiment and use the estimator to obtain the probability of drawing arms.

### 4.2 Sampling Rule with Adaptive Variance Estimation

Let $\widehat{w}_t$ be an estimator of $w^*$ at round $t \in [T]$ such that for all $a \in [K]$ and $x \in \mathcal{X}$, $\widehat{w}_t(a|x) > 0$ and $\sum_{a \in [K]} \widehat{w}_t(a|x) = 1$ hold. In each round $t$, we obtain $\gamma_t$ from $\mathrm{Uniform}[0, 1]$, the uniform distribution on $[0, 1]$, and draw arm $A_t = 1$ if $\gamma_t \leq \widehat{w}_t(1|X_t)$ and $A_t = a$ for $a \geq 2$ if $\gamma_t \in \left( \sum_{b=1}^{a-1} \widehat{w}_t(b|X_t), \sum_{b=1}^{a} \widehat{w}_t(b|X_t) \right]$; that is, we draw arm $a$ with probability $\widehat{w}_t(a|X_t)$.

In round $t \leq K$, we draw arm $A_t = t$ as an initialization and set $\widehat{w}_t(a|X_t) = 1/K$ for all $a \in [K]$. In round $t > K$, we estimate the target allocation ratio $w^*$ using the past observations $\mathcal{F}_{t-1}$.

In the sampling rule and the following recommendation rule, we use $\mathcal{F}_{t-1}$-measurable estimators of $\mu^a(P)(X_t)$, $\nu^a(P)(X_t)$, and $w^*(a|X_t)$ at each round $t \in [T]$. We denote $\mathcal{F}_{t-1}$-measurable estimators of $\mu^a(P)(X_t)$, $\nu^a(P)(X_t)$ and $(\sigma^a(X_t))^2$ as $\widehat{\mu}_t^a(X_t)$, $\widehat{\nu}_t^a(X_t)$, and $(\widehat{\sigma}_t^a(X_t))^2$. For $t \leq K$, we set $\widehat{\mu}_t^a(X_t) = \widehat{\nu}_t^a = (\widehat{\sigma}_t^a(x))^2 = 0$. For $t > K$, we use estimators constructed as follows. For $t > K$, we estimate $\mu^a(P)(x)$ and $\nu^a(P)(x)$ using only past samples $\mathcal{F}_{t-1}$. The requirement for the estimators is convergence to the true parameter almost surely. We use a bounded estimator for $\widehat{\mu}_t^a$ such that $|\widehat{\mu}_t^a| < \overline{C}$. Let $(\widehat{\sigma}_t^{\dagger a}(x))^2 = \widehat{\nu}_t^a(x) - (\widehat{\mu}_t^a(x))^2$ for all $a \in [K]$ and $x \in \mathcal{X}$. Then, we estimate the variance $(\sigma^a(x))^2$ for all $a \in [K]$ and $x \in \mathcal{X}$ in a round $t$ as $(\widehat{\sigma}_t^a(x))^2 = \max\{\min\{((\widehat{\sigma}_t^{\dagger a}(x))^2, \overline{C}\}, 1/\underline{C}\}$ and define $\widehat{w}_t$ by replacing the variances in $w^*$ with corresponding estimators; that is, when $K = 2$, for each $a \in \{1, 2\}$, $\widehat{w}(a|X_t) = \frac{\widehat{\sigma}_t^1(X_t)}{\widehat{\sigma}_t^1(X_t) + \widehat{\sigma}_t^2(X_t)}$; when $K \geq 3$, for each $a \in [K]$, $\widehat{w}(a|X_t) = \arg\min_{w \in \mathcal{W}} \max_{a \in [K]} \sqrt{\sum_{a \in [K]} \frac{(\widehat{\sigma}_t^a(X_t))^2}{w(a|X_t)}}$. If there is no contextual information, we obtain a closed-form $\widehat{w}(a|X_t) = \frac{(\widehat{\sigma}_t^a(X_t))^2}{\sum_{b \in [K]}(\widehat{\sigma}_t^b(X_t))^2}$ when $K \geq 3$.

When there is no contextual information ($\mathcal{X}$ is a singleton), we estimate $\mu^a(P)(x) = \mu^a(P)$ and $w^*$ by using the sample averages. When there is contextual information, we can employ nonparametric estimators, such as the nearest neighbor regression estimator and kernel regression estimator, to estimate $\mu^a(P)$ and $w^*$. These estimators have been proven to converge to the true function almost surely under a bounded sampling probability $\widehat{w}_t$ by Yang & Zhu (2002) and Qian & Yang (2016). Provided that these conditions are satisfied, any estimator can be used. It should be noted that we do not assume specific convergence rates for estimators $\mu^a(P)(x)$ and $w^*$ as the asymptotic optimality of the AIPW estimator can be shown without them (van der Laan, 2008; Kato et al., 2020; 2021).

### 4.3 RECOMMENDATION RULE USING THE AIPW ESTIMATOR

Finally, we present our recommendation rule. After the final round $T$, for each $a \in [K]$, we estimate $\mu^a(P)$ for each $a \in [K]$ and recommend the maximum as an estimate of the best arm. To estimate $\mu^a(P)$, the AIPW estimator is defined as

$$\widehat{\mu}_T^{\mathrm{AIPW},a} = \frac{1}{T} \sum_{t=1}^{T} \varphi_t^a\Big(Y_t, A_t, X_t\Big), \quad \varphi_t^a(Y_t, A_t, X_t) = \frac{\mathbb{1}[A_t = a]\big(Y_t^a - \widehat{\mu}_t^a(X_t)\big)}{\widehat{w}_t(a|X_t)} + \widehat{\mu}_t^a(X_t).$$

Then, we recommend $\widehat{a}_T^{\mathrm{AIPW}} \in [K]$, defined as $\widehat{a}_T^{\mathrm{AIPW}} = \arg\max_{a \in [K]} \widehat{\mu}_T^{\mathrm{AIPW},a}$.

The AIPW estimator debiases the sample selection bias resulting from arm draws depending on past observations and contextual information. Additionally, the AIPW estimator possesses the following properties: (i) its components $\varphi_t^a(Y_t, A_t, X_t)_{t=1}^T$ are a martingale difference sequence, allowing us to employ the martingale limit theorems in the derivation of the upper bound; (ii) it has the minimal asymptotic variance among the possible estimators. For example, other estimators with a martingale property, such as the inverse probability weighting estimator, may be employed, yet their asymptotic variance would be larger than that of the AIPW estimator. The variance-reduction effect has also been employed in the studies for adversarial bandits by Ito et al. (2022), independently of this study. The $t$-th element of the sum in the AIPW estimator utilizes nuisance parameters ($\mu^a(P)(x)$ and $w^*$) estimated from past observations up to round $t-1$ for constructing a martingale difference sequence (van der Laan, 2008; Hadad et al., 2021; Kato et al., 2020; 2021; Ito et al., 2022).

The AS-AIPW strategy constitutes a generalization of the Neyman allocation (Neyman, 1934), which has been utilized for the efficient estimation of the ATE with two treatment arms (van der Laan, 2008; Hahn et al., 2011; Tabord-Meehan, 2022; Kato et al., 2020)[3] and two-armed fixed-budget BAI without contextual information (Glynn & Juneja, 2004; Kaufmann et al., 2016).

## 5 ASYMPTOTIC MINIMAX OPTIMALITY OF THE AS-AIPW STRATEGY

In this section, we derive upper bounds for the worst-case expected simple regret under local location-shift models with our proposed AS-AIPW strategy.

---

[3]The AS-AIPW strategy is also similar to those proposed for efficient estimation of the ATE with two armed bandits (van der Laan, 2008; Kato et al., 2020).

---

**Algorithm 1** AS-AIPW strategy.

---

**Parameter:** Positive constants $C_\mu$ and $\overline{C}$.
**Initialization:**
**for** $t = 1$ to $K$ **do**
  Draw $A_t = t$. For each $a \in [K]$, set $\widehat{w}_t(a|x) = 1/K$ for all $a \in [K]$.
**end for**
**for** $t = K + 1$ to $T$ **do**
  Observe $X_t$.
  Construct $\widehat{w}_t$ by using the estimators of the variances.
  Draw $\gamma_t$ from the uniform distribution on $[0, 1]$.
  $A_t = 1$ if $\gamma_t \leq \widehat{w}_t(1|X_t)$ and $A_t = a$ for $a \geq 2$ if $\gamma_t \in \left( \sum_{b=1}^{a-1} \widehat{w}_t(b|X_t), \sum_{b=1}^{a} \widehat{w}_t(b|X_t) \right]$.
**end for**
Construct the AIPW estimator $\widehat{\mu}_T^{\mathrm{AIPW},a}$ for each $a \in [K]$ and recommend $\widehat{a}_T^{\mathrm{AIPW}}$.

---

First, we make the following assumption, which holds for a wide class of estimators (Yang & Zhu, 2002; Qian & Yang, 2016).

**Assumption 5.1.** For all $P \in \mathcal{P}^*$, $a \in [K]$, $x \in \mathcal{X}$, the followings hold:

$$\left| \widehat{\mu}_t^a(x) - \mu^a(P)(x) \right| \xrightarrow{\text{a.s.}} 0, \text{ and } \left| \widehat{\nu}_t^a(x) - \nu^a(P)(x) \right| \xrightarrow{\text{a.s.}} 0 \quad \text{as } t \to \infty.$$

Then, we show the worst-case upper bound for the expected simple regret as follows. The proof is shown in Appendix G, where we employ the martingale CLT and Chernoff bound as in Appendix B.

**Theorem 5.2** (Worst-case upper bound). *Under Assumption 5.1 and the AS-AIPW strategy* $\pi^{\mathrm{AS\text{-}AIPW}}$,

$$\sup_{P \in \mathcal{P}^*} \sqrt{T} \mathbb{E}_P \left[ r_T(P) \left( \pi^{\mathrm{AS\text{-}AIPW}} \right) \right] \leq \max_{a,b \in [K]:\ a \neq b} \sqrt{\log(K) \mathbb{E}^X \left[ \frac{(\sigma^a(X))^2}{w^*(a|X)} + \frac{(\sigma^b(X))^2}{w^*(b|X)} \right]} + o(1)$$

*holds as* $T \to \infty$.

By substituting specific values into $w^*$ in the upper bound, we can confirm that the leading factors of the upper bounds match our derived lower bounds in Section 3; that is,

$$\sup_{P \in \mathcal{P}^*} \sqrt{T} \mathbb{E}_P \left[ r_T(P) \left( \pi^{\mathrm{AS\text{-}AIPW}} \right) \right]$$

$$\leq \begin{cases} \frac{1}{2} \sqrt{\mathbb{E}^X \left[ (\sigma^1(X) + \sigma^2(X))^2 \right]} + o(1) & \text{when } K = 2 \\ 2 \min_{w \in \mathcal{W}} \max_{a \in [K]} \sqrt{\log(K) \mathbb{E}^X \left[ \frac{(\sigma^a(X))^2}{w^*(a|X)} \right]} + o(1) & \text{when } K \geq 3 \end{cases},$$

as $T \to \infty$. Thus, we proved the asymptotic minimax optimality of our proposed strategy.

There is a $\log(K)$ factor in the upper bound, which appears in existing studies, such as Bubeck et al. (2011) and Komiyama et al. (2023). As existing studies discussed, we consider that further restrictions are required for a class of strategies or bandit models to fulfill the discrepancy.

We also provide non-asymptotic upper bounds in Appendix H. Note that the non-asymptotic upper bounds are dominated by the convergence rate of the second moment of $\varphi_t^a$ to a round-independent constant (Hall et al., 1980). The convergence rate is significantly influenced by the estimation error of the variances, which are used to estimate the target allocation ratio (sampling rule).

## 6   DISCUSSION AND RELATED WORK

This section briefly introduces related work. A more detailed survey is presented in Appendix A. Bubeck et al. (2011) shows that the uniform-EBA strategy is minimax optimal for the worst-case expected simple regret under bandit models with bounded supports. While Bubeck et al. (2011) provides an optimal strategy whose order of the expected simple regret aligns with their proposed

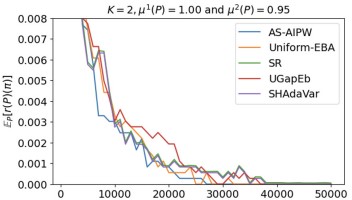 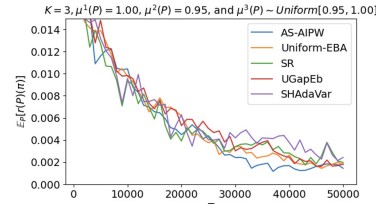

Figure 1: Experimental results. The $y$-axis and $x$-axis denote the expected simple regret $\mathbb{E}_P[r_T(P)(\pi)]$ under each strategy and $T$, respectively.

worst-case lower bound, their lower bounds only depend on the assumption of the bounded support and do not use the other distributional information (distribution-free analysis). In contrast, there is a longstanding open issue on the existence of asymptotic optimal strategies whose upper bound aligns with distribution-dependent lower bounds suggested by Kaufmann et al. (2016) (distribution-dependent analysis). For details, see related work, such as Kaufmann (2020), Ariu et al. (2021), and Degenne (2023). Our optimality criterion is an intermediate between the distribution-free (Bubeck et al., 2011) and distribution-dependent (Kaufmann et al., 2016) because our lower and upper bounds are derived under the worst-case but depend on the variances (distributional information).

Variance-dependent BAI strategies have been discussed by existing work, mainly for minimizing the probability of misidentification, such as Chen et al. (2000), Glynn & Juneja (2004), and Kaufmann et al. (2016). Glynn & Juneja (2004) discusses optimal strategy for Gaussian bandits under the large deviation principles. Kaufmann et al. (2016) develops a lower bound for two-armed Gaussian bandits and finds that when the variances are known, drawing each arm with a ratio of the standard deviation is asymptotically optimal. Independently of us, Lalitha et al. (2023) also proposes variance-dependent strategies by extending the sequential halving strategy (Karnin et al., 2013). However, the optimality is unclear since they do not provide lower bounds. In fact, their strategy's probability of misidentification is larger than that of Kaufmann et al. (2016) when $K = 2$.

## 7  EXPERIMENTS

In this section, we compare our AS-AIPW strategy with the Uniform-EBA (Uniform, Bubeck et al., 2011), Successive Rejection (SR, Audibert et al., 2010), UGapEb (Gabillon et al., 2012), and SHAdaVar (Lalitha et al., 2023). We investigate two setups with $K = 2, 3$ without contextual information for these strategies. The best arm is arm 1 and $\mu^1(P) = 1.00$. The expected outcomes of suboptimal arms are $\mu^2(P) = 0.95$ and $\mu^3(P) \sim \text{Uniform}[0.95, 1.00]$. Variances are generated from Uniform$[1.00, 5.00]$. We continue the experiments until $T = 50,000$ and conduct 150 independent trials. At each round $t \in \{1,000, 2,000, 3,000, \cdots, 49,000, 50,000\}$, we compute the simple regrets and plot the empirical average of the simple regrets in Figure 1. For the SR and SHAdaVar, we restart the experiments after reaching each $T$. Additional results with other settings, including contextual information, are presented in Appendix I.

From Figure 1 and Appendix I, we can confirm the soundness of our proposed strategy. When contextual information exists, our methods show better performances than the others. Although our strategies show preferable performances in many settings, other strategies also perform well. We conjecture that our strategies exhibit superiority against other methods when $K$ is small (mismatching term in the upper bound), the gap between the best and suboptimal arms is small, and the variances significantly vary across arms. As the superiority depends on the situation, we recommend a practitioner use several strategies in a hybrid way.

## 8  CONCLUSION

We conducted an asymptotic worst-case analysis of the simple regret in fixed-budget BAI with contextual information. Initially, we obtained lower bounds for local location-shift bandit models, where the variances of potential outcomes characterize the asymptotic lower bounds as a second-order approximation of the KL divergence. Based on these lower bounds, we derived target allocation ratios, which were used to define a sampling rule in the AS-AIPW strategy. Finally, we demonstrated that the AS-AIPW strategy achieves minimax optimality for the expected simple regret.

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

# A  RELATED WORK

In this section, we introduce related work, in addition to the work in Section 6.

## A.1  LITERATURE OF BAI

The MAB problem is an abstraction of the sequential decision-making process (Thompson, 1933; Robbins, 1952; Lai & Robbins, 1985). BAI is a paradigm of this problem (Even-Dar et al., 2006; Audibert et al., 2010; Bubeck et al., 2011), with its variants dating back to the 1950s in the context of sequential testing, ranking, and selection problems (Bechhofer et al., 1968). Additionally, ordinal optimization has been extensively studied in the field of operations research, with a modern formulation established in the early 2000s (Chen et al., 2000; Glynn & Juneja, 2004). Most of these studies have focused on determining optimal strategies under the assumption of known target allocation ratios. Within the machine learning community, the problem has been reframed as the BAI problem, with a particular emphasis on performance evaluation under unknown target allocation ratios. (Bubeck et al., 2009; 2011; Audibert et al., 2010).

In fixed-budget BAI, Bubeck et al. (2009) demonstrates minimax optimal strategies for the expected simple regret, and Audibert et al. (2010) proposes the UCB-E and Successive Rejects (SR) strategies. Kock et al. (2023) generalizes the results of Bubeck et al. (2011) to cases where parameters of interest are functionals of the distribution and find that target allocation ratios are not uniform, in contrast to the results of Bubeck et al. (2011).

Kaufmann et al. (2016) contributes to this field by deriving distribution-dependent lower bounds for BAI with fixed confidence and a fixed budget, using the change-of-measure arguments as well as (Lai & Robbins, 1985). In the setting of fixed-confidence BAI, Garivier & Kaufmann (2016) proposes an optimal strategy for the lower bounds derived by Kaufmann et al. (2016); however, in the fixed-budget setting, there is currently a lack of strategies whose upper bound matches the lower bound established by Kaufmann et al. (2016). Kaufmann (2020) points out this issue, and Ariu et al. (2021) finds a bandit model whose lower bound for the probability of misidentification is larger than those by Kaufmann (2020). Qin (2022) summarizes this problem. Degenne (2023) discusses the existence of optimal strategies.

Carpentier & Locatelli (2016) examines the lower bound for the probability of misidentification under the minimax framework and shows the optimality of the method proposed by Audibert et al. (2010) in terms of leading factors in the exponent. The lower bound of Carpentier & Locatelli (2016) is based on a minimax evaluation of the probability of misidentification under a large gap. Yang & Tan (2022) proposes minimax optimal linear BAI with a fixed budget by extending the result of Carpentier & Locatelli (2016).

In addition to minimax evaluation, Komiyama et al. (2023) develops an optimal strategy whose upper bound for a simple Bayesian regret lower bound matches their derived lower bound. Atsidakou et al. (2023) proposes a Bayes optimal strategy for minimizing the probability of misidentification, which shows a surprising result that $1/\sqrt{T}$-factor dominates the evaluation.

In Russo (2020), Qin et al. (2017), and Shang et al. (2020), respectively, the authors propose Bayesian BAI strategies that are optimal in terms of posterior convergence rate. However, it has been shown by Kasy & Sautmann (2021) and Ariu et al. (2021) that such optimality does not extend to asymptotic optimality for the probability of misidentification in fixed-budget BAI.

Adusumilli (2022; 2021) present an alternative minimax evaluation of bandit strategies for both regret minimization and BAI, which is based on a formulation utilizing a diffusion process proposed by Wager & Xu (2023). Furthermore, Armstrong (2022) extends the results of Hirano & Porter (2009) to a setting of adaptive experiments. The results of Adusumilli (2022; 2021) and Armstrong (2022) employ arguments on local asymptotic normality (Le Cam, 1960; 1972; 1986; van der Vaart, 1991; 1998), where the class of alternative hypotheses comprises of "local models," in which parameters of interest converge to true parameters at a rate of $1/\sqrt{T}$.

There are several studies in variance-dependent BAI. Chen et al. (2000), Glynn & Juneja (2004), and Kaufmann et al. (2016) propose variance-dependent strategies for fixed-budget BAI under their optimal criterion. However, when $K \geq 3$, the lower bounds and asymptotic optimality are unknown.

Therefore, it is unclear whether their use of variances leads to optimal strategies. Sauro (2020), Lu et al. (2021), and Lalitha et al. (2023) also provide variance-dependent BAI strategies, but their optimality is unclear.

Only when variances are known and the number of arms is equal to two, is it known that the target allocation ratio is the ratio of the standard deviation. However, Lalitha et al. (2023) proposes using the ratio of variances with successive halving, which gives a larger probability of misidentification compared to the strategy shown by Kaufmann et al. (2016).

In fixed-confidence BAI, Jourdan et al. (2023) Sauro (2020), Lu et al. (2021), and Lalitha et al. (2023) also provide variance-dependent BAI strategies.

Tekin & van der Schaar (2015), Guan & Jiang (2018), and Deshmukh et al. (2018) consider BAI with contextual information, but their analysis and setting are different from those employed in this study.

Barrier et al. (2022) also considers non-parametric models in BAI. While we consider the expected simple regret minimization, that work minimizes the probability of misidentification. These two settings are related but require different analyses for lower bounds. See (Komiyama et al., 2023). Additionally, there are the following critical differences between our study and theirs in non-parametric analysis using the KL divergence of Kaufmann et al. (2016). In the lower bound, our study approximates the KL divergence by the Fisher information (semiparametric influence function) around the gaps between the best and suboptimal arms are zero $\Delta^a(P) \to 0$. We do not assume the boundedness of $Y_t^a$. In the upper bound, we utilize the central limit theorem for deriving tight results. In contrast, Barrier et al. (2022) assumes the boundedness of $Y_t^a$. Then, bounding the KL divergences using the boundedness without using the small gap (fixed $\Delta^a(P)$). Because we employ the worst-case analysis, which implies $\Delta^a(P) \approx 1/\sqrt{T}$ (see Section 3), we could naturally develop non-parametric results. However, we cannot employ such an approximation in Barrier et al. (2022) because it considers lower and upper bounds under a fixed $P$ or fixed $\Delta^a(P)$.

### A.2 Efficient Average Treatment Effect Estimation

Efficient estimation of ATE via adaptive experiments constitutes another area of related literature. van der Laan (2008) and Hahn et al. (2011) propose experimental design methods for more efficient estimation of ATE by utilizing covariate information in treatment assignment. Despite the marginalization of covariates, their methods are able to reduce the asymptotic variance of estimators. Karlan & Wood (2014) applies the method of Hahn et al. (2011) to examine the response of donors to new information regarding the effectiveness of a charity. Subsequently, Tabord-Meehan (2022) and Kato et al. (2020) have sought to improve upon these studies, and more recently, Gupta et al. (2021) have proposed the use of instrumental variables in this context.

For two armed bandits, Adusumilli (2022) demonstrates the minimax optimality for the expected simple regret under the limit-of-experiment framework, utilizing a diffusion process framework. Armstrong (2022) also analyzes the minimax optimal strategy under the limit-of-experiment framework and establishes that the Neyman allocation is minimax optimal.

### A.3 Other Related Work

Our arguments are inspired by limit-of-experiments framework (Le Cam, 1986; van der Vaart, 1998; Hirano & Porter, 2009). Within this framework, we can approximate the statistical experiment by a Gaussian distribution using the CLT. Hirano & Porter (2009) relates the asymptotic optimality of statistical decision rules (Manski, 2000; 2002; 2004; Dehejia, 2005) to the framework.

The AIPW estimator has been extensively used in the fields of causal inference and semiparametric inference (Tsiatis, 2007; Bang & Robins, 2005; Chernozhukov et al., 2018). More recently, the estimator has also been utilized in other MAB problems, as seen in Kim et al. (2021) and Ito et al. (2022).

Our problem is also closely related to theories of statistical decision-making (Wald, 1949; Manski, 2000; 2002; 2004), limits of experiments (Le Cam, 1972; van der Vaart, 1998), and semiparametric theory (Hahn, 1998), not only to BAI. Among them, semiparametric theory plays an essential role because it allows us to characterize the lower bounds with the semiparametric analog of the Fisher information (van der Vaart, 1998).

## B  PRELIMINARIES

Let $W_i$ be a random variable with probability measure $P$. Let $\mathcal{F}_n = \sigma(W_1, W_2, \ldots, W_n)$.

**Definition B.1.** [Uniform integrability, Hamilton (1994), p. 191] Let $W_t \in \mathbb{R}$ be a random variable with a probability measure $P$. A sequence $\{W_t\}$ is said to be uniformly integrable if for every $\epsilon > 0$ there exists a number $c > 0$ such that

$$\mathbb{E}_P[|W_t| \cdot \mathbb{1}[|W_t| \geq c]] < \epsilon$$

for all $t$.

**Proposition B.2** (Sufficient conditions for uniform integrability; Proposition 7.7, p. 191. Hamilton (1994)). *Let $W_t, Z_t \in \mathbb{R}$ be random variables. Let $P$ be a probability measure of $Z_t$. (a) Suppose there exist $r > 1$ and $M < \infty$ such that $\mathbb{E}_P[|W_t|^r] < M$ for all $t$. Then $\{W_t\}$ is uniformly integrable. (b) Suppose there exist $r > 1$ and $M < \infty$ such that $\mathbb{E}_P[|Z_t|^r] < M$ for all $t$. If $W_t = \sum_{j=-\infty}^{\infty} h_j Z_{t-j}$ with $\sum_{j=-\infty}^{\infty} |h_j| < \infty$, then $\{W_t\}$ is uniformly integrable.*

**Proposition B.3** ($L^r$ convergence theorem, p 165, Loeve (1977)). *Let $W_t$ be a random variable with probability measure $P$ and $w$ be a constant. Let $0 < r < \infty$, suppose that $\mathbb{E}_P[|W_t|^r] < \infty$ for all $t$ and that $W_t \xrightarrow{P} z$ as $n \to \infty$. The following are equivalent:*

*(i) $W_t \to w$ in $L^r$ as $t \to \infty$;*

*(ii) $\mathbb{E}_P[|W_t|^r] \to \mathbb{E}_P[|w|^r] < \infty$ as $t \to \infty$;*

*(iii) $\{|W_t|^r, t \geq 1\}$ is uniformly integrable.*

**Definition B.4.** For $\mathcal{F}_t$ equal to the $\sigma$-field generated by $\xi_1, \ldots, \xi_t$, $\{W_t, \mathcal{F}_t, t \geq 1\}_{t=1}^{\infty}$ is a martingale if for all $t \geq 1$, we have

$$\mathbb{E}[W_{t+1} | \mathcal{F}_t] = W_t.$$

If $\mathbb{E}[W_{t+1} | \mathcal{F}_t] = 0$, $\{W_t, \mathcal{F}_t, t \geq 1\}_{t=1}^{\infty}$ is a martingale difference sequence.

**Proposition B.5** (Weak Law of Large Numbers for Martingale, Hall et al. (1980)). *Let $\{S_t = \sum_{s=1}^{t} W_s, \mathcal{F}_t, t \geq 1\}$ be a martingale and $\{b_t\}$ a sequence of positive constants with $b_t \to \infty$ as $t \to \infty$. Then, writing $W_{ts} = W_s \mathbb{1}[|W_s| \leq b_t]$, $1 \leq s \leq t$, we have that $b_t^{-1} S_t \xrightarrow{P} 0$ as $t \to \infty$ if*

**(i)** $\sum_{s=1}^{t} P(|W_s| > b_t) \to 0$;

**(ii)** $b_t^{-1} \sum_{s=1}^{t} \mathbb{E}[W_{ts} | \mathcal{F}_{s-1}] \xrightarrow{P} 0$, *and;*

**(iii)** $b_t^{-2} \sum_{s=1}^{t} \left\{ \mathbb{E}[W_{ts}^2] - \mathbb{E}[\mathbb{E}[W_{ts} | \mathcal{F}_{s-1}]]^2 \right\} \to 0$.

**Proposition B.6** (Central Limit Theorem for a Martingale Difference Sequence; from Proposition 7.9, p. 194, Hamilton (1994); also see White (1984)). *Let $\{(S_t = \sum_{s=1}^{t} W_t, \mathcal{F}_t)\}_{t=1}^{\infty}$ be a martingale with $\mathcal{F}_t$ equal to the $\sigma$-field generated by $W_1, \ldots, W_t$. Suppose that*

**(a)** $\mathbb{E}[W_t^2] = \sigma_t^2$, *a positive value with $(1/T) \sum_{t=1}^{T} \sigma_t^2 \to \sigma^2$, a positive value;*

**(b)** $\mathbb{E}[|W_t|^r] < \infty$ *for some $r > 2$;*

**(c)** $(1/T) \sum_{t=1}^{T} W_t^2 \xrightarrow{p} \sigma^2$.

*Then $S_T \xrightarrow{d} \mathcal{N}(\mathbf{0}, \sigma^2)$.*

**Proposition B.7** (Rate of convergence in the CLT; from Theorem 3.8, p 88, Hall et al. (1980)). *Let $\{(S_t = \sum_{s=1}^{t} W_t, \mathcal{F}_t)\}_{t=1}^{\infty}$ be a martingale with $\mathcal{F}_t$ equal to the $\sigma$-field generated by $W_1, \ldots, W_t$. Let*

$$V_t^2 = \sum_{s=1}^{t} \mathbb{E}[W_s^2 | \mathcal{F}_{t-1}] \qquad 1 \leq t \leq T.$$

*Suppose that for some $\alpha > 0$ and constants $M$, $C$ and $D$,*

$$\max_{t \leq T} \mathbb{E}_P[\exp(|\sqrt{t}W_t|^\alpha)] < M,$$

*and*

$$\mathbb{P}_P\left(|V_t^2 - 1| > D/\sqrt{t}(\log t)^{2+2/\alpha}\right) \leq Ct^{-1/4}(\log t)^{1+1/\alpha}.$$

*Then, for $T \geq 2$,*

$$\sup_{-\infty < x < \infty} \left|\mathbb{P}_P(S_T \leq x) - \Phi(x)\right| \leq AT^{-1/4}(\log T)^{1+1/\alpha},$$

*where the constant $A$ depends only on $\alpha$, $M$, $C$, and $D$.*

The Chernoff bound yields the following inequality.

**Proposition B.8.** *Let $\{S_t = \sum_{s=1}^t W_s, \mathcal{F}_t, t \geq 1\}_{t=1}^\infty$ be a martingale difference sequence with $\mathcal{F}_t$ equal to the $\sigma$-field generated by $W_1, \ldots, W_t$ and suppose that there exist $C_W$ such that for all $t \in \mathbb{N}$ and all $\lambda \in \mathbb{R}$,*

$$\mathbb{E}_P\left[\exp\left(\lambda W_t\right)|\mathcal{F}_{t-1}\right] \leq \exp\left(\frac{\lambda^2 C_W}{2}\right);$$

*that is, conditionally sub-Gaussian. Then, it holds that $\sum_{t=1}^T Z_t$ is sub-Gaussian, and for $\varepsilon \in \mathbb{R}$,*

$$\mathbb{P}_P\left(\sum_{t=1}^T W_t \geq \varepsilon\right) \leq \exp\left(-\frac{\varepsilon^2}{2TC_W}\right).$$

## C    NON-ASYMPTOTIC LOWER BOUNDS FOR BANDIT MODELS WITH BOUNDED SUPPORTS

First, we introduce an existing lower bound for bounded bandit models. Let us denote the class of bandit models with bounded outcomes by $P^{[0,1]}$, where each potential outcome $Y_t^a$ is in $[0, 1]$. Then, Bubeck et al. (2011) proposes the following lower bound, which holds for $P^{[0,1]}$.

**Proposition C.1.** *For all $T \geq K \geq 2$, any strategy $\pi \in \Pi$ satisfies $\sup_{P \in P^{[0,1]}} \mathbb{E}_P[r_T(P)(\pi)] \geq \frac{1}{20}\sqrt{\frac{K}{T}}$.*

This lower bound only requires that the support of the bandit models in $P^{[0,1]}$ is bounded.

For this non-asymptotic lower bound, Bubeck et al. (2011) shows that a strategy with the uniform sampling rule and empirical best arm (EBA) recommendation rule is optimal, where we draw $A_t = a$ with probability $1/K$ for all $a \in [K]$ and $t \in [T]$ and recommend an arm with the highest sample average of the observed outcomes. We call this strategy the uniform-EBA strategy.

**Proposition C.2** (Non-asymptotic optimality of the uniform-EBA strategy). *Under the uniform-EBA strategy $\pi^{\text{Uniform-EBM}}$, for $T = K\lfloor T/K \rfloor$, $\sup_{P \in \mathcal{P}^{[0,1]}} \mathbb{E}_P\left[r_T\left(\pi^{\text{Uniform-EBM}}\right)\right] \leq 2\sqrt{\frac{K \log K}{T+K}}$.*

Thus, the upper bound matches the distribution-free lower bound if we ignore the $\log K$ and constant terms.

Although the uniform-EBA strategy is nearly optimal, a question remains whether more knowledge about the class of bandit models could be used to derive a tight lower bound and propose another optimal strategy consistent with the novel lower bound. To answer this question, we consider the asymptotic evaluation and derive a tight lower bound for bandit models with a fixed variance.

## D    PROOF OF THE ASYMPTOTIC LOWER BOUND FOR MULTI-ARMED BANDITS (THEOREM 3.4)

In this section, we provide the proof of Theorems 3.4. Our lower bound derivation is based on arguments of a change-of-measure and semiparametric efficiency. The change-of-measure arguments

have been extensively used in the bandit literature (Lai & Robbins, 1985). The semiparametric efficiency is employed for deriving the lower bound of the KL divergence with a two-order Taylor expansion. Our proof is inspired by van der Vaart (1998), and Murphy & van der Vaart (1997).

We prove the asymptotic lower bound through the following steps. We first introduce lower bounds for the probability of misidentification, shown by Kaufmann et al. (2016). In Appendix D.1, we define observed-data bandit models, which are distributions of observations that differ from full-data bandit models $P \in \mathcal{P}^*$. In Appendix D.2, we define submodels of the observed-data bandit models, which parametrize nonparametric bandit models by using parameters of gaps of the expected outcomes of the best and suboptimal arms. These parameters serve as technical devices for the proof. In Appendix D.3, we then decompose the expected simple regret into the gap parameters and the probability of misidentification, and apply the lower bound of Kaufmann et al. (2016) for the probability of misidentification. The lower bound is characterized by the KL divergence of the observed-data bandit models, which we expand around the gap parameters in Appendix D.4. We then derive the semiparametric efficient influence function, which bounds the second term of the Taylor expansion of the KL divergence in Appendix D.5, and compute the worst-case bandit model in Appendix D.6. Finally, we derive the target allocation from the lower bound in Appendix D.7.

Let $f_P^a(y^a|x)$ and $\zeta_P(x)$ be a density function of $Y_t^a$ and $X_t$ under a model $P$. Kaufmann et al. (2016) derives the following result based on the change-of-measure argument, which is the principal tool in our lower bound. Let us define a density of $(Y^1, Y^2, \ldots, Y^K, X)$ under a bandit model $P \in \mathcal{P}^*$ as

$$p(y^1, y^2, \ldots, y^K, x) = \prod_{a \in [K]} f_P^a(y^a|x)\zeta_P(x)$$

**Proposition D.1** (Lemma 1 and Remark 2 in Kaufmann et al. (2016))**.** *For any two bandit model $P, Q \in \mathcal{P}^*$ with $K$ arms such that for all $a \in [K]$, the distributions $P^a$ and $Q^a$ are mutually absolutely continuous. Then,*

$$\sup_{\mathcal{E} \in \mathcal{F}_T} \left| \mathbb{P}_P(\mathcal{E}) - \mathbb{P}_Q(\mathcal{E}) \right| \leq \sqrt{\frac{\mathbb{E}_P\left[L_T(P, Q)\right]}{2}}$$

Recall that $d(p, q)$ indicates the KL divergence between two Bernoulli distributions with parameters $p, q \in (0, 1)$.

This "transportation" lemma provides the distribution-dependent characterization of events under a given bandit model $P$ and corresponding perturbed bandit model $P'$.

Between two bandit models $P, Q \in \mathcal{P}^*$, following the proof of Lemma 1 in Kaufmann et al. (2016), we define the log-likelihood ratio as

$$L_T(P, Q) = \sum_{t=1}^{T} \sum_{a \in [K]} \mathbb{1}[A_t = a] \log\left(\frac{f_P^a(Y_t^a|X_t)\zeta_P(X_t)}{f_Q^a(Y_t^a|X_t)\zeta_Q(X_t)}\right).$$

We consider an approximation of $\mathbb{E}_Q[L_T]$ under an appropriate alternative hypothesis $Q \in \mathcal{P}^*$ when the gaps between the expected outcomes of the best arm and suboptimal arms are small.

## D.1    OBSERVED-DATA BANDIT MODELS

For each $x \in \mathcal{X}$, let us define an average allocation ratio under a bandit model $P, Q \in \mathcal{P}^*$ as

$$\frac{1}{T} \sum_{t=1}^{T} \mathbb{E}_P\left[\mathbb{1}[A_t = a] | X_t = x\right] = \kappa_{T,P}(a|x)$$

This quantity represents an average sample allocation to each arm $a$ under a strategy.

**Lemma D.2.** *For $P, Q \in \mathcal{P}^*$,*

$$\mathbb{E}_P[L_T(P, Q)] = T \sum_{a \in [K]} \mathbb{E}_P\left[\mathbb{E}_P\left[\log \frac{f_P^a(Y^a|X)\zeta_P(X)}{f_Q^a(Y^a|X)\zeta_Q(X)} | X\right] \kappa_{T,P}(a|X)\right].$$

Here, recall that $A_t$ is only based on the past observations $\mathcal{F}_{t-1}$ and observed context $X_t$ and independent from $(Y_t^1, \ldots, Y_t^K)$. According to this proposition, we can consider hypothetical observed data generated as

$$(\widetilde{Y}_t, \widetilde{A}_t, X_t) \overset{\text{i.i.d}}{\sim} \prod_{a \in [K]} \{f_P^a(y^a|x)\kappa_{T,P}(a|X)\}^{\mathbb{1}[d=a]} \zeta_P(x).$$

We present the proof in Appendix E. Then, the expectation of $L_T(P,Q) = \sum_{t=1}^T \sum_{a \in [K]} \mathbb{1}[A_t = a] \log\left(\frac{f_P^a(Y_t^a|X_t)\zeta_P(X_t)}{f_Q^a(Y_t^a|X_t)\zeta_Q(X_t)}\right)$ is the same as that under the original observation $P$. Also see Eq. (2). Note that this observed data is induced by the bandit model $P \in \mathcal{P}^*$. For simplicity, we also denote $(\widetilde{Y}_t, \widetilde{A}_t, X_t)$ by $(Y_t, A_t, X_t)$ without loss of generality.

For a bandit model $P \in \mathcal{P}^*$, we consider observed-data distribution $\overline{R}_P$ with the density function given as

$$\overline{r}_P^\kappa(y, d, x) = \prod_{a \in [K]} \{f_P^a(y|x)\kappa_{T,P}(a|x)\}^{\mathbb{1}[d=a]} \zeta_P(x),$$

Let $\mathcal{R}_{\mathcal{P}^*} = \{\overline{R}_P : P \in \mathcal{P}^*\}$ be a set of all observed-data bandit models $\overline{R}_P$. Then, we have

$$\mathbb{E}_P[L_T(P,Q)] = \mathbb{E}_{\mathcal{R}_{\mathcal{P}^*}}[L_T(P,Q)] \tag{2}$$

### D.2 PARAMETRIC SUBMODELS FOR THE OBSERVED-DATA DISTRIBUTION AND TANGENT SET

The purpose of this section is to introduce parametric submodels for observed-data distribution, which is indexed by a real-valued parameter and a set of distributions contained in the larger set $\mathcal{R}$, and define the derivative of a parametric submodel as a preparation for the Taylor expansion of the log-likelihood; that is, we consider the approximation of the log-likelihood $L_T = \sum_{t=1}^T \sum_{a \in [K]} \mathbb{1}[A_t = a] \log\left(\frac{f_P^a(Y_t^a|X_t)\zeta_P(X_t)}{f_Q^a(Y_t^a|X_t)\zeta_Q(X_t)}\right)$ using $\mu^a(P)$.

This section consists of the following three parts. In the first part, we define parametric submodels as Eq. (3). Then, in the following part, we confirm the differentiability Eq. (5) and define score functions. Finally, we define a set of score functions, called a tangent set in the final paragraph.

By using the parametric submodels and tangent set, in Section D.4, we demonstrate the Taylor expansion of the log-likelihood (Lemma D.5). In this section and Section D.4, we abstractly provide definitions and conditions for the parametric submodels and do not specify them. However, in Section D.5, we show a concrete form of the parametric submodel by finding score functions satisfying the conditions imposed in this section.

**Definition of parametric submodels for the observed-data distribution** First, we define parametric submodels for the observed-data distribution $\overline{R}_P$ with the density function $\overline{r}_P(y, d, x)$ by introducing a parameter $\boldsymbol{\Delta} = (\Delta^a)_{a \in [K]}$ $\Delta^a \in \Theta$ with some compact space $\Theta$. We denote a set of parametric submodels by $\{\overline{R}_{P,\boldsymbol{\Delta}} : \boldsymbol{\Delta} \in \Theta^K\} \subset \mathcal{R}_{\mathcal{P}^*}$, which is defined as follows: for some $g : \mathbb{R} \times [K] \times \mathcal{X} \to \mathbb{R}^K$ satisfying $\mathbb{E}_P[g^a(Y_t, A_t, X_t)] = 0$ and $\mathbb{E}_P[(g^a(Y_t, A_t, X_t))^2] < \infty$, a parametric submodel $\overline{R}_{P,\boldsymbol{\Delta}}$ has a density such that

$$\overline{r}_{\boldsymbol{\Delta}}^\kappa(y, d, x) := 2c(y, d, x; \boldsymbol{\Delta}) \left(1 + \exp\left(-2\boldsymbol{\Delta}^\top g(y, d, x)\right)\right)^{-1} \overline{r}_P^a(y, d, x), \tag{3}$$

$$\mathbb{E}_{\overline{R}_{P,\boldsymbol{\Delta}}}[Y_t^d] = \int \int y \overline{r}_{\boldsymbol{\Delta}}^\kappa(y, d, x) \mathrm{d}y \mathrm{d}x = \mu^a(P) + \Delta^a + O((\Delta^a)^2). \tag{4}$$

where $c(y, d, x; \boldsymbol{\Delta})$ is some function such that $c((y, d, x; \mathbf{0}) = 1$ and $\frac{\partial}{\partial \Delta^a}\big|_{\boldsymbol{\Delta}=\mathbf{0}} \log c(y, d, x; \boldsymbol{\Delta}) = 0$ for all $(y, d, x) \in \mathbb{R} \times [K] \times \mathcal{X}$.[4] Note that the parametric submodels are usually not unique. For

---

[4]In Eq. (3), $\overline{r}_{\boldsymbol{\Delta}}^\kappa(y, d, x)$ satisfies the definition of the probability density as discussed in Example 25.15 of van der Vaart (1998).

$a \in [K]$, the parametric submodel is equivalent to $\overline{r}_P(y, a, x)$ if $\Delta^a = 0$. Let $f^a_{\Delta^a}(y|x)$ and $\zeta_{\Delta}(x)$ be the conditional density of $\widetilde{Y}^a_t$ given $X_t$ and the density of $X_t$, satisfying Eq. (3), as

$$\overline{r}^{\kappa}_{\Delta}(y, d, x) = \prod_{a \in [K]} \{f^a_{\Delta^a}(y|x)\kappa(a|x)\}^{\mathbb{1}[d=a]} \zeta_{\Delta}(x).$$

**Differentiablity and score functions of the parametric submodels for the observed-data distribution.** Next, we confirm the differentiablity of $\overline{r}^{\kappa}_{\Delta}(y, d, x)$. From the definition of the parametric submodel Eq. (3), because $\sqrt{\overline{r}^{\kappa}_{\Delta}(y, d, x)}$ is continuously differentiable for every $y, x$ given $d \in [K]$, and $\int \left(\frac{\dot{\overline{r}}^{\kappa}_{\Delta}(y,d,x)}{\overline{r}^{\kappa}_{\Delta}(y,d,x)}\right)^2 \overline{r}^{\kappa}_{\Delta}(y, d, x)\mathrm{d}m$ are well defined and continuous in $\Delta$, where $m$ is some reference measure on $(y, d, x)$, from Lemma 7.6 of van der Vaart (1998), we see that the parametric submodel has the score function $g$ in the $L_2$ sense; that is, the density $\overline{r}^{\kappa}_{\Delta}(y, d, x)$ is differentiable in quadratic mean:

$$\int \left[\overline{r}^{\kappa,1/2}_{\Delta}(y, d, x) - \overline{r}^{\kappa,1/2}_P(y, d, x) - \frac{1}{2}\Delta^{\top}g(y, d, x)\overline{r}^{\kappa,1/2}_P(y, d, x)\right]^2 \mathrm{d}m = o\left(\|\Delta\|^2\right). \quad (5)$$

In other words, the parametric submodel $\overline{r}^{\kappa,1/2}_Q$ is differentiable in quadratic mean at $\Delta = 0$ with the score function $g$.

In the following section, we specify a measurable function $g$ satisfying the conditions Eq. (3). For each $\Delta^a$ $a \in [K]$, we define the score as

$$S(y, d, x) = \frac{\partial}{\partial\Delta}\Big|_{\Delta=0} \log \overline{r}^{\kappa}_{\Delta}(y, d, x) = \begin{pmatrix} \mathbb{1}[d=1]S^1_f(y|x) + S^1_{\zeta}(x) \\ \mathbb{1}[d=2]S^2_f(y|x) + S^2_{\zeta}(x) \\ \vdots \\ \mathbb{1}[d=K]S^K_f(y|x) + S^K_{\zeta}(x) \end{pmatrix}$$

where for each $a \in [K]$, let $S^a(y, d, x) = \mathbb{1}[d=a]S^a_f(y|x) + S_{\zeta}(x)$,

$$S^a_f(y|x) = \frac{\partial}{\partial\Delta^a}\Big|_{\Delta=0} \log f^a_{\Delta^a}(y|x), \qquad S^a_{\zeta}(x) = \frac{\partial}{\partial\Delta^a}\Big|_{\Delta=0} \log \zeta_{\Delta}(x).$$

Note that $\frac{\partial}{\partial\Delta^a} \log \kappa_{T,P}(a|x) = 0$. Here, we specify $g$ in Eq. (3) as the score function of the parametric submodel as $S(y, d, x) = g(y, d, x)$, where $S^a(y, d, x) = g^a(y, d, x)$. This relationship is derived from

$$\frac{\partial}{\partial\Delta}\Big|_{\Delta=0} \log \frac{1}{1 + \exp\left(-2\Delta^{\top}g(y, d, x)\right)} = \begin{pmatrix} \frac{2g^1(y,d,x)}{\exp(2\Delta^{\top}g(y,d,x))+1} \\ \frac{2g^2(y,d,x)}{\exp(2\Delta^{\top}g(y,d,x))+1} \\ \vdots \\ \frac{2g^K(y,d,x)}{\exp(2\Delta^{\top}g(y,d,x))+1} \end{pmatrix}\Bigg|_{\Delta=0} = \begin{pmatrix} g^1(y, d, x) \\ g^2(y, d, x) \\ \vdots \\ g^K(y, d, x) \end{pmatrix}.$$

**Definition of the tangent set.** Recall that parametric submodels and corresponding score functions are not unique. Here, we consider a set of score functions. For a set of the parametric submodels $\{\overline{R}_{P,\Delta} : \Delta \in \Theta^K\}$, we obtain a corresponding set of score functions in the Hilbert space $L_2(\overline{R}_P)$, which we call a tangent set of $\mathcal{R}$ at $\overline{R}_P$ and denote it by $\dot{\mathcal{R}}$. Because $\mathbb{E}_{\overline{R}_P}[g^2]$ is automatically finite, the tangent set can be identified with a subset of the Hilbert space $L_2(\overline{R}_P)$, up to equivalence classes. For our parametric submodels, the tangent set at $\overline{R}_P$ in $L_2(\overline{R}_P)$ is given as

$$\dot{\mathcal{R}} = \left\{\begin{pmatrix} \mathbb{1}[d=1]S^1_f(y|x) + S^1_{\zeta}(x) \\ \mathbb{1}[d=2]S^2_f(y|x) + S^2_{\zeta}(x) \\ \vdots \\ \mathbb{1}[d=K]S^K_f(y|x) + S^K_{\zeta}(x) \end{pmatrix}\right\}.$$

A linear space of the tangent set is called a *tangent space*. We also define $\dot{\mathcal{R}}^a = \Big\{\left(\mathbb{1}[d=a]S^a_f(y|x) + S^a_{\zeta}(x)\right)\Big\}$.

## D.3 CHANGE-OF-MEASURE

We consider a set of bandit models $\mathcal{P}^\dagger \subset \mathcal{P}^*$ such that $P \in \mathcal{P}^\dagger$, $a \in [K]$, and $x \in \mathcal{X}$, $\mu^a(P)(x) = \mu^a(P)$. Before a bandit process begins, we fix $P^\sharp \in \mathcal{P}^\dagger$ such that $\mu^1(P^\sharp) = \cdots = \mu^K(P^\sharp) = \mu(P^\sharp)$. We choose one arm $d \in [K]$ as the best arm following a multinomial distribution with parameters $(e^1, e^2, \ldots, e^K)$, where $e^a \in [0,1]$ for all $a \in [K]$ and $\sum_{a \in [K]} e^a = 1$; that is, the expected outcome of the chosen arm $d$ is the highest among the arms. Let $\boldsymbol{\Delta}$ be a set of parameters such that $\boldsymbol{\Delta} = (\Delta^c)_{c \in [K]}$, where $\Delta^c \in (0, \infty)$. Let $\boldsymbol{\Delta}^{(d)}$ be a set of parameters such that $\boldsymbol{\Delta}^{(d)} = (0, \ldots, \Delta^d, \ldots, 0)$. Then, for each chosen $d \in [K]$, let $Q_{\boldsymbol{\Delta}^{(d)}} \in \mathcal{P}^\dagger$ be another bandit model such that $d = \arg\max_{a \in [K]} \mu^a(Q_{\boldsymbol{\Delta}^{(d)}})$, $\mu^b(Q_{\boldsymbol{\Delta}^{(d)}}) = \mu(P^\sharp)$ for $b \in [K]\backslash\{d\}$, and $\mu^d(Q_{\boldsymbol{\Delta}^{(d)}}) - \mu(P^\sharp) = \Delta^d + O\left((\Delta^d)^2\right)$. For each $d \in [K]$, we consider $\overline{R}_{P^\sharp, \boldsymbol{\Delta}^{(d)}} \in \mathcal{R}_{\mathcal{P}^\dagger} \subset \mathcal{R}_{\mathcal{P}^*}$ such that the following equation holds:

$$
\begin{aligned}
L_T(P^\sharp, Q_{\boldsymbol{\Delta}^{(d)}}) &= \sum_{t=1}^{T} \sum_{a \in [K]} \left\{ \mathbb{1}[A_t = a] \log\left(\frac{f_{P^\sharp}^a(Y_t^a|X_t)}{f_{Q_{\boldsymbol{\Delta}^{(d)}}}^a(Y_t^a|X_t)}\right) + \log\left(\frac{\zeta_{P^\sharp}(X_t)}{\zeta_{Q_{\boldsymbol{\Delta}^{(d)}}}(X_t)}\right) \right\} \\
&= \sum_{t=1}^{T} \left\{ \mathbb{1}[A_t = d] \log\left(\frac{f_{P^\sharp}^d(Y_t^d|X_t)}{f_{Q_{\boldsymbol{\Delta}^{(d)}}}^d(Y_t^d|X_t)}\right) + \log\left(\frac{\zeta_{P^\sharp}(X_t)}{\zeta_{Q_{\boldsymbol{\Delta}^{(d)}}}(X_t)}\right) \right\} \\
&= \sum_{t=1}^{T} \left\{ \mathbb{1}[A_t = d] \log\left(\frac{f_{P^\sharp}^d(Y_t^d|X_t)}{f_{\boldsymbol{\Delta}^{(d)}}^d(Y_t^d|X_t)}\right) + \log\left(\frac{\zeta_P(X_t)}{\zeta_{\boldsymbol{\Delta}^{(d)}}(X_t)}\right) \right\}.
\end{aligned}
$$

Then, let $L_T^a(P^\sharp, Q_{\boldsymbol{\Delta}^{(d)}})$ be $\sum_{t=1}^{T} \left\{ \mathbb{1}[A_t = d] \log\left(\frac{f_{P^\sharp}^d(Y_t^d|X_t)}{f_{\boldsymbol{\Delta}^{(d)}}^d(Y_t^d|X_t)}\right) + \log\left(\frac{\zeta_P(X_t)}{\zeta_{\boldsymbol{\Delta}^{(d)}}(X_t)}\right) \right\}$. Under the class of bandit models, we show the following lemma.

**Lemma D.3.** *Any null consistent BAI strategy satisfies*

$$
\sup_{P \in \mathcal{P}^*} \mathbb{E}_P[r_T(P)(\pi)] \geq \sup_{\boldsymbol{\Delta} \in (0,\infty)^K} \sum_{d \in [K]} e^d \Delta^d \left\{ 1 - \mathbb{P}_{P^\sharp}(\widehat{a}_T = d) - \sqrt{\frac{\mathbb{E}_{P^\sharp}\left[L_T^d(P^\sharp, Q_{\boldsymbol{\Delta}^{(d)}})\right]}{2}} + O\left(\Delta^d\right) \right\}.
$$

*Proof of Lemma D.3.* First, we decompose the expected simple regret by using the definition of $\mathcal{P}^\dagger$ as

$$
\begin{aligned}
&\sup_{P \in \mathcal{P}^*} \mathbb{E}_P[r_T(P)(\pi)] \\
&= \sup_{P \in \mathcal{P}^*} \sum_{b \in [K]} \left\{ \max_{a \in [K]} \mu^a(P) - \mu^b(P) \right\} \mathbb{P}_P(\widehat{a}_T = b) \\
&\geq \sup_{\boldsymbol{\Delta} \in (0,\infty)^K} \sum_{d \in [K]} e^d \sum_{b \in [K]\backslash\{d\}} \left(\mu^d(Q_{\boldsymbol{\Delta}^{(d)}}) - \mu^b(Q_{\boldsymbol{\Delta}^{(d)}})\right) \mathbb{P}_{Q_{\boldsymbol{\Delta}^{(d)}}}(\widehat{a}_T = b) \\
&\geq \sup_{\boldsymbol{\Delta} \in (0,\infty)^K} \sum_{d \in [K]} e^d \sum_{b \in [K]\backslash\{d\}} \left(\mu^d(Q_{\boldsymbol{\Delta}^{(d)}}) - \mu(P^\sharp)\right) \mathbb{P}_{Q_{\boldsymbol{\Delta}^{(d)}}}(\widehat{a}_T = b) \\
&= \sup_{\boldsymbol{\Delta} \in (0,\infty)^K} \sum_{d \in [K]} e^d \left\{ \sum_{b \in [K]\backslash\{d\}} \Delta^d \mathbb{P}_{Q_{\boldsymbol{\Delta}^{(d)}}}(\widehat{a}_T = b) + O\left((\Delta^d)^2\right) \right\} \\
&= \sup_{\boldsymbol{\Delta} \in (0,\infty)^K} \sum_{d \in [K]} e^d \left\{ \Delta^d \mathbb{P}_{Q_{\boldsymbol{\Delta}^{(d)}}}(\widehat{a}_T \neq d) + O\left((\Delta^d)^2\right) \right\} \\
&= \sup_{\boldsymbol{\Delta} \in (0,\infty)^K} \sum_{d \in [K]} e^d \left\{ \Delta^d \left(1 - \mathbb{P}_{Q_{\boldsymbol{\Delta}^{(d)}}}(\widehat{a}_T = d)\right) + O\left((\Delta^d)^2\right) \right\}.
\end{aligned}
$$

From Propositions D.5 and D.1. and the definition of null consistent strategies,

$$
\sup_{\boldsymbol{\Delta} \in (0,\infty)^K} \sum_{d \in [K]} e^d \left\{ \Delta^d \left(1 - \mathbb{P}_{Q_{\boldsymbol{\Delta}^{(d)}}}(\widehat{a}_T = d)\right) + O\left((\Delta^d)^2\right) \right\}
$$

$$= \sup_{\boldsymbol{\Delta} \in (0,\infty)^K} \sum_{d \in [K]} e^d \left\{ \Delta^d \left( 1 - \mathbb{P}_{P^\sharp} \left( \widehat{a}_T = d \right) + \mathbb{P}_{P^\sharp} \left( \widehat{a}_T = d \right) - \mathbb{P}_{Q_{\boldsymbol{\Delta}(d)}} \left( \widehat{a}_T = d \right) \right) + O \left( (\Delta^d)^2 \right) \right\}$$

$$\geq \sup_{\boldsymbol{\Delta} \in (0,\infty)^K} \sum_{d \in [K]} e^d \left\{ \Delta^d \left\{ 1 - \mathbb{P}_{P^\sharp} \left( \widehat{a}_T = d \right) - \sqrt{\frac{\mathbb{E}_{P^\sharp} \left[ L_T^d(P^\sharp, Q_{\boldsymbol{\Delta}(d)}) \right]}{2}} \right\} + O \left( (\Delta^d)^2 \right) \right\}.$$

The proof is complete. $\square$

## D.4 Semiparametric Likelihood Ratio

In this section and the next section (Appendix D.5), our goal is to prove the following lemma.

**Lemma D.4.**

$$\mathbb{E}_{P^\sharp} \left[ L_T^d(P^\sharp, Q_{\boldsymbol{\Delta}(d)}) \right] \leq \frac{T (\Delta^a)^2}{2 \mathbb{E}_P \left[ \frac{(\sigma^d(X))^2}{w(d|X)} \right]} + O \left( T (\Delta^a)^3 \right).$$

We consider Taylor expansion of the log-likelihood ratio $L_T$ defined between $P, Q \in \mathcal{P}^\dagger$. We consider an approximation of $L_T$ around a parametric submodel. Because there can be several score functions for our parametric submodel due to directions of the derivative, we find a parametric submodel that has a score function with the largest variance, called a least-favorable parametric submodel (van der Vaart, 1998). Our Taylor expansion is upper-bounded by the variance of the score function, which corresponds to the lower bound for the probability of misidentification.

Inspired by the arguments in Murphy & van der Vaart (1997), we define the semiparametric likelihood ratio expansion to characterize the lower bound for the probability of misidentification with the semiparametric efficiency bound. Note again that the details are different from them owing to the difference in the parameters submodels.

As a preparation, we define a parameter $\mathbb{E}_{\overline{R}_{P,\boldsymbol{\Delta}(a)}}[Y_t^a]$ as a function $\psi^a : \mathcal{R} \mapsto \mathbb{R}$ such that $\psi^a(\overline{R}_{P,\boldsymbol{\Delta}(a)}) = \mathbb{E}_{\overline{R}_{P,\boldsymbol{\Delta}(a)}}[Y_t^a]$. The information bound for $\psi^a(\overline{R}_{P,\boldsymbol{\Delta}(a)})$ of interest is called semiparametric efficiency bound. Let $\overline{\mathrm{lin}}\dot{\mathcal{R}}$ be the closure of the tangent set. Then, $\psi^a(\overline{R}_{P,\boldsymbol{\Delta}(a)})$ is pathwise differentiable relative to the tangent set $\dot{\mathcal{R}}^a$ if and only if there exists a function $\widetilde{\psi} \in \overline{\mathrm{lin}}\dot{\mathcal{R}}$ such that

$$\frac{\partial}{\partial \Delta^a} \Big|_{\Delta^a = 0} \psi^a(\overline{R}_{P,\boldsymbol{\Delta}(a)}) = \mathbb{E}_{\overline{R}_{P,\boldsymbol{\Delta}(a)}} \left[ \widetilde{\psi}_P^a(Y_t, A_t, X_t) S^a(Y_t, A_t, X_t) \right]. \tag{6}$$

This function $\widetilde{\psi}$ is called the *semiparametric influence function*. Note that the RHS of Eq. (6) is calculated as follows:

$$\mathbb{E}_{\overline{R}_{P,\boldsymbol{\Delta}(a)}} \left[ \widetilde{\psi}_P^a(Y_t, A_t, X_t) S^a(Y_t, A_t, X_t) \right] = \int \int y S_f^a(y|x) f_{\Delta^a}^a(y|x) \zeta_{\boldsymbol{\Delta}(a)}(x) \mathrm{d}y \mathrm{d}x + \int \mu^a(x) S_\zeta(x) \zeta_{\boldsymbol{\Delta}(a)}(x) \mathrm{d}x. \tag{7}$$

Then, we prove the following lemma:

**Lemma D.5.** *For $P \in \mathcal{P}^\dagger$,*

$$\mathbb{E}_P[L_T^a(P, Q)] \leq \frac{1}{2} \frac{T (\Delta^a)^2}{\mathbb{E}_P \left[ \left( \widetilde{\psi}_P^a(Y_t, A_t, X_t) \right)^2 \right]} + O \left( T (\Delta^a)^3 \right).$$

To prove this lemma, we define

$$\ell_{\boldsymbol{\Delta}}^a(y, d, x) = \mathbb{1}[d = a] \left\{ \log f_{\Delta^a}^a(y^a|x) \right\} + \log \zeta_{\boldsymbol{\Delta}}(x).$$

Then, by using the parametric submodel defined in the previous section,

$$L_T^a(P, Q) = \sum_{t=1}^T \mathbb{1}[A_t = a] \log \left( \frac{f_P^a(Y_t^a|X_t) \zeta_P(X_t)}{f_Q^a(Y_t^a|X_t) \zeta_Q(X_t)} \right)$$

$$= \sum_{t=1}^{T} \mathbb{1}[A_t = a] \log \left( \frac{f_P^a(Y_t^a | X_t) \zeta_P(X_t)}{f_{\Delta^a}^a(Y_t^a | X_t) \zeta_{\Delta}(X_t)} \right)$$

$$= \sum_{t=1}^{T} \left( -\frac{\partial}{\partial \Delta^a} \Big|_{\Delta^a = 0} \ell_{\Delta^{(a)}}^a(Y_t, A_t, X_t) \Delta^a - \frac{\partial^2}{\partial (\Delta^a)^2} \Big|_{\Delta^a = 0} \ell_{\Delta^{(a)}}^a(Y_t, A_t, X_t) \frac{(\Delta^a)^2}{2} + O\left((\Delta^a)^3\right) \right).$$

Here, note that

$$\frac{\partial}{\partial \Delta^a} \Big|_{\Delta^a = 0} \ell_{\Delta^{(a)}}^a(Y_t, A_t, X_t) = S^a(Y_t, A_t, X_t) = g^a(Y_t, A_t, X_t)$$

$$\frac{\partial}{\partial (\Delta^a)^2} \Big|_{\Delta^a = 0} \ell_{\Delta^{(a)}}^a(Y_t, A_t, X_t) = -\left(S^a(Y_t, A_t, X_t)\right)^2.$$

By using the expansion, we evaluate $\mathbb{E}_P[L_T^a]$. Here, by definition, $\mathbb{E}_P[S^a(Y_t, A_t, X_t)] = 0$. Therefore, we consider an upper bound of $\frac{1}{\mathbb{E}_P\left[(S^a(Y_t, A_t, X_t))^2\right]}$ for $S \in \dot{\mathcal{R}}$.

Then, we prove the following lemma on the upper bound for $\frac{1}{\mathbb{E}_P\left[(S^a(Y_t, A_t, X_t))^2\right]}$:

**Lemma D.6.** *For* $P \in \mathcal{P}^{\dagger}$,

$$\sup_{S \in \dot{\mathcal{R}}} \frac{1}{\mathbb{E}_P\left[(S^a(Y_t, A_t, X_t))^2\right]} \leq \mathbb{E}_P\left[\left(\widetilde{\psi}_P^a(Y_t, A_t, X_t)\right)^2\right]$$

*Proof.* From the Cauchy-Schwarz inequality, we have

$$1 = \mathbb{E}_P\left[\widetilde{\psi}_P^a(Y_t, A_t, X_t) S^a(Y_t, A_t, X_t)\right] \leq \sqrt{\mathbb{E}_P\left[\left(\widetilde{\psi}_P^a(Y_t, A_t, X_t)\right)^2\right]} \sqrt{\mathbb{E}_P\left[(S^a(Y_t, A_t, X_t))^2\right]}.$$

Therefore,

$$\sup_{S \in \dot{\mathcal{R}}} \frac{1}{\mathbb{E}_P\left[(S^a(Y_t, A_t, X_t))^2\right]} \leq \mathbb{E}_P\left[\left(\widetilde{\psi}_P^a(Y_t, A_t, X_t)\right)^2\right].$$

$\square$

According to this lemma, to derive the upper bound for $\frac{1}{\mathbb{E}_P\left[(S^a(Y_t, A_t, X_t))^2\right]}$, let us define the *semiparametric efficient score* $S_{\text{eff}}^a(Y_t, A_t, X_t) \in \overline{\text{lin}}\dot{\mathcal{R}}^a$ as

$$S_{\text{eff}}^a(Y_t, A_t, X_t) = \frac{\widetilde{\psi}_P^a(Y_t, A_t, X_t)}{\mathbb{E}_P\left[\left(\widetilde{\psi}_P^a(Y_t, A_t, X_t)\right)^2\right]}.$$

Then, by using the semiparametric efficient score $S_{\text{eff}}^a(Y_t, A_t, X_t)$, we approximate the likelihood ratio as follows:

*Proof of Lemma D.5.*

$$\mathbb{E}_{P'}[L_T^a(P, Q)] = T\mathbb{E}_P\left[\frac{1}{2}\left(S^a(Y_t, A_t, X_t)\right)^2 (\Delta^a)^2 + O((\Delta^a)^3)\right]$$

$$\leq T\mathbb{E}_P\left[\frac{1}{2}\left(S_{\text{eff}}^a(Y_t, A_t, X_t)\right)^2 (\Delta^a)^2 + O((\Delta^a)^3)\right]$$

$$= \frac{1}{2} \frac{T(\Delta^a)^2}{\mathbb{E}_P\left[\left(\widetilde{\psi}_P^a(Y_t, A_t, X_t)\right)^2\right]} + O\left(T(\Delta^a)^3\right).$$

$\square$

### D.5 OBSERVED-DATA SEMIPARAMETRIC EFFICIENT INFLUENCE FUNCTION

Our remaining is task is to find $\widetilde{\psi}_P^a \in \overline{\mathrm{lin}}\dot{\mathcal{R}}$ in Eq. (6). Our derivation mainly follows Hahn (1998). We guess that $\widetilde{\psi}_P^a(Y_t, A_t, X_t)$ has the following form:

$$\widetilde{\psi}_P^a(y, d, x) = \frac{\mathbb{1}[d = a](y - \mu^a(P)(x))}{\kappa_{T,P}(a|X)} + \mu^a(P)(x) - \mu^a(P). \tag{8}$$

Then, as shown by Hahn (1998), the condition $\frac{\partial}{\partial \Delta^a}\Big|_{\boldsymbol{\Delta}^{(a)}=\mathbf{0}} \psi^a(\overline{R}_{P,\boldsymbol{\Delta}^{(a)}}) = \mathbb{E}_{\overline{R}_Q}\left[\widetilde{\psi}_P^a(Y_t, A_t, X_t)S^a(Y_t, A_t, X_t)\right]$ holds under Eq. (8) when the score functions are given as

$$S_f^a(y|x) = \frac{(y - \mu^a(P)(x))}{\kappa_{T,P}(a|x)}/\widetilde{V}^a(\kappa_{T,P}), \qquad S_\zeta^a(x) = \left(\mu^a(P)(x) - \mu^a(P)\right)/\widetilde{V}^a(\kappa_{T,P}) \quad \text{for } a \in [K],$$

where

$$\widetilde{V}^a(\kappa_{T,P}) := \mathbb{E}_P\left[\left(\frac{\mathbb{1}[d = a](y - \mu^a(P)(x))}{\kappa_{T,P}(a|X)} + \mu^a(P)(x) - \mu^a(P)\right)^2\right] = \mathbb{E}_P\left[\frac{(\sigma^a(X_t))^2}{\kappa_{T,P}(a|X_t)} + (\mu^a(P)(X_t) - \mu^a(P))^2\right].$$

Therefore,

$$S^a(y, d, x) = \left(\frac{\mathbb{1}[d = a](y - \mu^a(P)(x))}{\kappa_{T,P}(a|X)} + \mu^a(P)(x) - \mu^a(P)\right)/\widetilde{V}^a(\kappa_{T,P}).$$

Our specified score function satisfies Eq. (4) because we can confirm that

$$\psi^a(\overline{R}_{P,\mathbf{0}}) = \mu^a(P),$$

and

$$\frac{\partial}{\partial \Delta^a}\Big|_{\Delta^a=0} \psi^a(\overline{R}_{P,\boldsymbol{\Delta}^{(a)}}) = \mathbb{E}_{\overline{R}_Q}\left[\widetilde{\psi}_P^a(Y_t, A_t, X_t)S^a(Y_t, A_t, X_t)\right]$$

$$= \mathbb{E}_{\overline{R}_Q}\left[\left(\frac{\mathbb{1}[d = a](y - \mu^a(P)(x))}{\kappa_{T,P}(a|X)} + \mu^a(P)(x) - \mu^a(P)\right)^2/\widetilde{V}^a(\kappa_{T,P})\right] = 1.$$

Then, from the first-order Taylor expansion of $\psi^a(\overline{R}_{P,\boldsymbol{\Delta}^{(a)}})$ around $\Delta^a = 0$, we obtain

$$\psi^a(\overline{R}_{P,\boldsymbol{\Delta}^{(a)}}) = \psi^a(\overline{R}_{P,\mathbf{0}}) + \Delta^a \frac{\partial}{\partial \Delta^a}\Big|_{\Delta^a=0} \psi^a(\overline{R}_{P,\boldsymbol{\Delta}^{(a)}}) + O((\Delta^a)^2) = \mu^a(P) + \Delta^a + O((\Delta^a)^2).$$

Summarizing the above arguments, we obtain the following lemma.

**Lemma D.7.** *For $P \in \mathcal{P}^\dagger$, the semiparametric efficient influence function is*

$$\widetilde{\psi}_P^a(y, d, x) = \widetilde{V}^a(\kappa_{T,P})\left(\mathbb{1}[d = a]S_f^a(y|x) + S_\zeta(x)\right)$$

$$= \frac{\mathbb{1}[d = a](y - \mu^a(P)(x))}{\kappa_{T,P}(a|x)} + \mu^a(P)(x) - \mu^a(P).$$

Thus, under $g$ with our specified score functions, we can confirm that the semiparametric influence function $\widetilde{\psi}_P^a(y, d, x) = \widetilde{V}^a(\kappa_{T,P})\left(\mathbb{1}[A_t = a]S_f^a(y|x) + S_\zeta(x)\right)$ belongs to $\overline{\mathrm{lin}}\dot{\mathcal{R}}$. Note that $\mathbb{E}_{\overline{R}_Q}[S_{\mathrm{eff}}^a(Y_t, A_t, X_t)] = 0$ and

$$\mathbb{E}_{\overline{R}_Q}\left[\left(S_{\mathrm{eff}}^a(Y_t, A_t, X_t)\right)^2\right] = \left(\mathbb{E}_{\overline{R}_Q}\left[\left(\widetilde{\psi}_P^a(Y_t, A_t, X_t)\right)^2\right]\right)^{-1}.$$

In summary, from Lemmas D.5, D.6, and D.7, we obtain Lemma D.4. Note that because $\mu^a(P)(x) = \mu^a$ for $P \in \mathcal{P}^\dagger$, $\widetilde{V}^a(\kappa_{T,P}) := \mathbb{E}_P\left[\frac{(\sigma^a(X_t))^2}{\kappa_{T,P}(a|X_t)}\right]$.

## D.6 THE WORST CASE BANDIT MODEL

We show the final step of the proof.

*Proof.* Then, from Lemmas D.3 and D.7, for all $d \in [K]$, and definition of the null consistent strategy, for any $\epsilon > 0$, there exists $T_0 > 0$ such that for all $T > T_0$,

$$
\sup_{P \in \mathcal{P}^*} \mathbb{E}_P[r_T(P)(\pi)] \geq \sup_{\boldsymbol{\Delta} \in (0,\infty)^K} \sum_{d \in [K]} e^d \Delta^d \left\{ 1 - \mathbb{P}_{P^\sharp}(\widehat{a}_T = d) - \sqrt{\frac{\mathbb{E}_{P^\sharp}\left[ L_T^d(P^\sharp, Q_{\boldsymbol{\Delta}^{(d)}}) \right]}{2}} + O\left(\Delta^d\right) \right\}
$$

$$
\geq \inf_{w \in \mathcal{W}} \sup_{\boldsymbol{\Delta} \in (0,\infty)^K} \sum_{d \in [K]} e^d \Delta^d \left\{ 1 - \frac{1}{K} - \sqrt{\frac{T(\Delta^d)^2}{2\mathbb{E}_{P^\sharp}\left[ \frac{(\sigma^d(X_t))^2}{w(d|X_t)} \right]}} + O\left(T((\Delta^d)^3\right) + O\left(\Delta^d\right) \right\} - \epsilon
$$

$$
\geq \inf_{w \in \mathcal{W}} \sup_{\boldsymbol{\Delta} \in (0,\infty)^K} \sum_{d \in [K]} e^d \Delta^d \left\{ \frac{1}{2} - \sqrt{\frac{T(\Delta^d)^2}{2\mathbb{E}_{P^\sharp}\left[ \frac{(\sigma^d(X_t))^2}{w(d|X_t)} \right]}} + O\left(T((\Delta^d)^3\right) + O\left(\Delta^d\right) \right\} - \epsilon.
$$

The maximizer of $\sup_{\boldsymbol{\Delta} \in (0,\infty)^K} \sum_{d \in [K]} e^d \Delta^d \left\{ \frac{1}{2} - \sqrt{\frac{T(\Delta^d)^2}{2\mathbb{E}_{P^\sharp}\left[ \frac{(\sigma^d(X_t))^2}{w(d|X_t)} \right]}} \right\}$ is given as $\Delta^a =$

$\frac{1}{4} \sqrt{\frac{2\mathbb{E}_{P^\sharp}\left[ \frac{(\sigma^a(X_t))^2}{w(a|X_t)} \right]}{T}}$. Therefore,

$$
\sup_{P \in \mathcal{P}^*} \mathbb{E}_P[r_T(P)(\pi)] \geq \inf_{w \in \mathcal{W}} \sup_{\boldsymbol{\Delta} \in (0,\infty)^K} \sum_{d \in [K]} e^d \Delta^d \left\{ \frac{1}{2} - \sqrt{\frac{T(\Delta^d)^2}{2\mathbb{E}_{P^\sharp}\left[ \frac{(\sigma^d(X_t))^2}{w(d|X_t)} \right]}} + O\left(T(\Delta^d)^3\right) + O\left(\Delta^d\right) \right\} - \epsilon
$$

$$
\geq \frac{1}{12} \inf_{w \in \mathcal{W}} \sum_{d \in [K]} e^d \left\{ \sqrt{\frac{\mathbb{E}_{P^\sharp}\left[ \frac{(\sigma^d(X_t))^2}{w(d|X_t)} \right]}{T}} + O\left( \frac{2\mathbb{E}_{P^\sharp}\left[ \frac{(\sigma^d(X_t))^2}{w(d|X_t)} \right]}{T} \right) \right\} - \epsilon.
$$

As $T \to \infty$, letting $\epsilon \to 0$,

$$
\sup_{P \in \mathcal{P}^*} \sqrt{T} \mathbb{E}_P[r_T(P)(\pi)] \geq \frac{1}{12} \inf_{w \in \mathcal{W}} \sum_{d \in [K]} e^d \sqrt{\mathbb{E}_{P^\sharp}\left[ \frac{(\sigma^d(X_t))^2}{w(d|X_t)} \right]} + o(1).
$$

$\square$

## D.7 CHARACTERIZATION OF THE TARGET ALLOCATION RATIO

*Proof of Theorem 3.4.* We showed that any null consistent BAI strategy satisfies

$$
\sup_{P \in \mathcal{P}^*} \mathbb{E}_P[r_T(P)(\pi)] \geq \frac{1}{12} \inf_{w \in \mathcal{W}} \sum_{d \in [K]} e^d \sqrt{\frac{(\sigma^d)^2}{w(d)}} + o(1).
$$

In the tight lower bound, $e^{\widetilde{d}} = 1$ for $\widetilde{d} = \arg\max_{d \in [K]} \frac{1}{12} \sqrt{\frac{(\sigma^d)^2}{w(d)}} + o(1)$[5]. Therefore, we consider solving

$$
\inf_{w \in \mathcal{W}} \max_{d \in [K]} \sqrt{\frac{(\sigma^d)^2}{w(d)}}.
$$

---

[5]If there are multiple candidates of the best arm, we choose one of the multiple arms as the best arm with probability 1.

If there exists a solution, we can replace the inf with the min. We consider the following constrained optimization:

$$\inf_{R \in \mathbb{R}, w \in \mathcal{W}} R \tag{9}$$

$$\text{s.t.} \quad R \geq \frac{\left(\sigma^d\right)^2}{w(d)} \quad \forall d \in [K]$$

$$\sum_{a \in [K]} w(a) = 1.$$

For this problem, we derive the first-order condition, which is sufficient for the global optimality of such a convex programming problem. For Lagrangian multipliers $\lambda^d \in (-\infty, 0]$ and $\gamma \in \mathbb{R}$, we consider the following Lagrangian function:

$$L(\boldsymbol{\lambda}, \gamma; R, w) = R + \sum_{d \in [K]} \lambda^d \left\{ \frac{\left(\sigma^d\right)^2}{w(d)} - R \right\} + \gamma \left\{ \sum_{d \in [K]} w(d) - 1 \right\}.$$

Then, the optimal solutions $w^*$, $\lambda^{*d}$, $\gamma^*$, and $R^*$ of the original problem satisfies

$$1 - \sum_{d \in [K]} \lambda^{d*} = 0 \qquad \forall x \in \mathcal{X} \tag{10}$$

$$-\lambda^{d*} \frac{\left(\sigma^d\right)^2}{(w^*(d))^2} = \gamma^* \qquad \forall d \in [K], \tag{11}$$

$$\lambda^{d*} \left\{ \frac{\left(\sigma^d\right)^2}{w(d)} - R^* \right\} = 0 \tag{12}$$

$$\gamma^*(x) \left\{ \sum_{a \in [K]} w^*(a) - 1 \right\} = 0 \qquad \forall a \in [K].$$

Here, the solutions are given as

$$w^*(d) = \frac{\left(\sigma^d\right)^2}{\sum_{b \in [K]} \left(\sigma^b\right)^2},$$

$$\lambda^{d*} = w^*(d),$$

$$\gamma^*(x) = -\sum_{b \in [K]} \left(\sigma^b\right)^2.$$

Therefore,

$$\inf_{w \in \mathcal{W}} \sum_{a \in [K]} e^a \frac{1}{12} \sqrt{\mathbb{E}_{P^\sharp}\left[ \frac{(\sigma^a(X))^2}{w(a|X)} \right]} + o(1) = \frac{1}{12} \sqrt{\mathbb{E}_{P^\sharp}\left[ \sum_{a \in [K]} (\sigma^a(X))^2 \right]} \sum_{a \in [K]} e^a + o(1).$$

Since $\sum_{a \in [K]} e^a = 1$ and $\zeta_P(x) = \zeta(x)$, the proof is complete.

$\square$

Here, $\widetilde{w}(a|x) = \frac{(\sigma^a(x))^2}{\sum_{b \in [K]} (\sigma^b(x))^2}$ works as a target allocation ratio in implementation of our BAI strategy because it represents the sample average of $\mathbb{1}[A_t = a]$; that is, we design our sampling rule $(A_t)_{t \in [T]}$ for the average to be the target allocation ratio.

Although this lower bound is applicable to a case with $K = 2$, we can tighten the lower bound by changing the definition of the parametric submodel.

# E   PROOF OF LEMMA D.2

*Proof.* Let us define $\Omega_t^{a,b}(P) = \sum_{s=1}^{t} \mathbb{E}\left[\left(\xi_s^{a,b}(P)\right)^2 | \mathcal{F}_{s-1}\right]$. We can also derive a non-asymptotic upper bound for the expected simple regret if we assume a certain convergence rate on $\Omega^{a,b}(P)$, which dominates the rate of the martingale limit theorems. We show the result in Appendix H.

We have

$$\mathbb{E}_Q[L_T] = \sum_{t=1}^{T} \mathbb{E}_Q\left[\sum_{a\in[K]} \mathbb{1}\{A_t = a\} \log \frac{f_Q^a(Y_t^a|X_t)\zeta_Q(X_t)}{f_{P_0}^a(Y_t^a|X_t)\zeta_{P_0}(X_t)}\right]$$

$$= \sum_{t=1}^{T} \mathbb{E}_Q^{X_t,\mathcal{F}_{t-1}}\left[\sum_{a\in[K]} \mathbb{E}_Q^{Y_t^a,A_t}\left[\mathbb{1}[A_t = a] \log \frac{f_Q^a(Y_t^a|X_t)\zeta_Q(X_t)}{f_{P_0}^a(Y_t^a|X_t)\zeta_{P_0}(X_t)}|X_t,\mathcal{F}_{t-1}\right]\right]$$

$$= \sum_{t=1}^{T} \mathbb{E}_Q^{X_t,\mathcal{F}_{t-1}}\left[\sum_{a\in[K]} \mathbb{E}_Q\left[\mathbb{1}[A_t = a]|X_t,\mathcal{F}_{t-1}\right]\mathbb{E}_Q^{Y_t^a}\left[\log \frac{f_Q^a(Y_t^a|X_t)\zeta_Q(X_t)}{f_{P_0}^a(Y_t^a|X_t)\zeta_{P_0}(X_t)}|X_t,\mathcal{F}_{t-1}\right]\right]$$

$$= \sum_{t=1}^{T} \mathbb{E}_Q^{X_t}\left[\mathbb{E}_Q^{\mathcal{F}_t}\left[\sum_{a\in[K]} \mathbb{E}_Q\left[\mathbb{1}[A_t = a]|X_t,\mathcal{F}_{t-1}\right]\mathbb{E}_Q^{Y_t^a}\left[\log \frac{f_Q^a(Y_t^a|X_t)\zeta_Q(X_t)}{f_{P_0}^a(Y_t^a|X_t)\zeta_{P_0}(X_t)}|X_t\right]\right]\right]$$

$$= \sum_{t=1}^{T} \int \left(\sum_{a\in[K]} \mathbb{E}_Q^{\mathcal{F}_t}\left[\mathbb{E}_Q\left[\mathbb{1}[A_t = a]|X_t = x,\mathcal{F}_{t-1}\right]\right]\mathbb{E}_Q^{Y_t^a}\left[\log \frac{f_Q^a(Y_t^a|X_t)\zeta_Q(X_t)}{f_{P_0}^a(Y_t^a|X_t)\zeta_{P_0}(X_t)}|X_t = x\right]\right)\zeta_Q(x)\mathrm{d}x$$

$$= \int \sum_{a\in[K]} \left(\mathbb{E}_Q^{Y^a}\left[\log \frac{f_Q^a(Y^a|X)\zeta_Q(X)}{f_{P_0}^a(Y^a|X)\zeta_{P_0}(X)}|X = x\right]\sum_{t=1}^{T}\mathbb{E}_Q^{\mathcal{F}_t}\left[\mathbb{E}_Q\left[\mathbb{1}[A_t = a]|X_t = x,\mathcal{F}_{t-1}\right]\right]\right)\zeta_Q(x)\mathrm{d}x$$

$$= \mathbb{E}_Q^X\left[\sum_{a\in[K]} \mathbb{E}_Q^{Y^a}\left[\log \frac{f_Q^a(Y^a|X)\zeta_Q(X)}{f_{P_0}^a(Y^a|X)\zeta_{P_0}(X)}|X\right]\sum_{t=1}^{T}\mathbb{E}_Q^{\mathcal{F}_{t-1}}\left[\mathbb{E}_Q\left[\mathbb{1}[A_t = a]|X,\mathcal{F}_{t-1}\right]\right]\right],$$

where $\mathbb{E}_Q^Z$ denotes an expectation of random variable $Z$ over the distribution $Q$. We used that the observations $(Y_t^1,\ldots,Y_t^K,X_t)$ are i.i.d. across $t \in \{1,2,\ldots,T\}$. $\square$

# F   PROOF OF THE ASYMPTOTIC LOWER BOUND FOR TWO-ARMED BANDITS (THEOREM 3.8)

When $K = 2$, we define different parametric submodels from those in Section D.

**Parametric submodels for the observed-data distribution and tangent set.** In a case with $K = 2$, we consider one-parameter parametric submodels for the observed-data distribution $\overline{R}_P$ with the density function $\overline{r}_P(y,d,x)$ by introducing a parameter $\Delta \in \Theta$ with some compact space $\Theta$. We denote a set of parametric submodels by $\{\overline{R}_{P,\Delta} : \Delta \in \Theta\} \subset \mathcal{R}_{\mathcal{P}^*}$, which is defined as follows: for some $g : \mathbb{R} \times [2] \times \mathcal{X} \to \mathbb{R}$ satisfying $\mathbb{E}_P[g(Y_t,A_t,X_t)] = 0$ and $\mathbb{E}_P[(g(Y_t,A_t,X_t))^2] < \infty$, a parametric submodel $\overline{R}_{P,\Delta}$ has a density such that

$$\overline{r}_\Delta^\kappa(y,d,x) := 2c(y,d,x;\Delta)\left(1 + \exp\left(-2\Delta g(y,d,x)\right)\right)^{-1}\overline{r}_P^a(y,d,x),$$

where $c(y,d,x;\Delta)$ is some function such that $c((y,d,x;0) = 1$ and $\frac{\partial}{\partial\Delta}\big|_{\Delta=0}\log c((y,d,x;\Delta) = 0$ for all $(y,d,x) \in \mathbb{R} \times [2] \times \mathcal{X}$. Note that the parametric submodels are usually not unique. The parametric submodel is equivalent to $\overline{r}_P(y,a,x)$ if $\Delta = 0$.

Let $f_\Delta^a(y|x)$ and $\zeta_\Delta(x)$ be the conditional density of $y$ given $x$ and some density of $x$, satisfying Eq. (3) as

$$\overline{r}_\Delta^\kappa(y,d,x) = \prod_{a\in[2]} \{f_\Delta^a(y|x)\kappa(a|x)\}^{\mathbb{1}[d=a]}\zeta_\Delta(x).$$

For this parametric submodel, we develop the same argument in Section D. Note that we consider a one-parameter parametric submodel for two-armed bandits, while in Section D, we consider $K$-dimensional parametric submodels for $K$-armed bandits.

**Change-of-measure.** We consider a set of bandit models $\mathcal{P}^{\dagger\dagger} \subset \mathcal{P}^*$ such that for all $P \in \mathcal{P}^{\dagger\dagger}$, $a \in [K]$, and $x \in \mathcal{X}$, $\mu^a(P)(x) = \mu^a$. Before a bandit process begins, we fix $P^{\sharp\sharp} \in \mathcal{P}^{\dagger\dagger}$ such that $\mu^1(P^{\sharp\sharp}) = \mu^2(P^{\sharp\sharp}) = \mu(P^{\sharp\sharp})$. We choose one arm $d \in [2]$ as the best arm following a Bernoulli distribution with parameter $e \in [0, 1]$; that is, the expected outcome of the chosen arm $d$ is the highest among the arms. We choose arm 1 with probability $e$ and arm 2 with probability $1-e$. Let $\Delta \in (0, \infty)$ be a gap parameter and $Q_\Delta \in \mathcal{P}^{\dagger\dagger}$ be another bandit model such that $d = \arg\max_{a \in [2]} \mu^a(Q_\Delta)$, $\mu^b(Q_\Delta) = \mu(P^{\sharp\sharp})$ for $b \neq d$, and $\mu^d(Q_\Delta) - \mu(P^{\sharp\sharp}) = \Delta + O(\Delta^2)$. For the parameter $\Delta$, we consider $\overline{R}_{P^{\sharp\sharp}, \Delta} \in \mathcal{R}_{\mathcal{P}^{\dagger\dagger}} \subset \mathcal{R}_{\mathcal{P}^*}$ such that the following equation holds:

$$L_T(P, Q) = \sum_{t=1}^{T} \left\{ \mathbb{1}[A_t = 1] \log\left( \frac{f_P^1(Y_t^1|X_t)}{f_Q^1(Y_t^1|X_t)} \right) + \mathbb{1}[A_t = 2] \log\left( \frac{f_P^2(Y_t^2|X_t)}{f_Q^2(Y_t^2|X_t)} \right) + \log\left( \frac{\zeta_P(X_t)}{\zeta_Q(X_t)} \right) \right\}$$

$$= \sum_{t=1}^{T} \left\{ \mathbb{1}[A_t = 1] \log\left( \frac{f_P^1(Y_t^1|X_t)}{f_\Delta^a(Y_t^1|X_t)} \right) + \mathbb{1}[A_t = 2] \log\left( \frac{f_P^2(Y_t^2|X_t)}{f_\Delta^2(Y_t^2|X_t)} \right) + \log\left( \frac{\zeta_P(X_t)}{\zeta_\Delta(X_t)} \right) \right\}.$$

*Proof of Theorem 3.8.* First, we decompose the expected simple regret by using the definition of $\mathcal{P}^{\dagger\dagger}$ as

$$\sup_{P \in \mathcal{P}^*} \mathbb{E}_P[r_T(P)(\pi)]$$

$$= \sup_{P \in \mathcal{P}^*} \sum_{b \in [2]} \left\{ \max_{a \in [2]} \mu^a(P) - \mu^b(P) \right\} \mathbb{P}_P(\widehat{a}_T = b)$$

$$\geq \sup_{\Delta \in (0, \infty)} \left\{ e\left(\mu^1(Q_\Delta) - \mu^2(Q_\Delta)\right) \mathbb{P}_{Q_\Delta}(\widehat{a}_T = 2) + (1-e)\left(\mu^2(Q_\Delta) - \mu^1(Q_\Delta)\right) \mathbb{P}_{Q_\Delta}(\widehat{a}_T = 1) \right\}$$

$$= \sup_{\Delta \in (0, \infty)} \left\{ e\left(\mu^1(Q_\Delta) - \mu(P^{\sharp\sharp})\right) \mathbb{P}_{Q_\Delta}(\widehat{a}_T = 2) + (1-e)\left(\mu^2(Q_\Delta) - \mu(P^\sharp)\right) \mathbb{P}_{Q_\Delta}(\widehat{a}_T = 1) \right\}$$

$$= \sup_{\Delta \in (0, \infty)} \left\{ e\left(\Delta + O(\Delta^2)\right) \mathbb{P}_{Q_\Delta}(\widehat{a}_T = 2) + (1-e)\left(\Delta + O(\Delta^2)\right) \mathbb{P}_{Q_\Delta}(\widehat{a}_T = 1) \right\}$$

$$= \sup_{\Delta \in (0, \infty)} \left\{ e\Delta \mathbb{P}_{Q_\Delta}(\widehat{a}_T = 2) + (1-e)\Delta \mathbb{P}_{Q_\Delta}(\widehat{a}_T = 1) + O(\Delta^2) \right\}$$

$$= \sup_{\Delta \in (0, \infty)} \left\{ e\Delta\left(1 - \mathbb{P}_{Q_\Delta}(\widehat{a}_T = 1)\right) + (1-e)\Delta\left(1 - \mathbb{P}_{Q_\Delta}(\widehat{a}_T = 2)\right) + O(\Delta^2) \right\}.$$

From Propositions D.5 and D.1 and definition of the null consistent strategy,

$$\sup_{\Delta \in (0, \infty)} \left\{ e\Delta\left(1 - \mathbb{P}_{Q_\Delta}(\widehat{a}_T = 1)\right) + (1-e)\Delta\left(1 - \mathbb{P}_{Q_\Delta}(\widehat{a}_T = 2)\right) + O(\Delta^2) \right\}$$

$$= \sup_{\Delta \in (0, \infty)} \left\{ e\Delta\left(1 - \mathbb{P}_{P^{\sharp\sharp}}(\widehat{a}_T = 1) + \mathbb{P}_{P^{\sharp\sharp}}(\widehat{a}_T = 1) - \mathbb{P}_{Q_\Delta}(\widehat{a}_T = 1)\right) \right.$$

$$\left. + (1-e)\Delta\left(1 - \mathbb{P}_{P^{\sharp\sharp}}(\widehat{a}_T = 2) + \mathbb{P}_{P^{\sharp\sharp}}(\widehat{a}_T = 2) - \mathbb{P}_{Q_\Delta}(\widehat{a}_T = 2)\right) + O(\Delta^2) \right\}$$

$$= \sup_{\Delta \in (0, \infty)} \left\{ e\Delta\left(1 - \mathbb{P}_{P^{\sharp\sharp}}(\widehat{a}_T = 1) - \sqrt{\frac{\mathbb{E}_{P^{\sharp\sharp}}[L_T(P^{\sharp\sharp}, Q_\Delta)]}{2}}\right) \right.$$

$$\left. + (1-e)\Delta\left(1 - \mathbb{P}_{P^{\sharp\sharp}}(\widehat{a}_T = 2) - \sqrt{\frac{\mathbb{E}_{P^{\sharp\sharp}}[L_T(P^{\sharp\sharp}, Q_\Delta)]}{2}}\right) + O(\Delta^2) \right\}$$

$$= \sup_{\Delta \in (0, \infty)} \left\{ e\Delta\left(1 - \frac{1}{2} - \sqrt{\frac{\mathbb{E}_{P^{\sharp\sharp}}[L_T(P^{\sharp\sharp}, Q_\Delta)]}{2}}\right) + (1-e)\Delta\left(1 - \frac{1}{2} - \sqrt{\frac{\mathbb{E}_{P^{\sharp\sharp}}[L_T(P^{\sharp\sharp}, Q_\Delta)]}{2}}\right) + O(\Delta^2) \right\}$$

$$= \sup_{\Delta \in (0, \infty)} \left\{ \Delta\left(\frac{1}{2} - \sqrt{\frac{\mathbb{E}_{P^{\sharp\sharp}}[L_T(P^{\sharp\sharp}, Q_\Delta)]}{2}}\right) + O(\Delta^2) \right\}$$

$$\geq \inf_{w \in \mathcal{W}} \sup_{\Delta \in (0,\infty)} \left\{ \Delta \left( \frac{1}{2} - \sqrt{\frac{T\Delta^2}{2\mathbb{E}_P\left[\frac{(\sigma^1(X_t))^2}{w(1|X)} + \frac{(\sigma^2(X))^2}{w(2|X_t)}\right]} + O(T\Delta^3)} \right) + O(\Delta^2) \right\}.$$

Let $\Delta = \frac{1}{4}\sqrt{\frac{2\mathbb{E}_P\left[\frac{(\sigma^1(X))^2}{w(1|X)} + \frac{(\sigma^2(X))^2}{w(2|X)}\right]}{T}}$. Then,

$$\inf_{w \in \mathcal{W}} \sup_{\Delta \in (0,\infty)} \left\{ \Delta \left( \frac{1}{2} - \sqrt{\frac{T\Delta^2}{2\mathbb{E}_P\left[\frac{(\sigma^1(X_t))^2}{w(1|X)} + \frac{(\sigma^2(X))^2}{w(2|X_t)}\right]} + O(T\Delta^3)} \right) + O(\Delta^2) \right\}$$

$$\geq \frac{1}{12} \inf_{w \in \mathcal{W}} \sqrt{\frac{\mathbb{E}_P\left[\frac{(\sigma^1(X))^2}{w(1|X_t)} + \frac{(\sigma^2(X_t))^2}{w(2|X_t)}\right]}{T}} + O\left(\Delta^2\right)$$

$$\geq \frac{1}{12} \sqrt{\frac{\mathbb{E}_P\left[\left(\sigma^1(X) + \sigma^2(X)\right)^2\right]}{T}} + O\left(\frac{\mathbb{E}_P\left[\left(\sigma^1(X) + \sigma^2(X)\right)^2\right]}{T}\right).$$

Here, the minimizer regarding $w$ is $\widetilde{w}(1|x) = \frac{\sigma^1(x)}{\sigma^1(x)+\sigma^2(x)}$ ($\widetilde{w}(2|x) = 1 - \widetilde{w}(1|x)$) (van der Laan, 2008; Hahn et al., 2011; Kato et al., 2020). Because $\zeta_P(x) = \zeta(x)$, $\sup_{P' \in \mathcal{P}^*} \sqrt{\mathbb{E}_{P'}[r_T(P)(\pi)]} \geq \frac{1}{12}\sqrt{\int (\sigma^1(X) + \sigma^2(X))^2 \zeta(x)\mathrm{d}x} + o(1)$. $\qquad\square$

## G  PROOF OF THEOREM 5.2

Let us define $\Delta^{a,b}(P) = \mu^a(P) - \mu^b(P)$ and $\Delta^{a,b}(P)(x) = \mu^a(P)(x) - \mu^b(P)(x)$ for all $a, b \in ]K]$ and $x \in \mathcal{X}$.

For $a, b \in [K]$, define[6]

$$\xi_t^{a,b}(P) = \frac{\varphi_t^a(Y_t, A_t, X_t) - \varphi_t^b(Y_t, A_t, X_t) - \Delta^b(P)}{\sqrt{TV^{a,b*}(P)}},$$

$$\text{where } V^{a,b*}(P) = \mathbb{E}_P\left[\frac{(\sigma^a(X))^2}{w^*(a|X)} + \frac{(\sigma^b(X))^2}{w^*(b|X)} + \left(\Delta^{a,b}(P)(X) - \Delta^{a,b}(P)\right)^2\right].$$

We show that $\{\xi_t^{a,b}(P)\}_{t=1}^T$ is a martingale difference sequence. The proof is shown in Appendix G.1.

**Lemma G.1.** *Under the AS-AIPW strategy, $\mathbb{E}_P[\xi_t^{a,b}(P)|\mathcal{F}_{t-1}] = 0$ holds.*

Note that $\sum_{t=1}^T \xi_t^{a,b}(P) = \sqrt{T}\left(\widehat{\mu}_T^{\mathrm{AIPW},a} - \widehat{\mu}_T^{\mathrm{AIPW},b} - \Delta^{a,b}(P)\right)/\sqrt{V^{a,b}(P)}$. Theorem 5.2 can be derived from Proposition B.7 and Chernoff bound. To prove Theorem 5.2, we use the following lemmas.

**Lemma G.2.** *Fix $a, b \in [K]^2$. Suppose that $\xi_t^{a,b}(P)$ is conditionally sub-Gaussian; that is, there exits an absolute constant $C_\xi > 0$ such that for all $P \in \mathcal{P}$ and all $\lambda \in \mathbb{R}$,*

$$\mathbb{E}_P\left[\exp\left(\lambda\xi_t^{a,b}(P)\right)|\mathcal{F}_{t-1}\right] \leq \exp\left(\frac{\lambda^2 C_\xi}{2}\right).$$

*Also suppose that*

**(a)** $\sum_{t=1}^T \mathbb{E}[(\xi_t^{a,b}(P))^2] \to 1$;

**(b)** $\mathbb{E}[\sqrt{T}|\xi_t^{a,b}(P)|^r] < \infty$ *for some $r > 2$ and all $t \in \mathbb{N}$;*

---

[6]More rigorously, $\xi_t^{a,b}(P)$ and $\Omega_t^{a,b}(P)$ should be denoted as double arrays such as $\xi_{Tt}^{a,b}(P)$ and $\Omega_{Tt}^{a,b}(P)$ because they dependent on $T$. However, we omit the subscript $T$ for simplicity.

**(c)** $\sum_{t=1}^{T}(\xi_t^{a,b}(P))^2 \xrightarrow{p} 1$.

*Then, the following inequalities hold:*

$$
\begin{cases}
\mathbb{P}\left(\widehat{\mu}_T^{\mathrm{AIPW},a} - \widehat{\mu}_T^{\mathrm{AIPW},b} \leq 0\right) - \exp\left(-\dfrac{T\left(\Delta^{a,b}(P)\right)^2}{V^{a,b*}(P)}\right) \leq o(1) \quad \text{as } T \to \infty, \quad \text{if } E_0 < \sqrt{T}\Delta^{a,b}(P) \leq E \qquad \forall T \in \mathbb{N}; \\[3mm]
\mathbb{P}\left(\widehat{\mu}_T^{\mathrm{AIPW},a} - \widehat{\mu}_T^{\mathrm{AIPW},b} \leq 0\right) - \exp\left(-\dfrac{T\left(\Delta^{a,b}(P)\right)^2}{2C_\xi^2}\right) \leq 0 \quad \forall T \in \mathbb{N} \qquad\qquad \text{if } E_0 < \sqrt{T}\Delta^{a,b}(P),
\end{cases}
\tag{13}
$$

*where $E_0 > E > 0$ are some constants independent from $T$ and $\Delta^{a,b}(P)$.*

**Lemma G.3.** *Under Assumption 5.1, there exists constants $M, C_\xi > 0$ independent from $P \in \mathcal{P}^*$ such that for all $t \in \mathbb{N}$,*

$$
\mathbb{E}_P\left[\exp\left(\left(\sqrt{T}\xi_t^{a,b}(P)\right)^2\right)\right] < M;
$$

*and*

$$
\mathbb{E}_P\left[\exp\left(\lambda\sqrt{T}\xi_t^{a,b}(P)\right)|\mathcal{F}_{t-1}\right] \leq \exp\left(\frac{\lambda^2 C_\xi}{2}\right).
$$

**Lemma G.4.** *Under Assumption 5.1 and the AS-AIPW strategy, the following properties hold:*

**(a)** $\sum_{t=1}^{T}\mathbb{E}_P[(\xi_t^{a,b}(P))^2] \to 1$, *a positive value;*

**(b)** $\mathbb{E}_P[|\sqrt{T}\xi_t^{a,b}(P)|^r] < \infty$ *for some $r > 2$ and for all $t \in \mathbb{N}$;*

**(c)** $\sum_{t=1}^{T}(\xi_t^{a,b}(P))^2 \xrightarrow{p} 1$.

Proofs of these lemmas are shown in Appendixes G.2–G.4.

Then, we prove Theorem 5.2 as follows.

*Proof of Theorem 5.2.* From Lemma G.2, for some constants $0 < E_0 < E$, if $E_0 < \sqrt{T}\Delta^{a,c}(P) \leq E$, for any $\epsilon > 0$, there exists $T_0 > 0$ such that for all $T > T_0$, the expected simple regret is bounded as

$$
\mathbb{P}_P\left(\widehat{\mu}_T^{\mathrm{AIPW},a} \geq \widehat{\mu}_T^{\mathrm{AIPW},b}\right) \leq \exp\left(-\frac{T\left(\Delta^{a,b}(P)\right)^2}{V^{a,b}(P)}\right) + \epsilon.
$$

As well as the proof of Corollary 3 of Bubeck et al. (2011), we consider two cases where a given $\Delta^a$ is more or less than a threshold $\ell^1, \ell^2, \ldots, \ell^K > 0$. We have

$$
\mathbb{E}_P\left[r_T(P)\left(\pi^{\mathrm{AS\text{-}AIPW}}\right)\right] = \sum_{a \in [K]} \Delta^a \mathbb{P}_P\left(\widehat{\mu}_T^{\mathrm{AIPW},a^*(P)} \leq \widehat{\mu}_T^{\mathrm{AIPW},a}\right)
$$

$$
\leq \sum_{a \in [K]}\left\{\ell^a \mathbb{P}_P\left(\widehat{\mu}_T^{\mathrm{AIPW},a^*(P)} \leq \widehat{\mu}_T^{\mathrm{AIPW},a}\right) + \mathbb{1}[\Delta^a \geq \ell^a]\Delta^a \mathbb{P}_P\left(\widehat{\mu}_T^{\mathrm{AIPW},a^*(P)} \leq \widehat{\mu}_T^{\mathrm{AIPW},a}\right)\right\}
$$

$$
\leq \max_{a \in [K]} \ell^a + \sum_{a \in [K]}\left\{\mathbb{1}[\Delta^a \geq \ell^a]\Delta^a \mathbb{P}_P\left(\widehat{\mu}_T^{\mathrm{AIPW},a^*(P)} \leq \widehat{\mu}_T^{\mathrm{AIPW},a}\right)\right\}
$$

Because $x \in [0, C_\Delta] \mapsto z\exp(-Cz^2)$ is decreasing on $[1/\sqrt{2C}, C_\Delta]$, for any $C > 0$ and $C_\Delta$, where $\Delta^a < C_\Delta$ for all $a \in [K]$. Therefore, taking $C = \lfloor \frac{T}{2V^a(P)}\rfloor$, for $\ell^a \geq 1/\sqrt{2\left\lfloor\frac{T}{2V^a(P)}\right\rfloor}$,

$$
\mathbb{E}_P\left[r_T(P)\left(\pi^{\mathrm{AS\text{-}AIPW}}\right)\right]
$$

$$
\leq \max_{a \in [K]}\left\{\ell^a + \sum_{a \in [K]} \ell^a \left\{\exp\left(-\frac{T(\ell^a)^2}{2V^a(P)}\right) + \varepsilon\right\}\right\}
$$

$$\leq \max_{a\in[K]} \left\{ \ell^a + (K-1)\max_{a\in[K]} \ell^a \left\{ \exp\left(-\frac{T(\ell^a)^2}{2V^a(P)}\right) + \varepsilon \right\} \right\}$$

$$\leq \max_{a\in[K]} \left\{ \ell^a + (K-1)\ell^a \left\{ \exp\left(-\frac{T(\ell^a)^2}{2V^a(P)}\right) + \varepsilon \right\} \right\}.$$

Substituting $\ell^a = \sqrt{\log K / \lfloor \frac{T}{2V^a(P)} \rfloor}$, we have

$$\mathbb{E}_P\left[r_T\left(\pi^{\mathrm{HIR}}\right)(P)\right] \leq \max_{a\in[K]} \left\{ \sqrt{\log K / \left\lfloor \frac{T}{2V^a(P)} \right\rfloor} \right.$$

$$+ (K-1)\sqrt{\log K / \left\lfloor \frac{T}{2V^a(P)} \right\rfloor} \exp\left(-\frac{T\log K / \left\lfloor \frac{T}{2V^a(P)} \right\rfloor}{2V^a(P)}\right)$$

$$\left. \times \exp\left(\left\{ \frac{\sqrt{T\log K / \lfloor \frac{T}{2V^a(P)} \rfloor}}{\sqrt{V^a(P)}} + \frac{T\log K / \lfloor \frac{T}{2V^a(P)} \rfloor}{2V^a(P)} \right\} \varepsilon \right) \right\}.$$

Letting $T \to \infty$ and $\varepsilon \to 0$, we conclude the proof.

If $E_0 < \sqrt{T}\Delta^{a,b}(P)$, for all $T \in \mathbb{N}$, we obtain

$$\mathbb{P}_P\left(\widehat{\mu}_T^{\mathrm{AIPW},b} \geq \widehat{\mu}_T^{\mathrm{AIPW},a}\right) \leq \exp\left(-\frac{T\left(\Delta^{a,b}(P)\right)^2}{2C_\xi^2}\right).$$

Similarly, for the second inequality with $E_0 < \sqrt{T}\Delta^{a,b}(P)$, we obtain the maximizer as $\Delta^{a,b*} = \sqrt{\frac{C_\xi(P)}{2T}}$.

As an upper bound, we can use both $\Delta^{a,b*} = \sqrt{\frac{V^{a,b*}(P)}{2T}}$ and $\Delta^{a,b*} = \sqrt{\frac{C_\xi(P)}{2T}}$. In our analysis, we use $\Delta^{a,b*} = \sqrt{\frac{V^{a,b*}(P)}{2T}}$ to show the minimax optimality. Therefore, we use $\Delta^{a,b*} = \sqrt{\frac{V^{a,b}(P)}{2T}}$.

$\square$

## G.1 Proof of Lemma G.1

We have

$$\mathbb{E}_P\left[\frac{\mathbb{1}[A_t = a]\left(Y_t^a - \widehat{\mu}_t^a(X_t)\right)}{\widehat{w}_t(a|X_t)} + \widehat{\mu}_t^a(X_t)|X_t, \mathcal{F}_{t-1}\right]$$

$$= \frac{\mathbb{E}_P\left[\mathbb{1}[A_t = a]\left(Y_t^a - \widehat{\mu}_t^a(X_t)\right)|X_t, \mathcal{F}_{t-1}\right]}{\widehat{w}_t(a|X_t)} + \widehat{\mu}_t^a(X_t)$$

$$= \frac{\widehat{w}_t(a|X_t)\left(\mu^a(X_t)(P) - \widehat{\mu}_t^a(X_t)\right)}{\widehat{w}_t(a|X_t)} + \widehat{\mu}_t^a(X_t) = \mu^a(X_t)(P),$$

$$\mathbb{E}_P[\xi_t^{a,b}(P)|\mathcal{F}_{t-1}] = \mathbb{E}^X\left[\frac{\mu^a(x)(P) - \mu^b(x)(P) - \Delta^b(P)}{\sqrt{TV^{a,b*}(P)}}\right]\zeta(x)\mathrm{d}x = 0.$$

## G.2 Proof of Lemma G.2

We prove Lemma G.2. Lemma G.2 can be derived from Proposition B.6 and Chernoff bound.

*Proof.* The second statement directly holds from and large deviation bound. Therefore, we focus on the proof of the first statement, $\mathbb{P}\left(\widehat{\mu}_T^{\mathrm{AIPW},a} - \widehat{\mu}_T^{\mathrm{AIPW},c} \leq 0\right) - \exp\left(-\frac{T\left(\Delta^{a,b}(P)\right)^2}{V^{a,b*}(P)}\right) \leq o(1).$

This inequality follows from the martingale CLT of White (1984) (Proposition B.6) on $\sum_{t=1}^{T} \xi_t^{a,b}(P)$ because

$$\mathbb{P}\left(\widehat{\mu}_T^{\mathrm{AIPW},a} - \widehat{\mu}_T^{\mathrm{AIPW},b} \leq 0\right) = \mathbb{P}\left(\sqrt{T}\left(\widehat{\mu}_T^{\mathrm{AIPW},a} - \widehat{\mu}_T^{\mathrm{AIPW},b}\right) - \sqrt{T}\Delta^{a,b}(P) \leq -\sqrt{T}\Delta^{a,b}(P)\right)$$

$$= \mathbb{P}\left(\sqrt{\frac{T}{V^{a,b*}(P)}}\left(\widehat{\mu}_T^{\mathrm{AIPW},a} - \widehat{\mu}_T^{\mathrm{AIPW},b} - \Delta^{a,b}(P)\right) \leq -\sqrt{\frac{T}{V^{a,b*}(P)}}\Delta^{a,b}(P)\right)$$

$$= \mathbb{P}\left(\sum_{t=1}^{T} \xi_t^{a,b}(P) \leq -\sqrt{\frac{T}{V^{a,b*}(P)}}\Delta^{a,b}(P)\right).$$

Thus, we are interested in $= \mathbb{P}\left(\sum_{t=1}^{T} \xi_t^{a,b}(P) \leq -\sqrt{\frac{T}{V^{a,b*}(P)}}\Delta^{a,b}(P)\right)$ and show the bound by using the martingale CLT.

Under the following three conditions, we can apply the martingale CLT,

(a) $\sum_{t=1}^{T} \mathbb{E}_P[(\xi_t^{a,b}(P))^2] \to 1$, a positive value;

(b) $\mathbb{E}_P[|\sqrt{T}\xi_t^{a,b}(P)|^r] < \infty$ for some $r > 2$ and for all $t \in \mathbb{N}$;

(c) $\sum_{t=1}^{T}(\xi_t^{a,b}(P))^2 \xrightarrow{P} 1$.

By using the martingale CLT, as $T \to \infty$,

$$\sum_{t=1}^{T} \xi_t^{a,b}(P) \xrightarrow{d} \mathcal{N}(0,1).$$

This result implies that for each $-\infty < x < \infty$, as $T \to \infty$,

$$\left|\mathbb{P}\left(\sum_{t=1}^{T} \xi_t^{a,b}(P) \leq x\right) - \Phi(x)\right| = \left|\mathbb{P}\left(\sqrt{T}\frac{\widehat{\mu}_T^{\mathrm{AIPW},a} - \widehat{\mu}_T^{\mathrm{AIPW},b} - \Delta^{a,b}(P)}{V^{a,b*}(P)} \leq x\right) - \Phi(x)\right| = o(1).$$

From the martingale CLT, if there exists a constant $E > 0$ such that $\sqrt{T}\Delta^{a,b}(P) < E$,

$$\mathbb{P}\left(\sqrt{T}\frac{\widehat{\mu}_T^{\mathrm{AIPW},a} - \widehat{\mu}_T^{\mathrm{AIPW},b} - \Delta^{a,b}(P)}{V^{a,b*}(P)} \leq -\sqrt{\frac{T}{V^{a,b*}(P)}}\right) - \Phi\left(-\sqrt{\frac{T}{V^{a,b*}(P)}}\Delta^{a,b}(P)\right) + o(1).$$

Let $\phi(\cdot)$ be the density function of the standard normal distribution. Then,

$$\Phi\left(-\sqrt{\frac{T}{V^{a,b*}(P)}}\Delta^{a,b}(P)\right) = \int_{-\infty}^{-\sqrt{\frac{T}{V^{a,b*}(P)}}\Delta^{a,b}(P)} \phi(u)\mathrm{d}u$$

$$\leq \phi\left(-\sqrt{\frac{T}{V^{a,b*}(P)}}\Delta^{a,b}(P)\right) / \sqrt{\frac{T}{V^{a,b*}(P)}}\Delta^{a,b}(P)$$

$$\leq \frac{e}{(e-1)\sqrt{2\pi}} \cdot \frac{\exp\left(-\frac{T(\Delta^{a,b}(P))^2}{2V^{a,b*}(P)}\right)}{\sqrt{\frac{T}{V^{a,b*}(P)}}\Delta^{a,b}(P)}.$$

Here, we used $\frac{e}{(e-1)\sqrt{2\pi}} \approx 0.252$, $\sqrt{\frac{T}{V^{a,b*}(P)}}\Delta^{a,b}(P) > 1$ for large $T$, $\phi(u + 1/u) = \frac{1}{\sqrt{2\pi}}\exp(-(u + 1/u)^2/2) = e^{-1}\phi(u)\exp(-1/(2u^2)) = e^{-1}\phi(u)$ and

$$\int_{-\infty}^{-x} \phi(u)\mathrm{d}u = \int_{x}^{\infty} \phi(u)\mathrm{d}u \leq \sum_{k=0}^{\infty} \frac{1}{x}\phi(x + k/x) \leq \frac{\phi(x)}{x}\sum_{k=0}^{\infty} \exp(-k) = \frac{e}{e-1} \cdot \frac{\phi(x)}{x},$$

Therefore, if $0 < \sqrt{T}\Delta^{a,b}(P) \leq C$,

$$\mathbb{P}\left(\widehat{\mu}_T^{\mathrm{AIPW},a} - \widehat{\mu}_T^{\mathrm{AIPW},b} \leq 0\right) - \exp\left(-T\frac{(\Delta^{a,b}(P))^2}{2V^{a,b}(P,w^*)}\right) \leq o(1).$$

Thus. we proved the first inequality. $\qquad \square$

### G.3 PROOF OF LEMMA G.3

*Proof.* We show the following inequality: there exists a constant $M > 0$ such that for any $P \in \mathcal{P}^*$ and $\mathcal{F}_{t-1} = \sigma(X_1, A_1, Y_1, \ldots, X_t, A_t, Y_t)$,

$$\mathbb{E}_P \left[ \exp\left( \left( \sqrt{T} \xi_t^{a,b}(P) \right)^2 \right) | \mathcal{F}_{t-1} \right] < M. \tag{14}$$

From Proposition 2.5.2 (iv) of Vershynin (2018), this inequality implies that $\xi_t^{a,b}(P)$ is conditionally sub-Gaussian.

If the above inequality Eq. (14) holds, the first statement $\mathbb{E}_P \left[ \exp\left( \left( \sqrt{T} \xi_t^{a,b}(P) \right)^2 \right) \right] < M$ holds directly. In addition, because $\xi_t^{a,b}(P)$ is conditionally sub-Gaussian and $\mathbb{E}[\xi_t^{a,b}(P)|\mathcal{F}_{t-1}] = 0$, from Proposition 2.5.2 (v) of Vershynin (2018), there exists a constant $C_\xi > 0$ such that for any $P \in \mathcal{P}^*$ and $\mathcal{F}_{t-1} \in \mathbb{R} \times [K] \times \mathcal{X} \times \cdots \mathbb{R} \times [K] \times \mathcal{X}$, $\mathbb{E}_P \left[ \exp\left( \lambda \sqrt{T} \xi_t^{a,b}(P) \right) | \mathcal{F}_{t-1} \right] \leq \exp\left( \frac{\lambda^2 C_\xi}{2} \right)$. Thus, the two statement hold from Eq. (14). Therefore, we consider showing Eq. (14).

Recall that $\varphi_t^a \left( Y_t, A_t, X_t \right)$ is constructed as

$$\varphi_t^a \left( Y_t, A_t, X_t \right) = \frac{\mathbb{1}[A_t = a]\left( Y_t^a - \widehat{\mu}_t^a(X_t) \right)}{\widehat{w}_t(a|X_t)} + \widehat{\mu}_t^a(X_t).$$

For all $t = 1, 2, \ldots,$

$$\mathbb{E}_P \left[ \exp\left( \left( \sqrt{T} \xi_t^{a,b}(P) \right)^2 \right) | \mathcal{F}_{t-1} \right]$$

$$= \mathbb{E}_P \left[ \exp\left( \frac{\left( \varphi_t^a \left( Y_t, A_t, X_t \right) - \varphi_t^b \left( Y_t, A_t, X_t \right) - \Delta^{a,b}(P) \right)^2}{V^{a,b*}(P)} \right) \Big| \mathcal{F}_{t-1} \right].$$

Here, we have

$$\left( \varphi_t^a \left( Y_t, A_t, X_t \right) - \varphi_t^b \left( Y_t, A_t, X_t \right) - \Delta^{a,b}(P) \right)^2$$

$$= \left( \frac{\mathbb{1}[A_t = a]\left( Y_t^a - \widehat{\mu}_t^a(X_t) \right)}{\widehat{w}_t(a|X_t)} - \frac{\mathbb{1}[A_t = b]\left( Y_t^b - \widehat{\mu}_t^b(X_t) \right)}{\widehat{w}_t(b|X_t)} + \widehat{\mu}_t^a(X_t) - \widehat{\mu}_t^b(X_t) - \Delta^{a,b}(P) \right)^2$$

$$= \left( \frac{\mathbb{1}[A_t = a]\left( Y_t^a - \mu^a(P)(X_t) \right)}{\widehat{w}_t(a|X_t)} - \frac{\mathbb{1}[A_t = b]\left( Y_t^b - \mu^b(P)(X_t) \right)}{\widehat{w}_t(b|X_t)} \right.$$

$$\left. + \frac{\mathbb{1}[A_t = a]\left( \mu^a(P)(X_t) - \widehat{\mu}_t^a(X_t) \right)}{\widehat{w}_t(a|X_t)} - \frac{\mathbb{1}[A_t = b]\left( \mu^b(P)(X_t) - \widehat{\mu}_t^b(X_t) \right)}{\widehat{w}_t(b|X_t)} \right.$$

$$\left. + \widehat{\mu}_t^a(X_t) - \widehat{\mu}_t^b(X_t) - \Delta^{a,b}(P) \right)^2$$

$$= \frac{\mathbb{1}[A_t = a]\left( Y_t^a - \mu^a(P)(X_t) \right)^2}{\widehat{w}_t^2(a|X_t)} + \frac{\mathbb{1}[A_t = b]\left( Y_t^b - \mu^b(P)(X_t) \right)^2}{\widehat{w}_t^2(b|X_t)}$$

$$+ \frac{\mathbb{1}[A_t = a]\left( \mu^a(P)(X_t) - \widehat{\mu}_t^a(X_t) \right)^2}{\widehat{w}_t^2(a|X_t)} + \frac{\mathbb{1}[A_t = b]\left( \mu^b(P)(X_t) - \widehat{\mu}_t^b(X_t) \right)^2}{\widehat{w}_t^2(b|X_t)}$$

$$+ \left( \widehat{\mu}_t^a(X_t) - \widehat{\mu}_t^b(X_t) - \Delta^{a,b}(P) \right)^2$$

$$+ 2 \left( \frac{\mathbb{1}[A_t = a]\left( Y_t^a - \mu^a(P)(X_t) \right)\left( \mu^a(P)(X_t) - \widehat{\mu}_t^a(X_t) \right)}{\widehat{w}_t^2(a|X_t)} - \frac{\mathbb{1}[A_t = b]\left( Y_t^b - \mu^b(P)(X_t) \right)\left( \mu^b(P)(X_t) - \widehat{\mu}_t^b(X_t) \right)}{\widehat{w}_t^2(b|X_t)} \right)$$

$$+ 2 \left( \frac{\mathbb{1}[A_t = a]\left( \mu^a(P)(X_t) - \widehat{\mu}_t^a(X_t) \right)}{\widehat{w}_t(a|X_t)} - \frac{\mathbb{1}[A_t = b]\left( \mu^b(P)(X_t) - \widehat{\mu}_t^b(X_t) \right)}{\widehat{w}_t(b|X_t)} \right) \left( \widehat{\mu}_t^a(X_t) - \widehat{\mu}_t^b(X_t) - \Delta^{a,b}(P) \right)$$

$$+ 2 \left( \widehat{\mu}_t^a(X_t) - \widehat{\mu}_t^b(X_t) - \Delta^{a,b}(P) \right) \left( \frac{\mathbb{1}[A_t = a]\left(Y_t^a - \mu^a(P)(X_t)\right)}{\widehat{w}_t(a|X_t)} - \frac{\mathbb{1}[A_t = b]\left(Y_t^b - \mu^b(P)(X_t)\right)}{\widehat{w}_t(b|X_t)} \right).$$

where we used $\mathbb{1}[A = a]\mathbb{1}[A = b] = 0$.

We can show Eq. (14) by using the properties of sub-Gaussian random variables and boundedness of parameters and estimators. Since $Y_t^a$ is a sub-Gaussian random variable, there exists some universal constant $C_Y, C_Y' > 0$ such that for all $P \in \mathcal{P}$, $c \in \{a, b\}$, $x \in \mathcal{X}$, and $\lambda \in \mathbb{R}$, $\mathbb{E}[\exp(\lambda(Y_t^c - \mu^c(P)(x)))] \le \exp(C_Y^2\lambda^2/2)$ and $\mathbb{E}_P[\exp\left((Y_t^c - \mu^c(P)(x))^2\right)] \le \exp(C_Y'^2)$ (Proposition 2.7.1, Vershynin, 2018). In addition, from Definition 3.2 and Assumption 5.1, there exist constants $\overline{C}$, $C_{\widehat{\mu}}$ and $C_{\widehat{\nu}}$ such that for all $x \in \mathcal{X}$, and $c \in \{a, b\}$, $|\mu^c(P)(x)| \le C_\mu$, $|\widehat{\mu}_t^c(x)| \le C_{\widehat{\mu}}$, and $|\widehat{\nu}_t^c(x)| \le C_{\widehat{\nu}}$. Furthermore, because $\widehat{w}_t(a|X_t)$ is constructed by $\widehat{\mu}_t^c(x)$ and $\widehat{\nu}_t^c(x)$, there exists a constant $C_{\widehat{w}}$ such that for all $P \in \mathcal{P}$, $x \in \mathcal{X}$, $|1/\widehat{w}_t(a|x), \widehat{w}_t(a|x)| \ge C_{\widehat{w}}$. Therefore, there exists a constant $M > 0$ such that for any $P \in \mathcal{P}^*$ and $\mathcal{F}_{t-1} \in \mathbb{R} \times [K] \times \mathcal{X} \times \cdots \mathbb{R} \times [K] \times \mathcal{X}$,

$\mathbb{E}_P\left[\exp\left(\left(\sqrt{T}\xi_t^{a,b}(P)\right)^2\right)|\mathcal{F}_{t-1}\right] < M$. This proof is complete. $\qquad\square$

### G.4  PROOF OF LEMMA G.4

We prove Lemma G.4. Our proof is inspired by Kato et al. (2020).

*Proof.* Recall that

$$\varphi_t^a(Y_t, A_t, X_t) = \frac{\mathbb{1}[A_t = a]\left(Y_t^a - \widehat{\mu}_t^a(X_t)\right)}{\widehat{w}_t(a|X_t)} + \widehat{\mu}_t^a(X_t)$$

$$\xi_t^{a,b}(P) = \frac{\varphi_t^a\left(Y_t, A_t, X_t\right) - \varphi^{a,b}\left(Y_t, A_t, X_t\right) - \Delta^b(P)}{\sqrt{TV^{a,b*}(P)}},$$

$$\Omega_t^{a,b}(P) = \sum_{s=1}^t \mathbb{E}\left[\left(\xi_s^{a,b}(P)\right)^2|\mathcal{F}_{s-1}\right].$$

**Step 1: check of condition (a).**  Because $\sqrt{TV^{a,b*}(P)}$ is non-random variable, we consider the conditional expectation of $\varphi_t^a\left(Y_t, A_t, X_t\right) - \varphi^{a,b}\left(Y_t, A_t, X_t\right) - \Delta^{a,b}(P)$.

Instead of $\sum_{t=1}^T \mathbb{E}_P[(\xi_t^{a,b}(P))^2]$, we first consider the convergence of $\sum_{t=1}^T \mathbb{E}_P[(\xi_t^{a,b}(P))^2|\mathcal{F}_{t-1}] = \Omega_t^{a,b}(P)$; that is, we show $\Omega_t^{a,b}(P) - 1 \xrightarrow{P} 0$. Then, by using the $L^r$-convergence theorem (Proposition B.3), we show $\sum_{t=1}^T \mathbb{E}_P[(\xi_t^{a,b}(P))^2] - 1 \to 0$.

The conditional expectation is computed as follows:

$$\mathbb{E}_P\left[\left(\varphi_t^a\left(Y_t, A_t, X_t\right) - \varphi_t^b\left(Y_t, A_t, X_t\right) - \Delta^{a,b}(P)\right)^2\Big|\mathcal{F}_{t-1}\right]$$

$$= \mathbb{E}_P\left[\left(\frac{\mathbb{1}[A_t = a]\left(Y_t^a - \widehat{\mu}_t^a(X_t)\right)}{\widehat{w}_t(a|X_t)} - \frac{\mathbb{1}[A_t = b]\left(Y_t^b - \widehat{\mu}_t^b(X_t)\right)}{\widehat{w}_t(b|X_t)} + \widehat{\mu}_t^a(X_t) - \widehat{\mu}_t^b(X_t) - \Delta^{a,b}(P)\right)^2\Big|\mathcal{F}_{t-1}\right]$$

$$= \mathbb{E}_P\left[\left(\frac{\mathbb{1}[A_t = a]\left(Y_t^a - \widehat{\mu}_t^a(X_t)\right)}{\widehat{w}_t(a|X_t)} - \frac{\mathbb{1}[A_t = b]\left(Y_t^b - \widehat{\mu}_t^b(X_t)\right)}{\widehat{w}_t(b|X_t)}\right)^2\right.$$

$$+ 2\left(\frac{\mathbb{1}[A_t = a]\left(Y_t^a - \widehat{\mu}_t^a(X_t)\right)}{\widehat{w}_t(a|X_t)} - \frac{\mathbb{1}[A_t = b]\left(Y_t^b - \widehat{\mu}_t^b(X_t)\right)}{\widehat{w}_t(b|X_t)}\right)\left(\widehat{\mu}_t^a(X_t) - \widehat{\mu}_t^b(X_t) - \Delta^{a,b}(P)\right)$$

$$+ \left.\left(\widehat{\mu}_t^a(X_t) - \widehat{\mu}_t^b(X_t) - \Delta^{a,b}(P)\right)^2|\mathcal{F}_{t-1}\right]$$

$$= \mathbb{E}_P\left[\frac{\mathbb{1}[A_t = a]\left(Y_t^a - \widehat{\mu}_t^a(X_t)\right)^2}{\widehat{w}_t(a|X_t)} + \frac{\mathbb{1}[A_t = b]\left(Y_t^b - \widehat{\mu}_t^b(X_t)\right)^2}{\widehat{w}_t(b|X_t)}\right.$$

$$+ 2\left(\frac{\mathbb{1}[A_t = a]\big(Y_t^a - \widehat{\mu}_t^a(X_t)\big)}{\widehat{w}_t(a|X_t)} - \frac{\mathbb{1}[A_t = b]\big(Y_t^b - \widehat{\mu}_t^b(X_t)\big)}{\widehat{w}_t(b|X_t)}\right)\big(\widehat{\mu}_t^a(X_t) - \widehat{\mu}_t^b(X_t) - \Delta^{a,b}(P)\big)$$

$$+ \big(\widehat{\mu}_t^a(X_t) - \widehat{\mu}_t^b(X_t) - \Delta^{a,b}(P)\big)^2 \Big| \mathcal{F}_{t-1}\Big]$$

$$= \mathbb{E}_P\left[\frac{\big(Y_t^a - \widehat{\mu}_t^a(X_t)\big)^2}{\widehat{w}_t(a|X_t)}\Big|\mathcal{F}_{t-1}\right] + \mathbb{E}_P\left[\frac{\big(Y_t^b - \widehat{\mu}_t^b(X_t)\big)^2}{\widehat{w}_t(b|X_t)}\Big|\mathcal{F}_{t-1}\right]$$

$$- \mathbb{E}_P\left[\big(\widehat{\mu}_t^a(X_t) + \widehat{\mu}_t^b(X_t) - \Delta^{a,b}(P)\big)^2 \Big| \mathcal{F}_{t-1}\right]. \tag{15}$$

Here, we used

$$\mathbb{E}_P\left[\frac{\mathbb{1}[A_t = a]\big(Y_t^a - \widehat{\mu}_t^a(X_t)\big)^2}{(\widehat{w}_t(a|X_t))^2}\Big|\mathcal{F}_{t-1}\right] = \mathbb{E}_P\left[\mathbb{E}_P\left[\frac{\widehat{w}_t(a|X_t)\big(Y_t^a - \widehat{\mu}_t^a(X_t)\big)^2}{(\widehat{w}_t(a|X_t))^2}\Big|X_t\mathcal{F}_{t-1}\right]\right]$$

$$= \mathbb{E}_P\left[\frac{\big(Y_t^a - \widehat{\mu}_t^a(X_t)\big)^2}{\widehat{w}_t(a|X_t)}\Big|\mathcal{F}_{t-1}\right]$$

and

$$\mathbb{E}_P\left[\frac{\mathbb{1}[A_t = a]\big(Y_t^a - \widehat{\mu}_t^a(X_t)\big)}{\widehat{w}_t(a|X_t)}\big(\widehat{\mu}_t^a(X_t) - \widehat{\mu}_t^b(X_t) - (\mu^a(P) - \mu^b(P))\big)\Big|\mathcal{F}_{t-1}\right]$$

$$= \mathbb{E}_P\left[\big(\widehat{\mu}_t^a(X_t) - \widehat{\mu}_t^b(X_t) - (\mu^a(P) - \mu^b(P))\big)\mathbb{E}_P\left[\frac{\widehat{w}_t(a|X_t)\big(Y_t^a - \widehat{\mu}_t^a(X_t)\big)}{\widehat{w}_t(a|X_t)}\Big|X_t, \mathcal{F}_{t-1}\right]\mathcal{F}_{t-1}\right].$$

We also have

$$\mathbb{E}_P\left[\frac{\big(Y_t^a - \widehat{\mu}_t^a(X_t)\big)^2}{\widehat{w}_t(a|X_t)}\Big|X_t, \mathcal{F}_{t-1}\right] = \frac{\mathbb{E}_P[(Y_t^a)^2|X_t] - 2\mu^a(P)(X_t)\widehat{\mu}_t^a(X_t) + (\widehat{\mu}_t^a(X_t))^2}{\widehat{w}_t(a|X_t)}$$

$$= \frac{\mathbb{E}_P[(Y_t^a)^2|X_t] - (\mu^a(P)(X_t))^2 + (\mu^a(P)(X_t) - \widehat{\mu}_t^a(X_t))^2}{\widehat{w}_t(a|X_t)}.$$

Then,

$$\mathbb{E}_P\left[\frac{\big(Y_t^a - \widehat{\mu}_t^a(X_t)\big)^2}{\widehat{w}_t(a|X_t)}\Big|\mathcal{F}_{t-1}\right] + \mathbb{E}_P\left[\frac{\big(Y_t^b - \widehat{\mu}_t^b(X_t)\big)^2}{\widehat{w}_t(b|X_t)}\Big|\mathcal{F}_{t-1}\right]$$

$$- \mathbb{E}_P\left[\big(\widehat{\mu}_t^a(X_t) + \widehat{\mu}_t^b(X_t) - (\mu^a(P) - \mu^b(P))\big)^2 \Big|\mathcal{F}_{t-1}\right]$$

$$= \mathbb{E}_P\left[\frac{\mathbb{E}_P[(Y_t^a)^2|X_t] - (\mu^a(P)(X_t))^2 + (\mu^a(P)(X_t) - \widehat{\mu}_t^a(X_t))^2}{\widehat{w}_t(a|X_t)}\right]$$

$$+ \mathbb{E}_P\left[\frac{\mathbb{E}_P[(Y_t^b)^2|X_t] - (\mu_0^b(X_t))^2 + (\mu_0^b(X_t) - \widehat{\mu}_t^b(X_t))^2}{\widehat{w}_t(b|X_t)}\right]$$

$$- \mathbb{E}_P\left[\big(\widehat{\mu}_t^a(X_t) + \widehat{\mu}_t^b(X_t) - \Delta^{a,b}(P)\big)^2\right].$$

From $\widehat{\mu}_t^a(x) \xrightarrow{\mathrm{P}} \mu^a(P)(x)$ and $\widehat{w}_t(a|x) \xrightarrow{\mathrm{P}} w^*(a|x)$, for each $P \in \mathcal{P}$, $a \in [K]$, and $x \in \mathcal{X}$,

$$\left|\left(\frac{\mathbb{E}_P[(Y_t^a)^2|x] - (\mu^a(P)(x))^2 + (\mu^a(P)(x) - \widehat{\mu}_t^a(x))^2}{\widehat{w}_t(a|x)}\right)\right.$$

$$+ \left(\frac{\mathbb{E}_P[(Y_t^b)^2|x] - (\mu^b(P)(x))^2 + (\mu^b(P)(x) - \widehat{\mu}_t^b(x))^2}{\widehat{w}_t(b|x)}\right)$$

$$- \big(\widehat{\mu}_t^a(x) + \widehat{\mu}_t^b(x) - (\mu^a(P) - \mu^b(P))\big)^2$$

$$\left.- \left(\frac{(\sigma^a(x))^2}{w^*(a|X)} + \frac{(\sigma^b(X))^2}{w^*(b|X)} + \big(\mu^a(P)(x) - \mu^b(P)(x) - (\mu^a(P) - \mu^b(P))\big)^2\right)\right|$$

$$\leq \left| \frac{\mathbb{E}_P[(Y_t^a)^2|x] - (\mu^a(P)(x))^2}{\widehat{w}_t(a|x)} - \frac{(\sigma^a(x))^2}{w^*(a|x)} \right| + \left| \frac{\mathbb{E}_P[(Y_t^a)^2|x] - (\mu^a(P)(x))^2}{\widehat{w}_t(a|x)} - \frac{(\sigma^b(X))^2}{w^*(b|x)} \right|$$

$$+ \frac{(\mu^a(P)(x) - \widehat{\mu}_t^b(x))^2}{\widehat{w}_t(a|X_t)} + \frac{(\mu_0^b(x) - \widehat{\mu}_t^b(x))^2}{\widehat{w}_t(b|X_t)}$$

$$+ \left| \left(\widehat{\mu}_t^a(x) - \widehat{\mu}_t^b(x) - \Delta^{a,b}(P)\right)^2 - \left(\mu^a(P)(x) - \mu^b(P)(x) - \Delta^{a,b}(P)\right)^2 \right|$$

$$\xrightarrow{\mathrm{P}} 0.$$

Note that $\mathbb{E}_P[(Y_t^a)^2|x] - (\mu^a(P)(x))^2 = (\sigma^a(x))^2$. This directly implies that

$$\frac{1}{T}\sum_{t=1}^{T}\mathbb{E}_P\left[\left(\varphi_t^a\Big(Y_t, A_t, X_t\Big) - \varphi_t^b\Big(Y_t, A_t, X_t\Big) - (\mu^a(P) - \mu^b(P))\right)^2 \Big| \mathcal{F}_{t-1}\right] - V^{a,b*}(P) \xrightarrow{\mathrm{P}} 0,$$

$$\Leftrightarrow \frac{1}{TV^{a,b*}(P)}\sum_{t=1}^{T}\mathbb{E}_P\left[\left(\varphi_t^a\Big(Y_t, A_t, X_t\Big) - \varphi_t^b\Big(Y_t, A_t, X_t\Big) - (\mu^a(P) - \mu^b(P))\right)^2 \Big| \mathcal{F}_{t-1}\right] - 1 \xrightarrow{\mathrm{P}} 0.$$

Thus, we showed $\Omega_t^{a,b}(P) - 1 \xrightarrow{\mathrm{P}} 0$.

To apply $L^r$-convergence theorem (Proposition B.3), we check that $\Omega_t^{a,b}(P) - 1$ is uniformly integrable. Here, recall that $\xi_t^{a,b}$ is conditionally sub-Gaussian as shown in Lemma G.3. From Lemma 2.7.6 of Vershynin (2018), the squared value $(\xi_t^{a,b})^2$ is conditionally sub-exponential. Therefore, $\Omega_t^{a,b}(P)$ is a sum of the sub-exponential random variable. This implies that $\Omega_t^{a,b}(P) - 1$ is uniformly integrable. As a result, from $L^r$-convergence theorem (Proposition B.3), $\mathbb{E}_P[\Omega_t^{a,b}(P)] - 1 = \sum_{t=1}^{T}\mathbb{E}_P[(\xi_t^{a,b}(P))^2] - 1 \to 0$ holds.

**Step 2: check of condition (b).** We showed that $\xi_t^{a,b}$ is sub-Gaussian in Lemma G.3. When $\xi_t^{a,b}$ is sub-Gaussian, the condition holds from Proposition 2.5.2 (ii) of Vershynin (2018).

**Step 3: check of condition (c).** Let $u_t$ be an MDS such that

$$u_t = (\xi_t^{a,b}(P))^2 - \mathbb{E}\big[(\xi_t^{a,b}(P))^2 \mid \mathcal{F}_{t-1}\big]$$

$$= \left( \frac{\mathbb{1}[A_t = a]\big(Y_t - \widehat{\mu}_t^a(X_t)\big)}{\widehat{w}_t(a|X_t)} - \frac{\mathbb{1}[A_t = b]\big(Y_t - \widehat{\mu}_t^b(X_t)\big)}{\widehat{w}_t(b|X_t)} + \widehat{\mu}_t^a(X_t) - \widehat{\mu}_t^b(X_t) - \Delta^{a,b}(P) \right)^2$$

$$- \mathbb{E}\left[ \left( \frac{\mathbb{1}[A_t = a]\big(Y_t - \widehat{\mu}_t^a(X_t)\big)}{\widehat{w}_t(a|X_t)} - \frac{\mathbb{1}[A_t = b]\big(Y_t - \widehat{\mu}_t^b(X_t)\big)}{\widehat{w}_t(b|X_t)} + \widehat{\mu}_t^a(X_t) - \widehat{\mu}_t^b(X_t) - \Delta^{a,b}(P) \right)^2 \Big| \mathcal{F}_{t-1}\right].$$

From the boundedness of each variable in $z_t$, we can apply weak law of large numbers for an MDS (Proposition B.5 in Appendix B), and obtain

$$\sum_{t=1}^{T}u_t = \sum_{t=1}^{T}\left((\xi_t^{a,b}(P))^2 - \mathbb{E}\big[(\xi_t^{a,b}(P))^2|\mathcal{F}_{t-1}\big]\right) \xrightarrow{\mathrm{P}} 0.$$

In Step 1, we showed

$$\sum_{t=1}^{T}\mathbb{E}\big[(\xi_t^{a,b}(P))^2|\mathcal{F}_{t-1}\big] - 1 \xrightarrow{\mathrm{P}} 0.$$

As a conclusion, we obtain

$$\sum_{t=1}^{T}(\xi_t^{a,b}(P))^2 - 1 = \sum_{t=1}^{T}\left(z_t^2 - \mathbb{E}\left[(\xi_t^{a,b}(P))^2|\mathcal{F}_{t-1}\right] + \mathbb{E}\big[(\xi_t^{a,b}(P))^2|\mathcal{F}_{t-1}\big] - 1\right) \xrightarrow{\mathrm{P}} 0.$$

$\square$

# H   NON-ASYMPTOTIC UPPER BOUND

The order of the expected simple regret is determined by the convergence rate of $\Omega_t^{a,b}(P) - 1$. When a specific convergence rate is assumed, a non-asymptotic upper bound is given as follows.

**Corollary H.1** (Worst-case upper bound). *Suppose that for some $\alpha > 0$ and constants $C$ and $D$,*

$$\mathbb{P}_P\left(|\Omega_t^{a,b}(P) - 1| > D/\sqrt{t}(\log t)^{2+2/\alpha}\right) \leq Ct^{-1/4}(\log t)^{1+1/\alpha}.$$

*Then, under the AS-AIPW strategy, when $K \geq 3$,*

$$\sup_{P \in \mathcal{P}^*} \mathbb{E}_P\left[r_T(P)\left(\pi^{\text{AS-AIPW}}\right)\right]$$

$$\leq \max_{b \in [K] \setminus \mathcal{A}^*(P)} \sqrt{\log(K)\mathbb{E}^X\left[\sum_{b \in [K]} (\sigma^b(X))^2\right]/T} + AT^{-1/4}(\log T)^{1+1/\alpha};$$

*when $K = 2$,*

$$\sup_{P \in \mathcal{P}^*} \mathbb{E}_P\left[r_T(P)\left(\pi^{\text{AS-AIPW}}\right)\right]$$

$$\leq max_{b \in [K] \setminus \mathcal{A}^*(P)} \sqrt{\log(K)\mathbb{E}^X\left[(\sigma^1(X) + \sigma^2(X))^2\right]/T} + AT^{-1/4}(\log T)^{1+1/\alpha}.$$

The convergence rate of $\Omega_t^{a,b}(P) - 1$ determines the non-asymptotic expected simple regret via the martingale CLT. It is also known that the rate of the martingale CLT is no better than the convergent rate of $\mathbb{E}\left[\Omega_t^{a,b}(P) - 1\right]^{1/2}$ (Hall et al., 1980).

Besides, to show the asymptotic minimax optimality for the lower bound, we do not have to derive the non-asymptotic tight upper bound using $\mathbb{P}_P\left(|\Omega_t^{a,b}(P) - 1| > D/\sqrt{t}(\log t)^{2+2/\alpha}\right) \leq Ct^{-1/4}(\log t)^{1+1/\alpha}$.

To prove this corollary, we use the following corollary in which we replace the martingale CLT in Lemma G.2 with the non-asymptotic representation (the rate of the martingale CLT in Proposition B.7).

**Corollary H.2.** *For all $a, b \in [K]^2$, suppose that $\xi_t^{a,b}(P)$ is conditionally sub-Gaussian; that is, there exits an absolute constant $C_\xi > 0$ such that for all $P \in \mathcal{P}$ and all $\lambda \in \mathbb{R}$,*

$$\mathbb{E}_P\left[\exp\left(\lambda \xi_t^{a,b}(P)\right)|\mathcal{F}_{t-1}\right] \leq \exp\left(\frac{\lambda^2 C_\xi}{2}\right).$$

*Also suppose that some $\alpha > 0$ and constants $M$, $C$ and $D$,*

$$\max_{t \in \mathbb{N}} \mathbb{E}_P\left[\exp\left(\left|\sqrt{T}\xi_t^{a,b}(P)\right|^\alpha\right)\right] < M,$$

*and*

$$\mathbb{P}\left(|\Omega_t^{a,b}(P) - 1| > D/\sqrt{t}(\log t)^{2+2/\alpha}\right) \leq Ct^{-1/4}(\log t)^{1+1/\alpha}.$$

*Then, for $a, b \in [K]$ and $T \geq 2$,*

$$\mathbb{P}\left(\widehat{\mu}_T^{\text{AIPW},a} - \widehat{\mu}_T^{\text{AIPW},b} \leq 0\right) \leq \begin{cases} \exp\left(-\frac{T(\Delta^{a,b}(P))^2}{V^{a,b*}(P)}\right) + AT^{-1/4}(\log T)^{1+1/\alpha} & \text{if } E_0 < \sqrt{T}\Delta^{a,b}(P) \leq E; \\ \exp\left(-\frac{T(\Delta^{a,b}(P))^2}{2C_\xi^2}\right) & \text{if } E_0 < \sqrt{T}\Delta^{a,b}(P), \end{cases}$$

$$\tag{16}$$

*where the constant $A$ depends only on $\alpha$, $M$, $C$, and $D$, and $E_0 > E > 0$ are some constants independent from $T$ and $\Delta^{a,b}(P)$.*

Then, we prove Corollary H.1 as follows.

*Proof of Corollary H.1.* From Corollary H.2 with Lemma G.3, the probability of misidentification $\mathbb{P}_P\left(\widehat{\mu}_T^{\mathrm{AIPW},c} \geq \widehat{\mu}_T^{\mathrm{AIPW},a}\right)$ is bounded as follows:

$$\mathbb{P}_P\left(\widehat{\mu}_T^{\mathrm{AIPW},c} \geq \widehat{\mu}_T^{\mathrm{AIPW},a}\right)$$
$$\leq \begin{cases} \exp\left(-\frac{T\left(\Delta^{a,b}(P)\right)^2}{V^{a,b*}(P)}\right) + AT^{-1/4}(\log T)^{1+1/\alpha} & \text{if } E_0 < \sqrt{T}\Delta^{a,c}(P) \leq E; \\ \exp\left(-\frac{T\left(\Delta^{a,b}(P)\right)^2}{2C_\xi^2}\right) & \text{if } E_0 < \sqrt{T}\Delta^{a,c}(P). \end{cases}$$

Then, as well as the proof of Theorem 5.2, when $K \geq 3$,

$$\sup_{P\in\mathcal{P}^*} \mathbb{E}_P\left[r_T(P)\left(\pi^{\mathrm{AS\text{-}AIPW}}\right)\right]$$
$$\leq \max_{b\in[K]\setminus\mathcal{A}^*(P)} \sqrt{\log(X)\mathbb{E}^X\left[\sum_{b\in[K]}\left(\sigma^b(X)\right)^2\right]/T} + AT^{-1/4}(\log T)^{1+1/\alpha};$$

when $K = 2$,

$$\sup_{P\in\mathcal{P}^*} \mathbb{E}_P\left[r_T(P)\left(\pi^{\mathrm{AS\text{-}AIPW}}\right)\right]$$
$$\leq \max_{b\in[K]\setminus\mathcal{A}^*(P)} \sqrt{\log(K)\mathbb{E}^X\left[\left(\sigma^1(X)+\sigma^2(X)\right)^2\right]/T} + AT^{-1/4}(\log T)^{1+1/\alpha}.$$

Thus, the proof is complete. $\square$

## H.1 PROOF OF THEOREM 5.2

*Proof.* First, we prove the first inequality in Eq. (16), which corresponds to the martingale CLT. From Proposition B.7,

$$\mathbb{P}\left(\widehat{\mu}_T^{\mathrm{AIPW},a} - \widehat{\mu}_T^{\mathrm{AIPW},b} \leq 0\right) = \mathbb{P}\left(\sqrt{T}\left(\widehat{\mu}_T^{\mathrm{AIPW},a} - \widehat{\mu}_T^{\mathrm{AIPW},b}\right) - \sqrt{T}\Delta^{a,b}(P) \leq -\sqrt{T}\Delta^{a,b}(P)\right)$$
$$= \mathbb{P}\left(\sqrt{\frac{T}{V^{a,b*}(P)}}\left(\widehat{\mu}_T^{\mathrm{AIPW},a} - \widehat{\mu}_T^{\mathrm{AIPW},b} - \Delta^{a,b}(P)\right) \leq -\sqrt{\frac{T}{V^{a,b*}(P)}}\Delta^{a,b}(P)\right)$$
$$= \mathbb{P}\left(\sum_{t=1}^T \xi_t^{a,b}(P) \leq -\sqrt{\frac{T}{V^{a,b*}(P)}}\Delta^{a,b}(P)\right)$$
$$\leq \Phi\left(-\sqrt{\frac{T}{V^{a,b*}(P)}}\Delta^{a,b}(P)\right) + AT^{-1/4}(\log T)^{1+1/\alpha}.$$

Let $\phi(\cdot)$ be the density function of the standard normal distribution. Then,

$$\Phi\left(-\sqrt{\frac{T}{V^{a,b*}(P)}}\Delta^{a,b}(P)\right) = \int_{-\infty}^{-\sqrt{\frac{T}{V^{a,b*}(P)}}\Delta^{a,b}(P)} \phi(u)\mathrm{d}u$$
$$\leq \phi\left(-\sqrt{\frac{T}{V^{a,b*}(P)}}\Delta^{a,b}(P)\right) / \sqrt{\frac{T}{V^{a,b*}(P)}}\Delta^{a,b}(P)$$
$$\leq \frac{e}{(e-1)\sqrt{2\pi}} \cdot \frac{\exp\left(-\frac{T\left(\Delta^{a,b}(P)\right)^2}{2V^{a,b*}(P)}\right)}{\sqrt{\frac{T}{V^{a,b*}(P)}}\Delta^{a,b}(P)}.$$

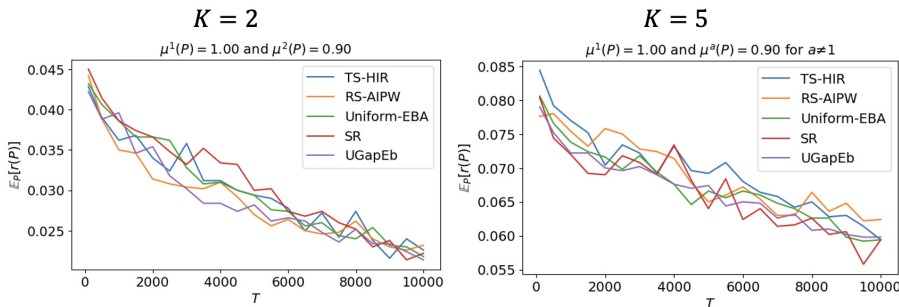

Figure 2: Experimental results. The $y$-axis and $x$-axis denote the expected simple regret $\mathbb{E}_P[r_T(P)(\pi)]$ under each strategy and $T$, respectively.

Here, we used $\frac{e}{(e-1)\sqrt{2\pi}} \approx 0.252$, $\sqrt{\frac{T}{V^{a,b*}(P)}}\Delta^{a,b}(P) > 1$ for large $T$, $\phi(u + 1/u) = \frac{1}{\sqrt{2\pi}}\exp(-(u + 1/u)^2/2) = e^{-1}\phi(u)\exp(-1/(2u^2)) = e^{-1}\phi(u)$ and

$$\int_{-\infty}^{-x}\phi(u)\mathrm{d}u = \int_x^\infty \phi(u)\mathrm{d}u \leq \sum_{k=0}^\infty \frac{1}{x}\phi(x + k/x) \leq \frac{\phi(x)}{x}\sum_{k=0}^\infty \exp(-k) = \frac{e}{e-1}\cdot\frac{\phi(x)}{x},$$

Therefore, if $0 \leq \sqrt{T}\Delta^{a,c}(P) \leq C$,

$$\mathbb{P}\left(\widehat{\mu}_T^{\mathrm{AIPW},a} - \widehat{\mu}_T^{\mathrm{AIPW},c} \leq 0\right) \leq \exp\left(-T\frac{\left(\Delta^{a,b}(P)\right)^2}{2V^{a,b}(P, w^*)}\right) + o(1).$$

Thus, we proved the first inequality.

Next, we show the second inequality of Eq. (16), which is a large deviation bound. $\square$

# I ADDITIONAL EXPERIMENTAL RESULTS

We show addition experimental results. In Appendix I.1, we show results with variances different from those in Section 7. In Appendix I.2, we show the result with continuous contextual information.

## I.1 ADDITION EXPERIMENTAL RESULTS WITHOUT CONTEXTUAL INFORMATION

Under the same setting with that in Section 7, we draw the variances from a uniform distribution with support $[10, 100]$. We show the result in Figure 2.

## I.2 CONTINUOUS CONTEXTUAL INFORMATION

We consider cases with $K = 2, 3, 5, 10$ and 2-dimensional contextual information ($d = 2$). We consider contextual information; therefore, we only investigate the AS-AIPW strategy. Because we cannot obtain a closed-form solution for $K \geq 3$, for simplicity, we fix $w^*(a|x) = \frac{(\sigma^a(x))^2}{\sum_{b\in[K]}(\sigma^b(x))^2}$ for $K \geq 3$, which still reduces the expected simple regret better than $w^*(a) = \frac{(\sigma^a)^2}{\sum_{b\in[K]}(\sigma^b)^2}$. In this section, we do not use SHAdaVar because it is unknown how to incorpolate contextual information to the strategy.

In each set up, the best arm is arm 1. The expected outcomes of suboptimal arms are equivalent and denoted by $\widetilde{\mu} = \mu^2(P) = \mu^K(P)$. We use $\widetilde{\mu} = 0.80, 0.90$. We generate the variance from a uniform distribution with a support $[0.1, 5]$ and contextual information $X_t = (X_{t1}, X_{t2})$ from a multinomial distribution with mean $(1, 1)$ and variance $\begin{pmatrix} 1 & 0.1 \\ 0.1 & 1 \end{pmatrix}$. Let $(\theta_1, \theta_2)$ be random variables generated from a uniform distribution with a support $[0, 1]$. We then generate $\mu^a(P)(X_t) = \theta_1 X_{t1}^2 + \theta_2 X_{t2}^2/c_\mu^a$ and $(\sigma^a(X_t))^2 = (\theta_1 X_{t1}^2 + \theta_2 X_{t2}^2)/c_\sigma^a$, where $c_\mu^a, c_\sigma^a$ are values that adjust the expectation to align with $\mu^a(P)$ and $(\sigma^a)^2$. We continue the experiments until $T = 5,000$ when $\widetilde{\mu} = 0.80$ and

$T = 10,000$ when $\widetilde{\mu} = 0.90$. We conduct $100$ independent trials for each setting. At each $t \in [T]$, we plot the empirical simple regret in Figure 1. Additional results are presented in Appendix I.

From Figure 1 and Appendix I, we can observe that the AS-AIPW performs well when $K = 2$. When $K \geq 3$, although the AS-AIPW tends to outperform the Uniform, other strategies also perform well. We conjecture that the AS-AIPW exhibits superiority against other methods when $K$ is small (mismatching term in the upper bound), the gap between the best and suboptimal arms is small, and the variances significantly vary across arms. As the superiority depends on the situation, we recommend a practitioner to use the AS-AIPW with several strategies in a hybrid way.

We show experimental results with $K = 2, 3, 5, 10$ in Figures 4–6, respectively.

## J   OPEN PROBLEMS

This section introduces several related open issues.

### J.1   OPEN PROBLEM (1): NON-ASYMPTOTIC LOWER BOUNDS

We showed the non-asymptotic upper bound in Appendix H. We are also intrested in deriving non-asympttic lower bound. However, there are the following issues.

- Although the estimation error of the variances can be ignored in the worst-case asymptotic analysis, it affects the lower bound in the non-asymptotic analysis. However, the analysis of variance estimation might be too complicated to analyze.

- Our technique is based on the information-theoretic lower bound provided by Kaufmann et al. (2016) and semiparametric efficiency bounds. Both techniques are used for asymptotic analysis. Therefore, we need to develop completely different approaches for non-asymptotic analysis.

- We deal with general distribution by approximating the KL divergences. If we focus on the non-asymptotic analysis, we need to restrict the class of distributions to specific distributions, such as the Gaussian distribution, even if possible.

One of the promising approaches for lower bounds is to employ lower bounds provided by Carpentier & Locatelli (2016). However, this lower bound is based on the boundedness of $Y_t^a$. Without the boundedness, non-asymptotic analysis would be more difficult. Additionally, the definitions of optimality might be changed to deal with the uncertainty of variance estimation. Thus, although the non-asymptotic upper bound has been derived, deriving non-asymptotic lower bounds requires different techniques and is not straightforward, even if possible.

### J.2   OPEN PROBLEM (2): GENERALIZATION OF THEOREM 3.8 FOR $K \geq 3$

We conjecture that there exist more tighter lower bounds for $K \geq 3$ than Theorem 3.4, as well as Theorem 3.8 for $K = 2$. This conjecture is also based on an intuition that while Theorem 3.8 connects to existing tight lower bounds in two-armed Gaussian bandits (Kaufmann et al., 2016), Theorem 3.4 does not.

For example, Kato (2023) shows that different target allocation ratios should be used for minimization of the probability of misidentification $\mathbb{P}_P(\hat{a}_T \neq a^*(P))$. That work considers Gaussian bandits without contextual information, defined as

$$\mathcal{P} :=$$
$$\left\{ P = \left( \mathcal{N}\left( \mu^a, (\sigma^a)^2 \right) \right)_{a \in [K]} \mid (\mu^a)_{a \in [K]} \in \mathbb{R}^K, \ (\sigma^a)^2_{a \in [K]} \in [\underline{C}, \overline{C}]^K, \ \exists a^* \in [K] \text{ s.t. } \mu^{a^*} > \max_{a \in [K] \setminus \{a^*\}} \mu^a \right\},$$

where $\underline{C}, \overline{C}$ are universal constants such that $0 < \underline{C} < \overline{C} < \infty$. Then, that work shows that the lower bound is given as

$$\sup_{P \in \mathcal{P}} \limsup_{T \to \infty} -\frac{1}{T} \log \mathbb{P}_P(\widehat{a}_T^\pi \neq a^*(P_0)) \leq \max_{w \in \mathcal{W}} \min_{b \in [K], a \in [K] \setminus \{b\}} \frac{\overline{\Delta}^2}{2\Omega^{b,a}(w)} + o\left(\overline{\Delta}^2\right),$$

as $\overline{\Delta} \to 0$, where $\overline{\Delta} = \max_{P \in \mathcal{P}} \max_{a \in [K] \setminus \{a^*(P)\}} \Delta^a(P)$, and $\Omega^{b,a}(w) = \frac{\left(\sigma^b\right)^2}{w(b)} + \frac{\left(\sigma^a\right)^2}{w(a)}$. Here, the target allocation ratio is given as

$$w^* = \arg\max_{w \in \mathcal{W}} \min_{b \in [K], a \in [K] \setminus \{b\}} \frac{1}{2\Omega^{b,a}(w)},$$

which is different from ours.

As well as Kato (2023) makes several additional assumptions for the strategy class and bandit models, we believe that our current assumptions are insufficient to extend Theorem 3.8 for $K \geq 3$. Such a necessity of additional assumptions also has been discussed in Komiyama et al. (2023), but an effective solution has not been proposed.

One of the promising candidates for solutions is to employ the framework of the limit of experiments van der Vaart (1998) is one of the promising directions. This framework is expected to allow us to derive tight lower and upper bounds by using the asymptotic normality. To apply the results, we need to restrict strategy class and underlying bandit models such that under which mean estimators in the recommendation phase follow a normal distribution asymptotically.

In summary, tightening Theorem 3.4 or extending Theorem 3.8 to cases with $K \geq 3$ are a crucial important issue. To address this open issue, we believe that further assumptions are required.

### J.3 OPEN PROBLEM (3): RELATIONSHIP WITH THE AIPW ESTIMATOR AND THE SAMPLE AVERAGE

We defined the AIPW estimator $\widehat{\mu}_T^{\mathrm{AIPW},a}$ in the recommendation phase. However, we also conjecture that we can employ the sample average $\widehat{\mu}_T^a$ when there is no contextual information.

In fact, Hahn et al. (2011) shows that when using the CLT, the AIPW and sample average have the same asymptotic distribution in the literature of efficient average treatment effect estimation, a setting related to BAI.

However, proving the regret upper bound for $\hat{\mu}$ requires a more complicated proof procedure compared to $\hat{\mu}$ or requires some additional assumptions. For example, we cannot employ the martingale theory for the proof.

Note that even if we use the sample average $\widehat{\mu}_T^a$ instead of the AIPW estimator $\widehat{\mu}_T^{\mathrm{AIPW},a}$, the AIPW estimator $\widehat{\mu}_T^{\mathrm{AIPW},a}$ still plays an important role in theoretical analysis of the sample average estimator $\widehat{\mu}_T^a$. This is because Hahn et al. (2011) shows the asymptotic distributions of the sample average by going through that of the (theoretically constructed) AIPW estimator as a result of the empirical process theory.

Thus, deriving a variance-dependent tight upper bound of a strategy using $\widehat{\mu}_T^a$ in the recommendation phase is an open issue.

### J.4 OPEN PROBLEM (4): TRACKING-BASED SAMPLING RULE

We proposed randomly sampling arms following $w_t$. However, there are other sampling strategies used in BAI. For example, Garivier & Kaufmann (2016) proposes using a tracking-type sampling rule. Here, we discuss the possibility of using such a sampling rule.

For simplicity, let us omit $X_t$. If we apply a tracking-type sampling rule, for example, we can draw arms as $A_t = \arg\min_a \sum_{s=1}^{t-1} \mathbb{1}[A_s = a] - t\hat{w}_t(a)$.

This sampling rule might outperform our proposed random sampling because it has less randomness in the choice of $A_t$ and stabilizes the behavior of strategies, as pointed out by Fiez et al. (2019).

Although we can expect performance improvement, the derivation of the upper bound requires additional theoretical techniques. In our study, the derivation of the upper bound is based on the martingale property, depending on $\mathbb{E}[\mathbb{1}[A_t = a] | \mathcal{F}_{t-1}] = w_t(a)$.

If we apply the tracking-based algorithm, we need to reconsider the definition of the AIPW estimator. Depending on the definition, there are some technical issues. For example, consider using the same definition of the AIPW estimator in the main text. Then, $w_t$ is an estimator of the optimal allocation, and $A_t$ is an actual arm draw. Then, $\mathbb{E}[\mathbb{1}[A_t = a] | \mathcal{F}_{t-1}] = w_t(a)$ does not hold. Therefore, we cannot employ the martingale property.

However, we consider that there is a possibility that we can show the same property between the AIPW estimator with random sampling in our manuscript and tracking-based sampling in your proposition, as discussed in Hahn et al. (2011) and Kato (2021). However, to conduct the proof in Hahn et al. (2011) and Kato (2021), we need to focus on more restricted strategies or make stronger assumptions. For example, Hahn et al. (2011) only considers a two-stage adaptive experiment, where we draw each arm with the equal ratio in the first stage and draw each arm to track the empirical optimal allocation ratio. Here, we cannot update $w_t$ in the second stage; that is, we can update the sampling rule $A_t$ only once between the first and second experiments. Kato (2021) makes complicated assumptions that are not easily verified. Showing the upper bound under the tracking-based algorithm without restricting the sampling rules or making additional assumptions is an open issue.

Furthermore, we point out that there is a trade-off between the fast convergence of $A_t$ and the bias of an estimator of $\mu_t^a(P)$. That is, although a tracking-based algorithm may stabilize the behavior of $A_t$, it loses the unbiasedness of an estimator of $\mu^a(P)$.

In summary, a tracking-based strategy might outperform random-sampling-based strategies. However, under such a strategy, we cannot employ the martingale property, which makes the theoretical analysis difficult. Revealing the theoretical properties of such an estimator is an open issue.

## J.5    OPEN PROBLEM (5): CONTEXT-SPECIFIC RECOMMENDATION

Although we defined BAI with contextual information as the problem of recommending an arm with the highest expected reward marginalized over a contextual distribution, we believe that conducting context-specific arm recommendation is possible when there is not much discrete contextual information (that is, $\mathcal{X} = \{S_1, S_2, \ldots, S_M\}$ with a discrete context $S_m$ and a small $M > 0$).

We are now extending the result for policy learning with BAI; that is, given a set $\Pi$ of policies $\pi : [K] \times \mathcal{X} \rightarrow (0, 1)$ such that $\sum_{a \in [K]} \pi(a|x) = 1$ (and potentially continuous contextual information), we train a policy $\pi \in \Pi$ to minimize the regret. Consider we restrict a policy class $\Pi$ to the one such that we discretize the contextual information and recommend an arm within each discretized contextual information. Then, such a strategy aligns with the strategy that the reviewer suggested.

The remaining open issue is how we bound the regret for general $\Pi$. Because samples are non-i.i.d., we cannot directly apply the standard complexity measure such as the Rademacher complexity. For example, we might employ the martingale-version of the Rademacher complexity proposed by Rakhlin et al. (2015).

This open problem has garnered attention in this literature Zhan et al. (2021) and Zhan et al. (2022).

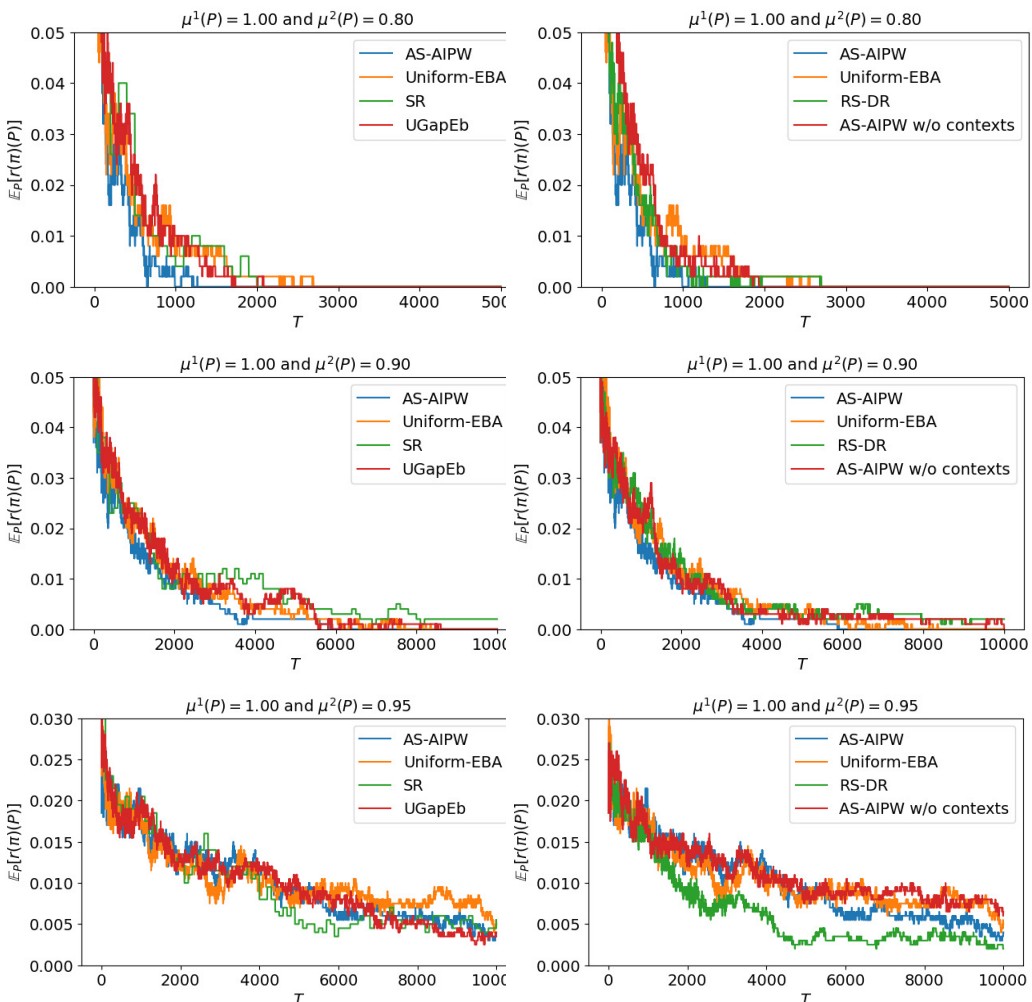

Figure 3: Results when $K = 2$.

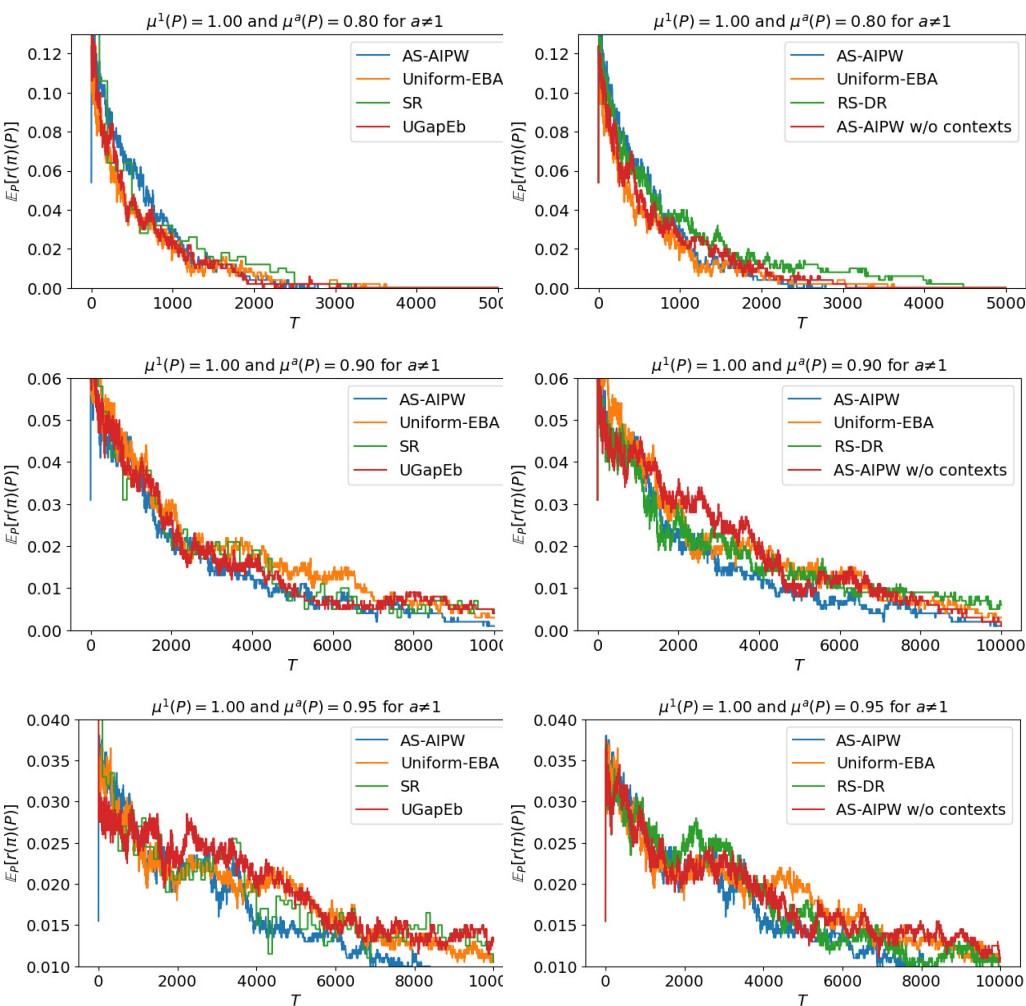

Figure 4: Results when $K = 3$.

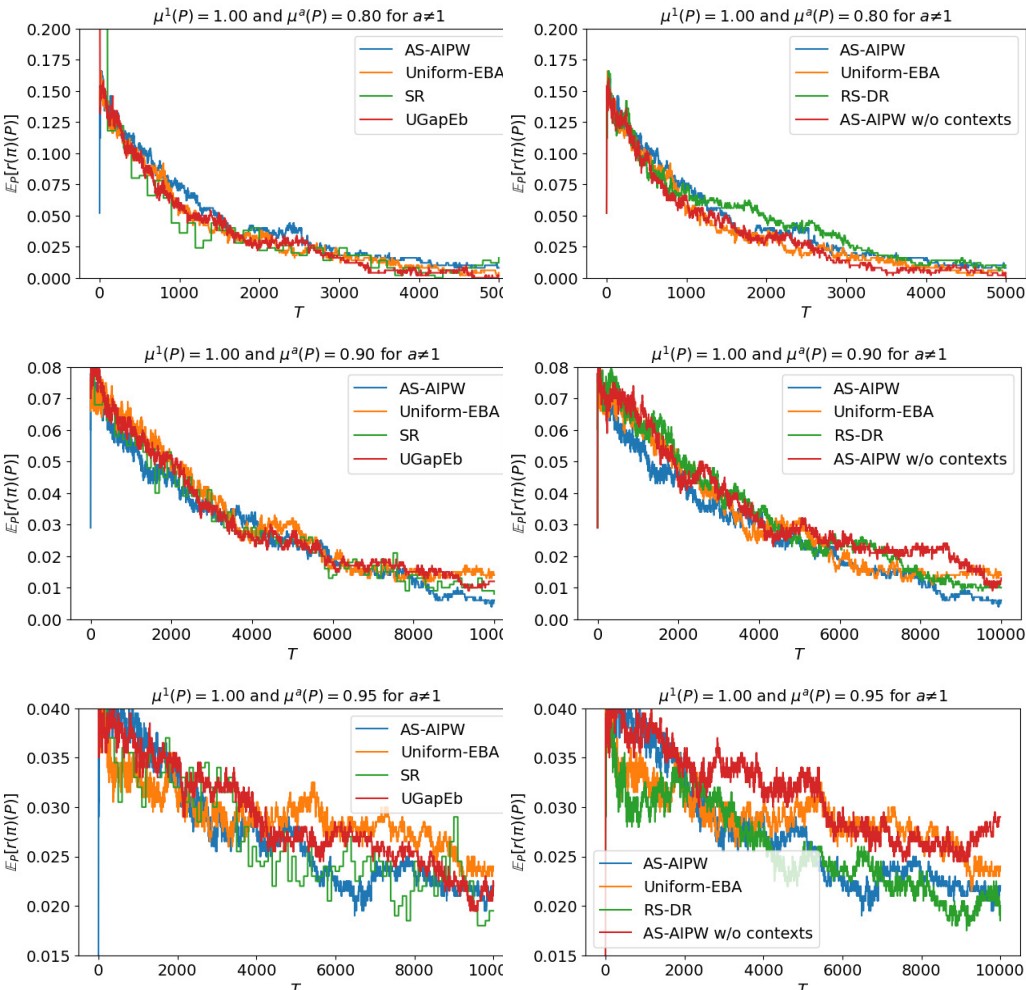

Figure 5: Results when $K = 5$.

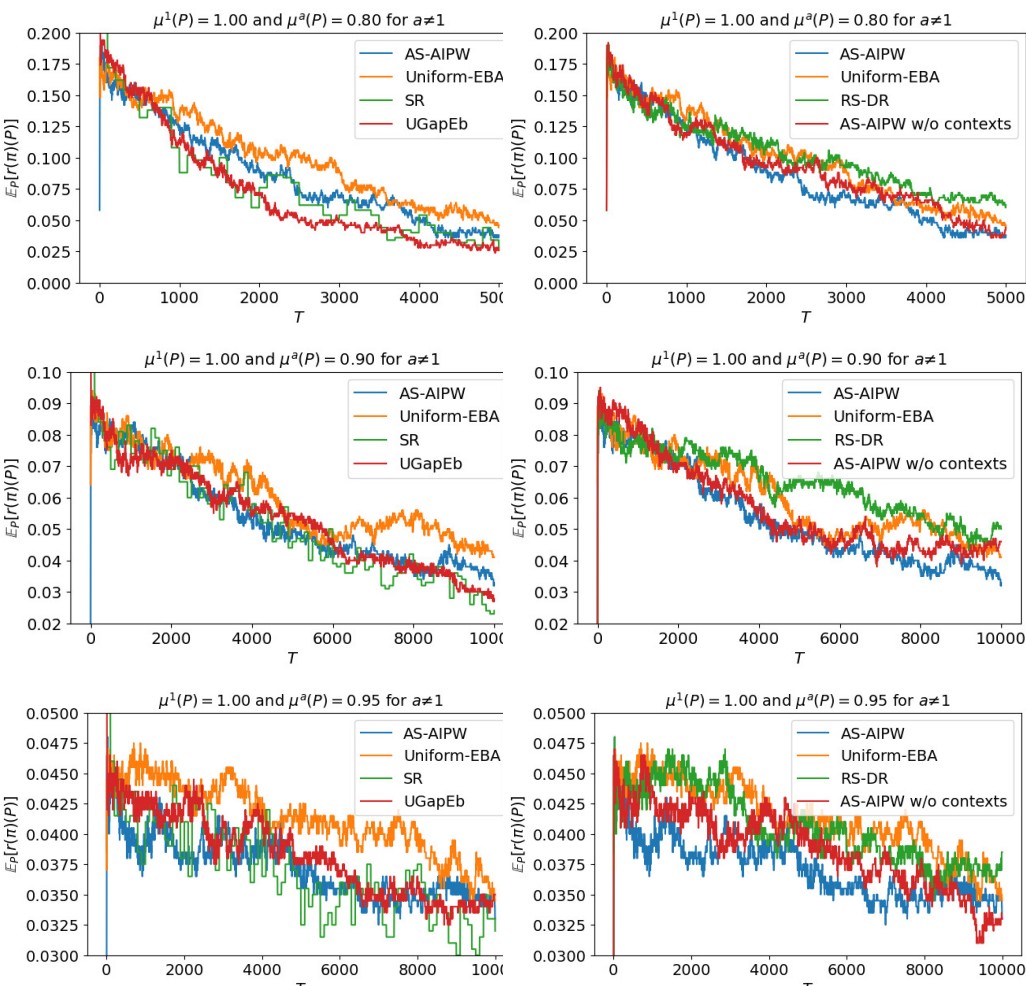

Figure 6: Results when $K = 10$.

