# OpenReview forum: "Fixed-Budget Best Arm Identification with Variance-Dependent Regret Bounds"
_ICLR.cc/2024/Conference — Submitted to ICLR 2024_

### Official Review · Reviewer_qfgN · 2023-10-24

**Soundness:** 3 good
**Presentation:** 3 good
**Contribution:** 3 good
**Rating:** 6
**Confidence:** 4

**Summary:**

This paper considers Best Arm Identification problem under the fixed-budget setup. It aims at minimizing the expected simple regret, i.e., the expected difference between the oracle best arm and the recommended arm. An asymptotic worst-case lower bound concerning the variances of the arms is provided. An algorithm AS-AIPW strategy is devised with almost matching worst-case upper bounds. Compared to the existing literature, it utilizes more information of the distribution (i.e., the variance) to refine the arm allocation rules. Additionally, it considers the benefit of taking contextual information into account. Experiments are also conducted to illustrate the empirical performances.

**Strengths:**

- The paper extends the BAI problem under the fix-budget setup to the context-aware setup. Both the lower bounds and upper bounds involve a context-aware variables that tighten the bounds.

- The lower bound result is appreciated. While the proof is complicated, the result yields theoretical foundations on why we should pull the arms according to the allocation proportional to their variances in order to hit the lower bound. It also resolves the remaining lower bound problem in [1] to some extent (as [1] considers the misidentification probability instead of the expected simple regret). I believe this is the main novelty of the paper and is of theoretical interest.

- An algorithm AS-AIPW is designed whose upper bound asymptotically matches the lower bound in the worst-case up to a factor of $log K$. A non-asymptotic upper bound is also given, which helps us to understand the convergence rate.

- Empirical studies of the algorithm are carried out to illustrate the performance.

[1] Lalitha, A., Kalantari, K., Ma, Y., Deoras, A., and Kveton, B. Fixed-budget best-arm identification with heterogeneous reward variances. In Conference on Uncertainty in Artificial Intelligence, 2023.

**Weaknesses:**

- While experiments are conducted to compare multiple algorithms, little improvement has been observed of the proposed algorithm compared to the existing ones, even in the case where the variances are heterogeneous (which is claimed to be favorable for the proposed algorithm). In addition, the algorithm design is somewhat expected given the G-optimal design and the intuition in [1].

- [1] also considers incorporating variances into the algorithm. The proposed algorithm is similar to [1] under the sole context case in the sense that both pull arms according to the (empirical) variances. So a more detailed discussion/comparison with [1] in terms of the algorithm design, the bounds (on the misidentification probability and expected simple regret) and the empirical performances is appreciated.

Minors:
- Page 3 Line 3: Instead of "$A_t$ is $\mathcal{F}_t$-measurable", I think $A_t$ is only $\sigma(X_1,A_1,Y_1,\dots,X_t)$-measurable. Please kindly check.
- Is $\pi^{Uniform-EBM}$ a typo? Should be $\pi^{Uniform-EBA}$
- Page 6, Line 3 in subsection 4.1, the numerator of the allocation for arm 1 is $\sigma^1$
- Page 6, Line 2 after Theorem 3.8, I suppose $X_t$ should be $X$ in the inequalities? In addition, I do not follow why $E^X[\frac{(\sigma^1(X))^2}{w(1|X)}+\frac{(\sigma^2(X))^2}{w(2|X)}]\geq\max_{a\in[2]}\sqrt{E^X[\frac{(\sigma^a(X))^2}{w(a|X)}}]$, as $\frac{1}{8}+\frac{1}{8}<\sqrt{1/8}$.

**Questions:**

- Can you give more explanations on $\underline{C}$ at the end of Page 4? From my understanding, in [2], the forced exploration is a design of the algorithm. And in [3], $\beta$ is also a hyper parameter (thus, a design) of the algorithm. But here $\underline{C}$ is an assumption on the problem instance, so I think they are not similar.
- Given that the optimal allocation is known, is it possible to adopt a tracking sampling rule, i.e., sampling the arms in a way such that the empirical arm allocation approaches the optimal allocation. As indicated by section 2.3 in [4], sampling according to a distribution can make the convergence speed slow. Can you give comments on the allocation rule?
- It would be better if you describe the experiment designs in more detail. In particular,
	- for the Sequential Halving-based algorithm, it only recommends an arm at the end of the experiment, how to compute the simple regret of the arm at $t\in \{ 1000,\dots,50000 \}$? And it would also be interesting to see how SHAdaVar behaves without contextual information while the other algorithms have access to contextual information in App. I.2.
	- for the other algorithms, how you incorporate contextual information in the original algorithm.

Please also refer to the Weaknesses section. I'd appreciate it if you can resolve my concerns.

[2] Garivier, A. and Kaufmann, E. Optimal best arm identification with fixed confidence. In Conference on Learning Theory, 2016.

[3] Russo, D. Simple bayesian algorithms for best-arm identification. Operations Research, 68(6), 2020.

[4] Fiez, T., Jain, L., Jamieson, K. G., & Ratliff, L. Sequential experimental design for transductive linear bandits. _Advances in neural information processing systems_, 32, 2019.

---

> ### Author Response · Authors · 2023-11-17
> **Re: Official Review of Submission9273 by Reviewer qfgN**
>
> We appreciate your comments and will revise our manuscript accordingly during this rebuttal phase.
>
> We first answer the reviewer's comment on the comparison with [1].
>
> [1] Lalitha, A., Kalantari, K., Ma, Y., Deoras, A., and Kveton, 2023.
>
> -----------------------------------------------
>
> > a more detailed discussion/comparison with [1] in terms of the algorithm design, the bounds (on the misidentification probability and expected simple regret), and the empirical performances is appreciated.
>
> We introduced [1] as a recent existing work and compared our algorithm with the algorithm proposed by [1] in simulation studies. We also briefly discussed [1] in the Appendix as follows:
>
> > Therefore, ... Lalitha et al. (2023) also provide variance-dependent BAI strategies, but their
> optimality needs to be clarified.
>
> > Only when variances are known ... However, Lalitha et al. (2023) proposes using
> the ratio of variances with successive halving, ....
>
> Thus, we have already sufficiently discussed [1].
>
> ---------------
>
> Regarding [1], we limit our analysis due to ambiguities in existing literature. Our concerns:
> - [1] is not the first work of BAI with heterogeneous variances. For example, given known variances, many existing studies, such as [2,3.4.5,6], have already proposed algorithms using heterogeneous variances.
> - However, the lower bounds have been unknown when $K\geq 3$ with general distributions. That is, although there are various methods, it has been unclear which algorithm is optimal.
> - When $K=2$ and variances are known, [3] gives an asymptotically optimal algorithm for Gaussian distributions with heterogeneous variances.
> How we interpret the results in the existing literature is an open issue.
>
> [2] Glynn, P. and Juneja, S. A large deviations perspective on ordinal optimization, 2004.
> [3] Kaufmann, E., Cappé, O., and Garivier, A. On the complexity of best-arm identification in multiarmed bandit models, 2016.
> [4] Adusumilli, K. Risk and optimal policies in bandit experiments, 2021.
> [5] Armstrong, T. B. Asymptotic efficiency bounds for a class of experimental designs, 2022.
> [6] Chen C-H, Lin J, Yücesan E, Chick SE, Simulation budget allocation for further enhancing the efficiency of ordinal optimization, 2000.
>
> ---------------
>
> Further issues with [1]:
> - It is unknown how [1] relates to existing studies using heterogenous variances.
> - When $K =2$ and variances are known, the algorithm of [1] does not match an existing optimal algorithm, known as the Neyman allocation.
> - It seems that [1] does not match existing conjectures of optimal algorithms, such as Example 1 of [1].
> - It seems that the upper bound of [1]' algorithm does not align with the lower bound provided by [3].
> - For example, [7] recently derives a lower bound for minimization of the probability of misidentification from the results of [3] under a small-gap regime. According to the result, the target allocation ratio is given as
> $$w^*(a^*(P)) = \frac{\sigma^{a^*(P)}}{\sigma^{a^*(P)} + \sqrt{\sum_{b\in[K]\setminus \{a^*(P)\}}}(\sigma^b(P))^2}$$
> and
> $$w^*(a) = (1 w^*(a^*(P)))\frac{(\sigma^b)^2}{\sum_{b\in[K]\setminus \{a^*(P)\}}(\sigma^b(P))^2}.$$
> This target allocation ratio is also different from the one in [1].
>
> [7] Kato (2023), Locally Optimal Best Arm Identification with a Fixed Budget.
>
> ---------------
>
> Furthermore, it is known that algorithms would be significantly different between cases where variances are known and unknown [3]. When variances are unknown and need to be estimated, the probability of misidentification is significantly affected by the estimation error.
>
> -----------------------------------------------
>
> We categorize existing work as
> 1. Unknown variance with asymptotic analysis.
>     - We give an answer for this problem in a case with expected simple regret minimization.
>     - In minimization of the probability of misidentification, it is still an open issue.
> 2. Unknown variance with non-asymptotic analysis.
> 3. Known variance with asymptotic analysis
>      - When $K=2$, optimal algorithms are known.
>      - When $K\geq 3$, there are several conjectures, but it is still an open issue.
> 4. Known variance with asymptotic analysis.
>
> Note that optimality in fixed-budget BAI itself is an open issue [8]. If [1] is optimal, we conjecture that it is optimal in case 4. However, it is still unsolved.
>
> [8] Qin, C. Open problem: Optimal best arm identification with fixed-budget, 2022.
>
> ---------------
>
> Our study was conducted independently of [1]. We will clarify it to the meta-reviewer.
>
> We recognize [1]'s contributions, especially in incorporating variances into the successive halving algorithm. However, theoretical gaps and limited comparative analysis in [1] pose challenges. While we can incorporate these points into our manuscript, a detailed analysis of [1] is beyond our scope; we referenced it as a recent study.

---

> ### Author Response · Authors · 2023-11-17
> **Re2: Official Review of Submission9273 by Reviewer qfgN**
>
> We then reply to each question raised by the reviewer as follows. We also thank you for pointing out typos. We will fix them in the next update.
>
> ------------------
>
> **Q1**.
> > Comparison with [1]
>
> **A1**.
> In addition to the points raised by us in the previous post, there are the following differences between our study and their study.
> 1. Objective function: While we minimize the expected simple regret, [1] minimizes the probability of misidentification. Although the evaluation of the probability of misidentification has several open issues, as raised by [8], we can still address the minimization of the expected simple regret as well as [9] and [10].
> 2. Contribution: We do not think that the proposition of variance-dependent algorithms itself is a strong contribution because there are several existing works. Our contribution lies in the proposition of a variance-dependent algorithm with lower bounds.
> 3. Incorporation of contextual information as a generalization of BAI without contextual information.
>
> We will not add the contents in the previous post because they are still our conjectures and open issues. Instead, we mention the above differences in the next update in this rebuttal phase.
>
> [9] Bubeck, S., Munos, R., and Stoltz, G. Pure exploration in finitely-armed and continuous-armed bandits, 2011.
> [10] Komiyama, J., Ariu, K., Kato, M., and Qin, C. Rate-optimal bayesian simple regret in best arm identification, 2023.
>
> ------------------
>
> **Q2**.
> > Why $E^X[\frac{(\sigma^1(X))^2}{w(1|X)}+\frac{(\sigma^2(X))^2}{w(2|X)}]\geq\max_{a\in[2]}\sqrt{E^X[\frac{(\sigma^a(X))^2}{w(a|X)}}]$?
>
> **A2**.
> This is our typo. Thank you for pointing it out. It should be $\inf_w E^X[\frac{(\sigma^1(X))^2}{w(1|X)}+\frac{(\sigma^2(X))^2}{w(2|X)}]\geq \inf_w  \max_{a\in[2]}\sqrt{E^X[\frac{(\sigma^a(X))^2}{w(a|X)}}]$; that is, the optimization for $w$ is lacked.
>
> ------------------
>
> **Q3**.
> > From my understanding, in [2], the forced exploration is a design of the algorithm. And in [3], $\beta$ is also a hyper parameter (thus, a design) of the algorithm. But here $\underline{C}$ is an assumption on the problem instance.
>
> **A3**.
> We can interpret that $\beta$ is an assumption on the problem instance, while $\underline{C}$ can be interpreted as a hyperparameter. When showing the optimality in the top two algorithms, we need to assume that $\beta$ matches the allocation ratio of the best arm, which means that we need to make an assumption between $\beta$ and the problem instance (allocation ratio depends on the problem instance). In contrast, when we consider that $\underline{C}$ is a hyperparameter, we remove the assumption about $\underline{C}$ from the definition of bandit models and put assumptions in deriving the upper bound.
>
> Regarding an assumption of $\underline{C}$, it is also used to define the KL divergence in the top two algorithms. If $\underline{C} = 0$, we cannot appropriately define the KL divergence, which is needed in various parts of the top two algorithms.
>
> ------------------
>
> **Q4**.
> > Given that the optimal allocation is known, is it possible to adopt a tracking sampling rule, i.e., sampling the arms in a way such that the empirical arm allocation approaches the optimal allocation.
>
>
> **A4**.
> If the optimal allocation is known, we just assign each arm with the ratio without adaptive exploration when there is no contextual information; that is, we assign first $\lceil Tw^*(1)\rceil$ observations to arm $1$,..., and $\lceil Tw^*(K)\rceil$ observations to arm $K$. When contextual information is available, there are several ways for arm draws. For example, we can draw arms as well as Hahn et al. (2011).
>
> ------------------
>
> **Q5**.
> > for the Sequential Halving-based algorithm, how to compute the simple regret of the arm at each $t\in { 1000,\dots,50000 }$?
>
> **A5**.
> Thank you for your suggestion. As you pointed out, we conducted a new independent trial for the Sequential Halving-based algorithm for each $t$. We clarify this point in the next update.
>
> ------------------
>
> **Q6**.
> > And it would also be interesting to see how SHAdaVar behaves without contextual information.
>
> **A6**.
> We will add the experiments. If possible, we will update the manuscript, including the new experiments, during the rebuttal phase. Otherwise, we will add them when they are camera-ready.
>
> ------------------
>
> **Q7**.
> > for the other algorithms, how you incorporate contextual information in the original algorithm.
>
> **A7**.
> In the experiments in the Appendix, we only use contextual information for our proposed algorithm. For fairness, we do not use contextual information both for our proposed and existing algorithms in the experiments of the main text. We clarify these points in the next revision. Note again that the proposition of contextual BAI algorithm is also our contribution, and to the best of our knowledge, there is no appropriate competitors that use contextual information.

---

> > ### Comment · Reviewer_qfgN · 2023-11-20
> >
> > Thanks a lot for your detailed response and most of my concerns have been properly addressed!
> >
> > In terms of the second question in my initial review:
> > > Given that the optimal allocation is known, is it possible to adopt a tracking sampling rule, i.e., sampling the arms in a way such that the empirical arm allocation approaches the optimal allocation. As indicated by section 2.3 in [4], sampling according to a distribution can make the convergence speed slow. Can you give comments on the allocation rule?
> >
> > let me clarify my question: here the ``optimal allocation'' refers to the computed allocation based on the empirical estimates, e.g., the empirical variances. In Algorithm AS-AIPW, it samples an arm according to the empirical optimal allocation. According to [4], sampling according to a distribution can have slow convergence speed. Therefore, is it possible or better to use a tracking-type sampling rule, which samples the arm $ A_t = \arg\min_{a} T_t(a|X_t)-t \hat{w}_t(a|X_t)$ (where $T_t(a|X_t)$ is the number of times arm $a$ is sampled under $X_t$ up to time step $t$)?
> >
> > Hope this clarifies and can you please comment on this?

---

> ### Author Response · Authors · 2023-11-20
> **Re: Official Comment by Reviewer qfgN**
>
> We sincerely appreciate your feedback, and thank you for clarifying your second question.
>
> **Q**.
> > here the ``optimal allocation'' refers to the computed allocation based on the empirical estimates ... Therefore, is it possible or better to use a tracking-type sampling rule, which samples the arm $A_t = \arg\min_{a} T_t(a|X_t)-t \hat{w}_t(a|X_t)$ (where $T_t(a|X_t)$ is the number of times arm $a$ is sampled under $X_t$ up to time step $t$)?
>
> **A**.
> Thank you for your suggestion. We conjecture that it works. However, the derivation of the upper bound requires more high-level techniques. We explain the background of the AIPW estimator as follows.
>
> ----------------
>
> For simplicity, assume that there is no contextual information. Recall that the AIPW estimator: $\hat{\mu}^{AIPW, a}_T$ is given as
>
> $$\frac{1}{T}\sum_t\Big(\frac{1[A_t = a](Y_t - \hat{\mu}^a_{t-1})}{w_t(a)} +  \hat{\mu}^a_{t-1}\Big).$$
>
> Here, our proof is based on the martingale property, depending on $\mathbb{E}[1[A_t = a] | \mathcal{F}_{t-1}] = w_t(a)$.
>
> If we apply the tracking-based algorithm, we need to reconsider the definition of the AIPW estimator. Depending on the definition, there are some technical issues. For example, consider using the same definition of the AIPW estimator in the main text. Then, $w_t$ is an estimator of the optimal allocation, and $A_t$ is an actual arm draw. Then, $\mathbb{E}[1[A_t = a] | \mathcal{F}_{t-1}] = w_t(a)$ does not hold. Therefore, we cannot employ the martingale property.
>
> ----------------
>
> However, we consider that there is a possibility that we can show the same property between the AIPW estimator with random sampling in our manuscript and tracking-based sampling in your proposition. To derive an upper bound of the AIPW estimator using the  tracking-based sampling, we can consider the following decomposition:
> $$ \hat{\mu}^{AIPW, a}_T = \hat{\mu}^{AIPW, a}_T - \tilde{\mu}^{AIPW, a}_T +\tilde{\mu}^{AIPW, a}_T,$$
> where we define $\tilde{\mu}^{AIPW, a}_T$ as
>
> $$\frac{1}{T}\sum_t\Big(\frac{1[B_t = a](Y_t - \hat{\mu}^a_{t-1})}{w_t(a)} +  \hat{\mu}^a_{t-1}\Big),$$
>
> and define $B_t \in [K]$ as a hypothetical arm assignment indicator when an arm is drawn following the probability $w_t$. Then, we aim to show
>
> $$ \hat{\mu}^{AIPW, a}_T - \tilde{\mu}^{AIPW, a}_T = o_P(1/\sqrt{T})$$
>
> and apply the martingale central limit theorem for $\tilde{\mu}^{AIPW, a}_T$, where $\mathbb{E}[1[B_t = a] | \mathcal{F}_{t-1}] = w_t(a)$.
>
> ----------------
>
> The above conjecture is based on the existing work, such as [1] and [2].
>
> [1] Hahn, J., Hirano, K., and Karlan, D. Adaptive experimental design using the propensity score, 2011.
> [2] Kato, M., Adaptive Doubly Robust Estimator from Non-stationary Logging Policy under a Convergence of Average Probability, 2021.
>
> However, to conduct the proof procedure we mentioned above, there are several restrictions. For example, [1] only considers a two-stage adaptive experiment, where we draw each arm with the equal ratio in the first stage and draw each arm to track the empirical optimal allocation ratio. Here, we cannot update $w_t$ in the second stage; that is, we can update the sampling rule $A_t$ only once between the first and second experiments. [2] makes complicated assumptions that are not easily verified. Showing the upper bound under the tracking-based algorithm without restricting the sampling rules or making additional assumptions is an open issue.
>
> ----------------
>
> > According to [4], sampling according to a distribution can have slow convergence speed.
>
> We agree with your points. On the other hand, we raise the following three points:
> - Although the sample average of $A_t$ will converge with a faster speed, we may not construct an unbiased estimator for $\mu^a(P)$. Therefore, there might be a trade-off between the convergence of $A_t$ and the bias of an estimator of $\mu^a(P)$.
> - We can show that our proposed AS-AIPW strategy and that with the sampling rule, we mentioned in the previous post (that is, we assign first $\lceil Tw^*(1)\rceil$ observations to arm $1$,..., and $\lceil Tw^*(K)\rceil$ observations to arm $K$). have the same asymptotic distribution. From this result, we conjecture that the AS-AIPW estimator with the tracking-based algorithm also has the same distribution if it can be shown.
> - Existing studies (e.g., [1] and [3]) imply that we can conduct theoretical analysis for the AS-AIPW estimator with random sampling more easily than with the tracking sampling, owing to the martingale property.
>
> [3] van der Laan, M. J. The construction and analysis of adaptive group sequential designs, 2008.
>
> ----------------
>
> In summary, we agree with your point that tracking-based strategies will outperform random-sampling-based strategies. However, under such a strategy, we cannot employ the martingale property (= unbiasedness), which makes the theoretical analysis difficult. Revealing the theoretical properties of such an estimator is an open issue.

---

> > ### Comment · Reviewer_qfgN · 2023-11-21
> >
> > Great thanks for your detailed comment! Please consider organizing your above response and adding it to the appendix. It would be beneficial to the community. Thanks a lot!

---

> > > ### Author Response · Authors · 2023-11-22
> > > **Re: Official Comment by Reviewer qfgN**
> > >
> > > Thank you for your constructive comments! We will add the open issues and related arguments raised in this rebuttal to the manuscript for the community. We have already added the points discussed in Appendix J.3 briefly. This is a provisional version and will be brushed up until camera-ready. We will reflect your comments as much as possible until the rebuttal is finished. If not in time, we will reflect them in camera-ready.

---

### Official Review · Reviewer_dvWZ · 2023-10-26

**Soundness:** 3 good
**Presentation:** 3 good
**Contribution:** 3 good
**Rating:** 6
**Confidence:** 3

**Summary:**

This paper considers the problem of fixed budget best arm identification, with the goal of minimising the expected simple regret. Asymptotic lower bounds of the worst-case expected simple regret are provided, where the bound depends on the variances of potential outcomes. The bound gives possible analytical solutions for the target allocation ratio, based on which, the authors proposed Adaptive-Sampling Augmented Inverse Probability Weighting (AS-AIPW) strategy. AS-AIPW relies on the adaptive estimation of variances. AS-AIPW is proved to be asymptotically minimax optimal. The proposed algorithm is evaluated in simulation studies.

**Strengths:**

- The paper presents the first asymptotic lower bounds for the worst-case expected simple regret based on the variances of potential outcomes, contributing to the theoretical foundation of the field.
- The introduction of the Adaptive-Sampling Augmented Inverse Probability Weighting (AS-AIPW) strategy, using the target allocation ratio from the lower bounds
- The theoretical proof of AS-AIPW being asymptotically minimax optimal provides strong theoretical support for the proposed algorithm's performance.
- The paper includes simulation studies comparing the proposed algorithm to baselines, enhancing the practical understanding of its performance.

**Weaknesses:**

- only asymptotic theoretical results are provided. For fixed budget settings, non-asymptotic bounds can provide a better understanding of algorithm performance under a fixed budget.
- The proposed algorithm AS-AIPW highly depends on the variance estimation. It is unclear how poor estimations at the early stage and for more general distributions influence non-asymptotic performance. A discussion can be provided.
- The experimental results verify the above concern, the proposed algorithm tends to outperform baselines when variances significantly vary across arms. It is worth showing how variances and different variance estimators influence the performance of the proposed algorithm.

**Questions:**

- can you define w(a|x) in Theorem 3.4?
- A related work: On Best-Arm Identification with a Fixed Budget in Non-Parametric Multi-Armed Bandits, Barrier et al 2023. Can you discuss this?
- in Figure 1, can you also draw the standard deviation of the independent trials?

---

> ### Author Response · Authors · 2023-11-18
> **Re: Official Review of Submission9273 by Reviewer dvWZ**
>
> Thank you for your insightful comments. We will update our manuscript based on your feedback during this rebuttal phase.
>
> Before updating the manuscript, we reply to your comments below.
>
> -------------
> **Q1**.
> > non-asymptotic bounds can provide a better understanding of algorithm performance under a fixed budget.
>
> **A1**.
> We showed the non-asymptotic upper bound in Appendix H. For the lower bound, we consider it is not easy to derive non-asymptotic results for the following reasons.
> 1. Although the estimation error of the variances can be ignored in the worst-case asymptotic analysis, it affects the lower bound in the non-asymptotic analysis. However, the analysis of variance estimation will be too complicated to analyze.
> 2. Our technique is based on the information-theoretic lower bound provided by Kaufmann et al. (2016) and semiparametric efficiency bounds. Both techniques are used for asymptotic analysis. Therefore, we need to develop completely different approaches for non-asymptotic analysis.
> 3. We deal with general distribution by approximating the KL divergences. If we focus on the non-asymptotic analysis, we need to restrict the class of distributions to specific distributions, such as the Gaussian distribution, even if possible.
>
> One of the promising approaches for lower bounds is to employ lower bounds provided by Carpentier and Locatteli (2016). However, this lower bound is based on the boundedness of $Y^a_t$. Without the boundedness, non-asymptotic analysis would be more difficult. Additionally, the definitions of optimality might be changed to deal with the uncertainty of variance estimation. Thus, although the non-asymptotic upper bound has been derived, deriving non-asymptotic lower bounds requires different techniques and is not straightforward, even if possible. We will explain this open issue in the next update.
>
> Carpentier, A. and Locatelli, A. Tight (lower) bounds for the fixed budget best arm identification bandit problem, 2016.
>
> -------------
>
> **Q2**.
> > It is unclear how poor estimations at the early stage and for more general distributions influence non-asymptotic performance of the AS-AIPW.
>
> **A2**.
> As we answered in A1 and you pointed out, the non-asymptotic performance will be affected by variance estimation. We showed the non-asymptotic upper bound in Appendix H by assuming the convergence rate of the variance estimation. Unless we assume that a specific convergence rate is given for the variance estimation or variances are known, it is difficult to derive asymptotic results. We will explain this limitation more clearly in the next update.
>
> -----------
>
> **Q3**.
> > The experimental results verify the above concern...
>
> **A3**.
> We appreciate your suggestion. We will add additional experimental results in the next update using different distributions with various variances.
>
> -----------
>
> **Q4**.
> > define w(a|x) in Theorem 3.4
>
> **A4**.
> It is defined as a element of $\mathcal{W}$, which is defined in the above of Theorem 3.4.
>
> -----------
>
> **Q4**.
> > Can you discuss Barrier et al 2023?
>
> **A4**.
> First of all, while we consider the expected simple regret minimization, that work considers minimization of the probability of misidentification. These two settings are related but require different analyses for lower bounds. See Komiyama et al. (2023).
>
> Additionally, there are the following critical differences between our study and theirs in non-parametric analysis using the KL divergence of Kaufmann et al. (2016).
>  - In the lower bound, our study approximates the KL divergence by the Fisher information (semiparametric influence function) around the gaps between the best and suboptimal arms are zero $\Delta^a(P) \to 0$. We do not assume the boundedness of $Y^a_t$. In the upper bound, we utilize the central limit theorem for deriving tight results.
> - Barrier et al. (2023) assumes the boundedness of $Y^a_t$. Then, bounding the KL divergences using the boundedness without using the small gap (fixed $\Delta^a(P)$).
>
> Because we employed the worst-case analysis, which implies $\Delta^a(P) \approx 1/\sqrt{T}$ (see Section 3), we could naturally develop non-parametric results. However, we cannot employ such an approximation in Barrier et al. (2023) because it considers lower and upper bounds under a fixed $P$ or fixed $\Delta^a(P)$. Furthermore, unlike the expected simple regret minimization, the optimality in minimization of the probability of misidentification is still an open issue. Also see Komiyama et al. (2023) and Qin (2022). We remark on these differences in the next update.
>
> Komiyama, J., Ariu, K., Kato, M., and Qin, C. Rate-optimal bayesian simple regret in best arm identification, 2023.
> Qin, C. Open problem: Optimal best arm identification with fixed-budget, 2022.
>
> -----------
>
> **Q5**.
> > in Figure 1, can you also draw the standard deviation?
>
> **A5**.
> Thank you for your suggestion. We will add the standard deviations or the boxplot in the next update.

---

### Official Review · Reviewer_65vH · 2023-10-28

**Soundness:** 3 good
**Presentation:** 4 excellent
**Contribution:** 3 good
**Rating:** 6
**Confidence:** 3

**Summary:**

This paper considers the fixed-budget best-arm identification problem in (contextual) multi-armed bandits and takes simple regret minimization as the objective. It first derives an asymptotic minimax lower bound that depends on the variance of the reward distribution. Then, it proposes an algorithm called **AS-AIPW** that nearly achieves this lower bound asymptotically. Finally, sanity check experiments are also provided to validate the effectiveness of the proposed algorithm.

**Strengths:**

- The variance dependent asymptotic lower bound is considered to be highly novel.
- Under specific scenarios, the optimal allocation strategy has closed-form expression.
- The proposed algorithm nearly achieves the lower bound.

**Weaknesses:**

One weakness is that the current Theorem 3.8 in this paper does not generalize to the case with $K\geq 3$ and it is not clear whether the difficulty is technical or fundamental.

Another weakness is that most provided experiment results do not show advantages of **AS-AIPW** over variance-unaware algorithms. Although the paper conjectures that the superiority of **AS-AIPW** can only appear when $K$ is small, from my perspective, the number of arms should not be the essential factor. In particular, the worst-case regret of variance-unaware algorithms scales with the magnitude of the reward while that of **AS-AIPW** scales with the standard deviation of the reward. Therefore, if my understanding is correct, the advantage of **AS-AIPW** should appear if we run it on a hard instance with large reward magnitude but small variance. Is that possible to design and run experiments on such an instance?

### Suggestions on Writing
- 3rd line of page 7, "$w^*(a\vert X_t)$" -> "$(\sigma^a_t(X_t))^2$".
- There is probably no need to explain how to sample an arm $a$ with probability $\widehat{w}_t(a\vert X_t)$ at the beginning of Section 4.2.
- The main context can contain a sketch of techniques used for proving the lower bound and a brief discussion of its novelty.

**Questions:**

- In the sixth requirement of Definition 3.2, does "$\mu^a(P)(x)\rightarrow \mu^a(P^\sharp)$" means that there exists a sequence of bandit models $\lbrace P_n\rbrace$ such that $\lim_{n\rightarrow\infty} \mu^a(P_n)(x)= \mu^a(P^\sharp)$?
- Based on the given results, it seems quite tempting to conjecture that Theorem 3.8 can be generalized to the case with $K\geq 3$. Is this fundamentally not doable or does it just require more sophisticated techniques?
- What are the disadvantages of using $\arg\max_{a\in[K]}\widehat{\mu}^a_T$ as the arm recommendation rule? Do these disadvantages exist when there is no context information?
- If my understanding is correct, when there is no context information, we have $\widehat{\mu}^a_t=\frac{1}{t}\sum_{s=1}^{t}\mathbf{1}\lbrace A_s=a\rbrace Y_s$. Then, it looks weird that $\widehat{\mu}^{\mathrm{AIPW}, a}\_T$ contains the term $$\frac{1} {T}\sum_{t=1}^{T}\widehat{\mu}^a_t=\frac{1}{T}\sum_{t=1}^{T}\left(\sum_{s=t}^{T}\frac{1}{s}\right)\mathbf{1}\lbrace A_t=a\rbrace Y_t,$$ since it means that more weights are explicitly put on earlier samples. Why will this happen?
- Suppose there are not many possible contexts and we can encounter each contexts for sufficiently many times, can we treat **AS-AIPW** as doing BAI for each context independently? That is, if we define $$\widehat{a}^{\mathrm{AIPW}}\_T(x)=\arg\max_{a\in[K]}\frac{1}{T}\sum_{t=1}^{T}\mathbf{1}\lbrace X_t=x\rbrace \varphi^a_t(Y_t, A_t, X_t),$$ can we bound the simple regret condition on $X=x$ by $$\max_{a, b\in[K]: a\neq b}\sqrt{\log(K)\left(\frac{(\sigma^a(x))^2}{w^*(a\vert x)}+\frac{(\sigma^b(x))^2}{w^*(b\vert x)}\right)}+o(1)?$$

---

> ### Author Response · Authors · 2023-11-16
> **Re: Official Review of Submission9273 by Reviewer 65vH**
>
> Thank you for your insightful comments. We updated our manuscript based on your feedback. We also make additional sections in the Appendix to answer the technical questions the reviewer raised (Q2--Q5 below) during this phase.
>
> Our brief replies to your comments are listed as follows.
>
> ---------------------------
> **Q1**.
> > Does "$\mu^a(P)(x)\to \mu^a(P^\sharp)$" means that there exists a sequence of bandit models $\{P_n\}$ such that $\lim_{n\to\infty}\mu^a(P_n)(x) = \mu^a(P^\sharp)$?
>
> **A1**.
> We appreciate your comment. Yes, we assume the existence of such a sequence of bandit models $\{P_n\}$. We will update our manuscript following your more rigorous definition.
>
> ---------------------------
>
> **Q2**.
> > It seems quite tempting to conjecture that Theorem 3.8 can be generalized to the case with $K\geq 3$.
>
> **A2**.
> We agree with the reviewer's comment and also consider that extending Theorem 3.8 for $K\geq 3$ is an important open issue. To address this issue, we believe that current assumptions for distributions and restrictions for strategies are insufficient. We need to add more assumptions and develop additional tools for this problem.
>
> For example, the framework of the limit of experiments (van der Vaart, 1998) is one of the promising directions. This framework is expected to allow us to derive tight lower and upper bounds by using the asymptotic normality. However, to apply the results, we need to restrict strategy class and underlying distribution appropriately.
>
> Recently, Kato (2023) derived results similar to Theorem 3.8 for $K \geq 3$ in minimization of the probability of misidentification ($\mathbb{P}_P(\hat{a}_T \neq a^*(P))$). However, the results are limited to cases with known variances, minimization of the probability of misidentification, and Gaussian distributions. It is an open issue how we derive similar results for expected simple regret minimization.
>
> Kato (2023), Locally Optimal Best Arm Identification with a Fixed Budget.
>
> We summarize this problem as an open issue.
>
> ---------------------------
>
> **Q3**.
> > What are the disadvantages of using $\arg\max_{a\in[K]}\hat{\mu}^a_T$?
>
> **A3**..
> We conjecture that both $\hat{\mu}^a_T$ and $\hat{\mu}^{AIPW, a}_T$ have almost the same theoretical properties when there is no contextual information. However, proving the regret upper bound for $\hat{\mu}$ requires a more complicated proof procedure compared to $\hat{\mu}$ or requires some additional assumptions. This problem is related to theories of empirical process and martingales, as discussed in Hirano et al. (2003) and Hahn et al. (2011).
>
> We believe that showing it is not straightforward and an open issue, even though it is probably possible . We also summarize this problem as an open issue in another reply or the Appendix.
>
> ---------------------------
>
> **Q4**.
> > It looks weired that $\hat{\mu}^{AIPW}$ contains the term $\frac{1}{T}\sum^T_{t=1}\hat{\mu}^a_t$.
>
> **A4**.
> This term is introduced for variance reduction with keeping the martingale properties for $\frac{1}{T}\sum^T_{t=1}\hat{\mu}^a_t$. Here, an effect of $\frac{1}{T}\sum^T_{t=1}\hat{\mu}^a_t$ vanishes very quickly because of the existence of the product
> $|\hat{w}(a) - w^*(a)||\hat{\mu}^a(P) - \hat{\mu}^a_t|$ in theoretical analysis, which accelerates the vanish of the estimation error of $w^*$ and $\mu^a(P)$. That is, the introduction of $\frac{1}{T}\sum^T_{t=1}\hat{\mu}^a_t$ reduces the variance of the main estimator without increasing bias (asymptotically). This technique is used in different literatures, such as van der Laan (2008) and Chernohukov et al. (2018), which are recently called double machine learning.
>
> This property is related to the Q3 the reviewer raised. We add more explanations on this problem in the Appendix.
>
> ---------------------------
>
> **Q5**.
> > Suppose there are not many possible contexts and we can encounter each contexts for sufficiently many times, can we treat AS-AIPW as doing BAI for each context independently?
>
> **A5**.
> Yes. We can conduct context-specific treatment recommendation is possible when there are not many discrete contextual information.
>
> We are now extending the result for policy learning with BAI; that is, given a set $\Pi$ of policies $\pi:[K]\times\mathcal{X} \to (0, 1)$ such that $\sum_{a\in[K]}\pi(a|x) = 1$ (and potentially continuous contextual information), we train a policy $\pi\in\Pi$ to minimize the regret. Consider we restrict a policy class $\Pi$ to the one such that we discretize the contextual information and recommend an arm within each discretized contextual information. Then, such a strategy aligns with the strategy that the reviewer suggested.
>
> The remaining open issue is how we bound the regret for general $\Pi$. Because samples are non-i.i.d., we cannot directly apply the standard complexity measure such as the Rademacher complexity. This open problem has garnered attention in this literature, and we summarize it in the next update.

---

> > ### Comment · Reviewer_65vH · 2023-11-17
> > **Response**
> >
> > Thank you very much for your reply! Could you also update the pdf and color the added details that you plan to elaborate (such as more discussions about the estimator and context-specific BAI)?
> >
> > Meanwhile, could you also discuss what your opinion is about the second weakness mentioned in my initial review?

---

> > > ### Author Response · Authors · 2023-11-20
> > > **Re: Response**
> > >
> > > Thank you for your reply. We will revise the manuscript as soon as possible.
> > >
> > > We reply to the following question.
> > >
> > > **Q**.
> > > > Another weakness is that most provided experiment results do not show advantages of AS-AIPW over variance-unaware algorithms. Although the paper conjectures that the superiority of AS-AIPW can only appear when $K$ is small, from my perspective, the number of arms should not be the essential factor. In particular, the worst-case regret of variance-unaware algorithms scales with the magnitude of the reward while that of AS-AIPW scales with the standard deviation of the reward. Therefore, if my understanding is correct, the advantage of AS-AIPW should appear if we run it on a hard instance with large reward magnitude but small variance. Is that possible to design and run experiments on such an instance?
> > >
> > > **A**.
> > > We appreciate your suggestion.
> > >
> > > First, we can run an experiment with a large reward magnitude but a small variance. We will add the experiment in the camera-ready if it cannot be finished during this rebuttal phase.
> > >
> > > Second, we believe that there are two effects.
> > >
> > > > the worst-case regret of variance-unaware algorithms scales with the magnitude of the reward while that of AS-AIPW scales with the standard deviation of the reward
> > >
> > > We conjecture that our proposed strategy outperforms existing methods in experiments with harder instances (smaller gaps) with larger $T$.
> > >
> > > >  the advantage of AS-AIPW should appear if we run it on a hard instance with large reward magnitude but small variance.
> > >
> > > Additionally, we agree with your comment above. In existing studies, such as economics and epidemiology, variance-aware adaptive experiments are often employed in situations with large reward magnitude. Therefore, such settings are more suitable for our algorithm.
> > >
> > > We will conduct experiments and add discussion in the future update (if possible, we will finish it during the rebuttal phase).

---

> > > > ### Comment · Reviewer_65vH · 2023-11-23
> > > > **Response**
> > > >
> > > > Thanks for the response. My concerns on the theoretical side are mostly addressed. However, I do believe it's important to have some empirical evidence showing the advantage of being variance-aware under certain scenarios. Nevertheless, I'm still inclilned to acceptance given the theoretical contribution. Therefore, I decide to lower my score to 6.

---

### Official Review · Reviewer_ZfPN · 2023-11-02

**Soundness:** 3 good
**Presentation:** 1 poor
**Contribution:** 3 good
**Rating:** 5
**Confidence:** 3

**Summary:**

The authors propose a work in the field of fixed-budget best arm identification for multi-armed bandits when contextual information is available, with the goal of minimizing the expected simple regret.
The authors derive asymptotic lower bounds on the expected simple regret depending on the variances of the potential outcomes rather than considering outcomes with bounded supports. The lower bound is provided in both the cases in which contextual information is available and when it is not.
Moreover, they provide an algorithm, namely AS-AIPW, showing that it matches (asymptotically) the lower bound.
Finally, the authors present a numerical validation of the presented results just on synthetic data.

**Strengths:**

The proposed work faces the problem of best arm identification in MABs. The authors discuss the theoretical differences when contextual information are available or not.
Moreover, the work presents two asymptotic lower bounds (one with and the other without contextual information) and an algorithm, which asympotically matches the lower bounds.

The analysis seems to be done properly, but I have not checked the correctness of all the proofs.

**Weaknesses:**

A weakness I found in the paper is linked to the feeling that the authors did not pay great attention to the details.

Indeed:
- The abstract is not so clear since it does not introduce the fact that the setting at hand will consider contextual information and that it will compare it with the case in which contextual information is not available;
- The introductory section is not clear;
- I would appreciate (at least a paragraph) on some motivating examples (also in the appendix) with some comments.
- Some crucial quantities are not commented on, such as the meaning of the AIPW estimator.

In the experimental section (even in Appendix I) I found too simple experimental settings. Even if I have appreciated the comparison with fixed-budget BAI when no contextual information is available, I do not believe that it is fair to compare the same baselines (that are not thought for contextual settings) with the proposed algorithm. I suggest employing at least another algorithm thought for the same setting proposed by the authors if present (and if no competitors are available, please write it).

**Questions:**

Besides the concerns related to the "weaknesses" section, here are other questions:
1. is it possible to adapt the lower and upper bounds not to be asymptotic?
2. will you release the code of the experiments to assess if the numerical validation is reproducible?

---

> ### Author Response · Authors · 2023-11-15
> **Re: Official Review of Submission9273 by Reviewer ZfPN**
>
> We appreciate the reviewer's detailed feedback. A revised manuscript will be submitted shortly. Below is a preliminary response to the comments:
>
> Before answering each question the reviewer raised, we remark that our research addresses both scenarios: with and without contextual information. We have:
> - Developed the best arm identification (BAI) strategy that leverages variances.
> - Proposed a BAI strategy that can utilize contextual information in fixed-budget BAI.
> The novelty lies in using variances (first point) and utilizing contextual information to BAI (second point). Even without contextual information, our study has a novelty in proposal BAI strategies using the variances. Our results cover both scenarios with and without contextual information, as non-contextual cases are inherently included in the contextual ones. Page limitation precluded separate discussions for each setting. However, on page 2, we noted, "Note that this setting is a generalization of fixed-budget BAI without contextual information, and our result holds novelty even in the absence of contextual information."
>
> We answer each question below.
>
> -------------------------------
>
> **Q1**.
> > The abstract is not so clear since it does not introduce the fact that the setting at hand will consider contextual information and that it will compare it with the case in which contextual information is not available;
>
> **A1**.
> We do **not** focus on comparing our method with contextual information with existing methods with contextual information. Our study proposes strategies for **both** cases with and without contextual information. In Section 3.4 and Section 7, we compare our method with existing methods in a case where we cannot use contextual information in both methods.
>
> -------------------------------
>
> **Q2**.
> > Even if I have appreciated the comparison with fixed-budget BAI when no contextual information is available, I do not believe it is fair to compare the same baselines (not thought for contextual settings) with the proposed algorithm.
>
> > I do not believe that it is fair to compare the same baselines (that are not thought for contextual settings) with the proposed algorithm.
>
> **A2**.
> In our main text experiments, **we compare our method with existing methods, both without using contextual information,** as stated in the third line of Section 7: "We investigate two setups with K = 2, 3 without contextual information for these strategies." This ensures a fair comparison.
>
> In the Appendix, we compare our method that uses contextual information with existing methods that do not use contextual information.
>
> -------------------------------
>
> **Q3**.
> > I suggest employing at least another algorithm thought for the same setting proposed by the authors if present (and if no competitors are available, please write it).
>
> **A3**.
> To the best of our knowledge, this setting has no appropriate competitors. In BAI, it is still unclear how we formulate the problem if contextual information is available, and it is an open issue. We will make it clearer in the next update.
>
> -------------------------------
>
> **Q4**.
> > Is it possible to adapt the lower and upper bounds not to be asymptotic?
>
> **A4**.
> Adapting the lower and upper bounds to non-asymptotic conditions seems unfeasible. The finite-sample analysis will be more complicated by the estimation error of the variances, which can be ignored in an asymptotic analysis. Our lower bound is based on the semiparametric efficiency bound, an extension of the Cramer-Rao lower bound, typically used for asymptotic optimality. Addressing finite-sample optimality would require novel and other methodologies.
>
> -------------------------------
>
> **Q5**.
> > will you release the code of the experiments to assess if the numerical validation is reproducible?
>
> **A5**.
> We plan to release our experiment code, ensuring it complies with the review process's anonymity requirements. It will be available during the rebuttal phase.
>
> -------------------------------
>
> **Q6**.
> > Some crucial quantities are not commented on, such as the meaning of the AIPW estimator.
>
> **A6**.
> We explained the details about the AIPW estimator in Section 4.3.  The name comes from the augmentation of the inverse probability of weighting estimator. We add some brief history of this estimator in the next update.
>
> -------------------------------
>
> Lastly, our study contributes to two open problems in BAI:
> - Integrating variances into BAI.
> - Utilizing contextual information in BAI.
> We approach these with asymptotic lower bounds via semiparametric analysis and a BAI strategy focusing on the highest marginalized expected reward across contextual distributions. The first point alone stands as an independent contribution. Although we show an approach for the second issue, how we use contextual information is still an open issue. Also, see our Russac et al. (2021) and our rebuttal to the Reviewer 65vH.

---

> > ### Comment · Reviewer_ZfPN · 2023-11-21
> >
> > Thank you for your response.
> >
> > I think that the problem faced is interesting and that the paper, technically, presents a good contribution. I agree that a lower bound depending on the time budget is hard to be derived in the setting you are considering.
> > However, my points are related to the fact that all the strengths of your work do not emerge clearly. For instance, the fact that you face both settings with and without contextual information is a strength of the work, thus it should also be highlighted in the abstract.
> > Moreover, the motivating examples are just cited and not discussed properly. I would like to see at least one truly motivating example fully explained.
> >
> > I remark that I appreciate the work itself, but the presentation should be improved.

---

> > > ### Author Response · Authors · 2023-11-22
> > > **Re: Official Comment by Reviewer ZfPN**
> > >
> > > Once again, we deeply appreciate your constructive comments. Following your advice, we will make our contribution more explicit in the Abstract and other sections, as we currently mention that "Note that this setting is a generalization of fixed-budget BAI without contextual information, and our result holds novelty even in the absence of contextual information." in Introduction. In future updates, we will add some examples to the Appendix, in the rebuttal phase, otherwise in the camera-ready.

---

### Author Response · Authors · 2023-11-20
**Revision**

Thank you to all reviewers for your constructive comments.

We have temporarily updated the manuscript to reply several questions raised by Reviewers 65vH and qfgN (we colored them red). In particular, we added Appendix J to reflect the open problems raised by Reviewers 65vH and qfgN. These updated parts will be more refined until the camera-ready.

We will update the manuscript as soon as possible in response to Reviewers ZfPN, and dvWZ, in addition to the remaining replies to  Reviewers 65vH and qfgN.

---

> ### Author Response · Authors · 2023-11-22
> **Revision (2)**
>
> We thank you again for your constructive suggestions. We have revised the manuscript again to reflect the reviewers' comments. Due to time constraints, we have not been able to fully reflect all of the reviewers' comments. We will continue to update the manuscript based on the comments until the rebuttal phase is complete or camera-ready.

---

### Meta-Review · Area_Chair_A34m · 2023-12-06

**Metareview:**

This paper studies best-arm identification (BAI) with heterogeneous reward variances. The proposed algorithm is analyzed and evaluated empirically. The pluses and minuses of the paper are:

* **Clarity:** Some concepts are not clearly introduced and explained. For instance, the word "contextual" is not in the abstract.

* **Lower bound:** This goes beyond the closest related work of Lalitha et al. (2023), which does not prove any lower bound. The lower bound is asymptotic. I think that a finite-sample lower bound would make more sense in the fixed-budget setting, which is motivated by low budget. See Carpentier and Locatelli (2016), which was also discussed in the rebuttal.

* **Contextual algorithm and analysis:** This goes beyond the closest related work of Lalitha et al. (2023), which does not consider context. The problem is that the context is trivial: a finite number of contexts.

* **Experiments:** Too simple and do not clearly demonstrate the benefit of the proposed method.

With the above in mind, this paper is a borderline and can go either way.

**Justification For Why Not Higher Score:**

If the proposed algorithm is empirically evaluated, it should perform well. This paper has mixed empirical results. The theory can also be improved and there is no clear evidence that it cannot be.

**Justification For Why Not Lower Score:**

N/A

---

### Decision · Program_Chairs · 2024-01-16

Reject